# Review Article: Dynamical Systems, Algebraic Topology, and the Climate Sciences

Michael Ghil [1,2,3] and Denisse Sciamarella [4]

[1]Geosciences Department and Laboratoire de Météorologie Dynamique (CNRS and IPSL), École Normale Supérieure and PSL University, 75231 Paris Cedex 05, France
[2]Department of Atmospheric & Oceanic Sciences, University of California at Los Angeles, Los Angeles, CA 90095-1567, USA
[3]Departments of Mathematics and of Finance, Imperial College London, London SW7 2BX, UK
[4]Institut Franco-Argentin d'Études sur le Climat et ses Impacts (IFAECI) International Reseach Laboratory 3351 (CNRS - IRD - CONICET - UBA) C1428EGA CABA, Argentina

**Correspondence:** ghil@atmos.ucla.edu; denisse.sciamarella@cnrs.fr

**Abstract.** The definition of climate itself cannot be given without a proper understanding of the key ideas of long-term behavior of a system, as provided by dynamical systems theory. Hence, it is not surprising that concepts and methods of this theory have percolated into the climate sciences as early as the 1960s. The major increase in public awareness of the socio-economic threats and opportunities of climate change has led more recently to two major developments in the climate sciences: (i) the Intergovernmental Panel on Climate Change's successive Assessment Reports; and (ii) an increasing understanding of the interplay between natural climate variability and anthropogenically driven climate change. Both of these developments have benefitted from remarkable technological advances in computing resources, in throughput as well as storage, and in observational capabilities, regarding both platforms and instruments.

Starting with the early contributions of nonlinear dynamics to the climate sciences, we review here the more recent contributions of (a) the theory of nonautonomous and random dynamical systems to an understanding of the interplay between natural variability and anthropogenic climate change; and (b) the role of algebraic topology in shedding additional light on this interplay. The review is thus a trip leading from the applications of classical bifurcation theory to multiple possible climates to the tipping points associated with transitions from one type of climatic behavior to another in the presence of time-dependent forcing, deterministic as well as stochastic.

## 1 Introduction and motivation

This paper is based on the invited talks given by the two authors in an online series on "Perspectives on climate sciences: From historical developments to research frontiers." The series had twice-monthly talks from July 2020 to July 2021 and its success led to the idea of having a special issue of *Nonlinear Processes in Geophysics*. The talks of the two co-authors are available at https://youtu.be/xjccOfptYII (MG) and https://youtu.be/W1yndTsvR0g (DS), respectively. In the present paper, we go beyond the lively but more perishable video version to what we hope is a more coherent and permanent record of the convergence between two strains of Henri Poincaré's heritage — dynamical systems theory (Poincaré, 1892, 1893,

1899, 2017) and algebraic topology (Poincaré, 1895; Siersma, 2012) — and their joint applications to the climate sciences. This convergence resulted from the two authors meeting in November 2018 at the University of Buenos Aires, where MG gave a series of six lectures on "Mathematical Problems in Climate Dynamics" at the invitation of DS; see Ghil (2021a, b, c, d, e).

## 1.1 Dynamical systems and climate dynamics

Many of the ideas and methods of dynamical systems theory were introduced into the climate sciences by a generation of pioneers in the 1960s. Stommel (1961) formulated a two-box and two-pipe model for the oceans' overturning circulation that had two stable, steady-state solutions with counter-rotating flows. Veronis (1963) used another form of reduced-order model, by projecting a one-layer, single-gyre wind-driven circulation model onto a small number of Fourier modes and likewise found multiple solutions, both steady and periodic. Most intriguingly, Lorenz (1963a) applied the same type of low-truncation Galerkin method to a Boussinesq model of flow between two horizontal plates, heated from below and cooled from above. In this setting, transitions from a quiescent fluid to two mutually symmetric flow patterns and on to chaotic solutions occurred. The latter, particularly simple convection model governed by only three ordinary differential equations (ODEs) provided inspiration for hundreds of papers on deterministically chaotic phenomena in the climate sciences and way beyond.

None of the pioneering papers mentioned above, though, nor any of the thousands of papers since, exhibits all the phenomena — mathematical and physical — of interest in this review paper. As we proceed, the illustrative examples will be taken from atmospheric, oceanographic and climate models that capture best one or a few of these phenomena.

It is important to realize that Poincaré already had seen the analogy between the chaos he found in the so-called reduced three-body problem of celestial mechanics (Poincaré, 1892, 1893, 1899; Gray, 2013; Poincaré, 2017) and the "sensitive dependence on intitial conditions" that he realized occurred in the evolution of the weather. In fact, in Book I, Ch. IV of Poincaré (1908), called "Le Hasard," he states that "it may happen that small differences in the initial conditions produce very great ones in the final phenomena." And his second example of sensitive dependence is weather:

"Our second example will be very analogous to the first and we shall take it from meteorology. Why have the meteorologists such difficulty in predicting the weather with any certainty? Why do the rains, the tempests themselves seem to us to come by chance, so that many persons find it quite natural to pray for rain or shine, when they would think it ridiculous to pray for an eclipse? We see that great perturbations generally happen in regions where the atmosphere is in unstable equilibrium. The meteorologists are aware that this equilibrium is unstable, that a cyclone is arising somewhere; but where they can not tell; one-tenth of a degree more or less at any point, and the cyclone bursts here and not there, and spreads its ravages over countries it would have spared. This we could have foreseen if we had known that tenth of a degree, but the observations were neither sufficiently close nor sufficiently precise, and for this reason all seems due to the agency of chance. Here again we find the same contrast between a very slight cause, unappreciable to the observer, and important effects, which are sometimes tremendous disasters." The translations are both from Poincaré (2003, Book I, Ch. IV, Chance).

The work of Lorenz (1963a), while actually referring to a highly simplified model of thermal convection, illustrates perfectly Poincaré's insights about the role of what we now call deterministic chaos, rather than pure chance. Ghil and Childress (1987) presented the applications of dynamical systems theory to large-scale atmospheric and climate dynamics, as well as to dynamo

theory and geomagnetism, in a systematic book form; they included a gradual introduction to the basic mathematical concepts and tools involved, and this book was reissued by Springer in 2012 as an eBook. Dijkstra (2013) provided a considerably expanded version of Ghil and Childress (1987), in terms of both the mathematical content and the areas of climatic applications, which include oceanographic and coupled ocean–atmosphere phenomena.

Aside from its applications to the climate sciences, the dynamical systems literature is quite extensive, in covering both mathematical fundamentals and applications to other areas. Holmes (2007) provides a fine historical overview of the field from 1885 to 1965, along with a fairly complete bibliography. Two important books are Arnol'd (2012) and Guckenheimer and Holmes (1983). Further references — including some that treat the subject for infinite-dimensional function spaces, like those that describe the solutions of the partial differential equations of fluid dynamics — are given in the subsequest list of

noteworthy insights that dynamical systems theory has provided for the climate sciences.

Following Ghil et al. (1991) and Ghil (2019), we summarize herewith some key insights in the climate sciences due to the theory of autonomous dynamical systems, in which time-dependent forcing or coefficients are absent.

1. The equations of continuum mechanics are nonlinear. Surprisingly many phenomena can be explained by linearization about a particular fixed basic state. Many more cannot.

2. Behavior of solutions to nonlinear equations — subject to some reasonable mathematical assumptions — changes qualitatively only at isolated points in phase-parameter space, called bifurcation points. Behavior along a single branch of solutions, between such points, is modified only quantitatively and can be explored by linearization about the basic state, which changes as the parameters change. That is, nonlinear dynamics is much like linear dynamics, only more so (Lorenz, 1963a, b; Ghil and Childress, 1987).

3. Bifurcation trees lead from the simplest, most symmetric states, to highly complex and realistic ones, with much lower symmetry in either space or time or both. These trees can be explored partially by analytic methods (Jin and Ghil, 1990; Jordan and Smith, 2007) and more fully by numerical ones, such as pseudo-arclength continuation (Legras and Ghil, 1985; Dijkstra, 2005).

4. The truly nonlinear behavior near bifurcation points involves robust transitions, of great generality, between single and
multiple fixed points (saddle-node, pitchfork and transcritical bifurcations), fixed points and limit cycles (Hopf bifurcation), limit cycles and strange attractors ("routes to chaos": Eckmann, 1981; Guckenheimer and Holmes, 1983). As the complexity of the behavior increases, its predictability decreases (Ghil, 2001).

5. Behavior in the most realistic, chaotic regime can be described by the ergodic theory of dynamical systems. In this regime, statistical information similar to, but more detailed than for truly random behavior, can be extracted and used for
predictive purposes (Eckmann and Ruelle, 1985; Mo and Ghil, 1987; Ghil and Robertson, 2000).

6. Chaos and strange attractors are not restricted to low-order systems. They can be shown to exist for the full equations governing continuum mechanics (Constantin et al., 1989; Temam, 2000). The detailed exploration of finite- but

high-dimensional attractors is in full swing (Legras and Ghil, 1985; Dijkstra, 2005; Simonnet et al., 2009; Doedel and Tuckerman, 2012; Dijkstra et al., 2014).

7. Single time series (Takens, 1981) and single numbers derived from them (e.g., Grassberger, 1983) have been used to describe chaotic behavior. This very simple and straightforward use of a nonlinear concept has attracted considerable attention to deterministically chaotic dynamics, including in the geosciences (Nicolis and Nicolis, 1984; Tsonis and Elsner, 1988). The use of single time series, while exciting in theory, is not very promising when the series are short and noisy (Ruelle, 1990; Smith, 1988). The increasing availability of a large number of similar series at different points in space, combined with physical insight, is compensating more and more for the shortcomings of each individual time series in describing the complexity of many phenomena in the geosciences, as well as advancing their prediction (Ghil et al., 2002).

Further details on the contributions of autonomous dynamical systems theory in general, and the concepts and methods of bifurcation theory in particular, appear in Sec. 2.1. The recent contributions of the theory of nonautonomous and random dynamical systems (NDSs and RDSs) — with their generalization of bifurcations to tipping points — are reviewed in Sec. 2.2.

## 1.2 Algebraic topology and chaotic dynamics

What is the topology of chaos and why is it important in the theory of dynamical systems and in the time series analysis for nonlinear and chaotic dynamics? We attempt here to provide answers to these questions, with an emphasis on applications to the climate sciences. Essentially, the concepts and tools of algebraic topology can be applied to the evolution of systems in both phase space and physical space, as well as to the interesting back-and-forth trip between the two spaces. This complementary view of the way that dynamics and topology interact is a main motivation of the present article.

The emphasis on time dependence and dynamics here should not allow us to forget, though, the huge role that homologies have already been playing in the fields of image processing and visualization (e.g., Heine et al., 2016; Singh Bansal et al., 2022, and references therein). Significant advances in computational topology (Edelsbrunner and Harer, 2022) have helped substantially in these more static area of applications and will clearly do so in the more dynamic ones contemplated herein.

In Sec. 3.1, we present the rather novel approach of Branched Manifold Analysis through Homologies (BraMAH) (Sciamarella and Mindlin, 2001; Charó et al., 2021b) for approximating the branched manifolds (Birman and Williams, 1983a, b) of dynamical systems by a cell complex that allows one to characterize the manifold by its homology group in phase space (Poincaré, 1895; Sciamarella and Mindlin, 1999). The detection and description of localized coherent sets (LCSs) in two-dimensional flows in physical space by BraMAH-based methods is reviewed in Sec. 3.2.

The most recent developments of the merging of the two strands of Poincaré's heritage, algebraic topology and dynamical systems, are covered in Secs. 3.3 and 3.4. In Sec. 3.3, we introduce the templex, a novel concept in algebraic topology (Charó et al., 2022a; Charó et al., 2022b), which complements the previously mentioned cell complexes of BraMAH by a directed graph (digraph), whose nodes are the cells and which approximates the flow on the branched manifold. The extension of this concept to the noise-perturbed chaotic attractors of RDS theory follows in Sec. 3.4.

We provide in the rest of this section a quick historical background to the current interest in the ways in which algebraic topology can help one infer a system's chaotic dynamics from one or more time series of its observables. The first methods of time series analysis that associated geometric properties with experimental time series appeared in the early 1980s (e.g., Packard et al., 1980). These geometric methods continue to be used, for instance, to analyze datasets of Lagrangian trajectories and understand the geometry of transport (Banisch and Koltai, 2017).

But is geometry the best lens one can use to classify data according to underlying differences in dynamics? Classifying dynamics is possible thanks to invariants or quasi-invariants in phase space. Gilmore (1998) classified invariants into three distinct categories:

- metric invariants: such as dimensions of various types, e.g., correlation dimension (Grassberger and Procaccia, 1983) or multifractal scaling functions (Halsey et al., 1986);

- dynamic invariants: such as Lyapunov exponents (Oseledec, 1968; Wolf et al., 1985), further discussed by (Eckmann and Ruelle, 1985; Abarbanel and Kennel, 1993); and finally

- topological invariants: linking numbers, relative rotation rates, Conway polynomials, and branched manifolds (Williams, 1974).

The first two kinds of invariants do not provide information on how to model the system's dynamics, while topological invariants actually do. Why is this so? Topology deals with the properties of a geometric object that do not change when continuous deformations are performed. Stretching, twisting, crumpling or bending preserve topology; while cutting or suturing holes, gluing separated pieces, or producing self-crossings do not. Volumes in phase space can be stretched or squeezed, folded or teared. The particular manner in which these processes are combined repetitively in phase space lead to a structure. The topology of such a structure is the signature of the mechanisms acting to build a certain dynamics.

The "recipe" to "knead" the Lorenz (1963a) strange attractor is illustrated in Fig. 1 as a sequence of steps that are topological in nature. Quoting Gilmore and Lefranc (2003), "sets of initial conditions (cubes) are sliced, by running into an axis with a stable and unstable direction (the $z$-axis for Lorenz-like systems), for example. The different parts flow off in different directions in the phase space, where they may encounter other sliced parts from different regions of the phase space. These are squeezed together and eventually return to regions they originated from (recursion)."

The advantage of using topology instead of geometry or fractality to describe chaos lies in the fact that topology provides information about the elementary stretching, folding, tearing or squeezing mechanisms that act in phase space to shape the flow. Geometric features may differ, but if the underlying dynamics obeys certain equivalence principles, the topology should be the same. Topological equivalence between branched manifolds is defined by isotopy. In other words, two objects are isotopic if it is possible to mold one into the other without tearing or gluing it. It is in this sense that we speak of dynamical equivalence. Different geometric deformations of the Lorenz attractor that preserve its topology are sketched in Fig. 2. There is a two-way correspondence between topology and dynamics, in a sense that will be clarified in Sec. 3.

A good starting point for this quick historical perspective is the pioneering paper of Henri Poincaré (Poincaré, 1895), who first described the way in which a dynamical system's properties depend upon its topology; see also (Gray, 2013, Ch. 8).

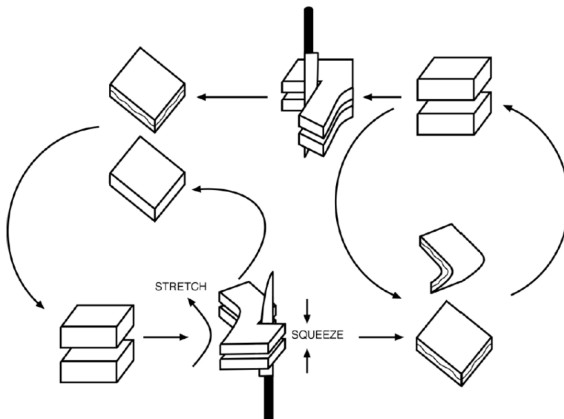

**Figure 1.** Sketch of the topological processes that intervene in obtaining the strange attractor of the Lorenz (1963a) convection model. From Letellier and Gilmore (2013) with permission by World Scientific Publishing Corp.

The concept of branched manifold, introduced by Robert F. Williams (1974), was anticipated in Edward N. Lorenz's famous convection paper: on Lorenz (1963a, p. 138), he remarks that "the [computed] trajectory is confined to a pair of surfaces which appear to merge in the lower portion of Fig. 3." The paper's Fig. 3 is reproduced here, coincidentally, as Fig. 3 as well. Lorenz plots the isopleths of $X$ as a function of $Y$ and $Z$ of the strange attractor, to approximate surfaces formed by all points on limiting trajectories. The etymology of "isopleth" combines "iso" with the ancient Greek word plêthos, "a great number," as in
the modern English word 'plethora'. It is generically used to refer to a curve of points sharing the same value of some quantity. We will return to a stochastically perturbed version of the Lorenz (1963a) model in Secs. 2.2 and 3.4.

Joan Birman and Robert F. Williams used branched manifolds to classify chaotic attractors in terms of the way periodic orbits are "knotted" in dynamical systems (Birman and Williams, 1983a, b). These authors discovered that systems whose

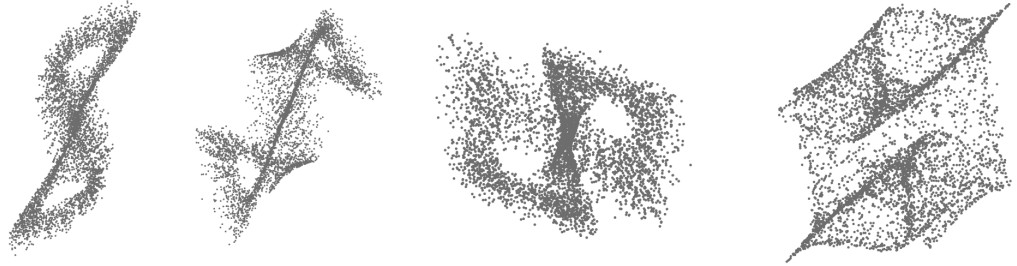

**Figure 2.** Point clouds associated with different geometrical representations of the Lorenz (1963a) attractor. They are obtained by integrating the model's governing equations using coordinate transformations for some of the variables. The butterfly is deformed but the topological structure of the butterfly is maintained.

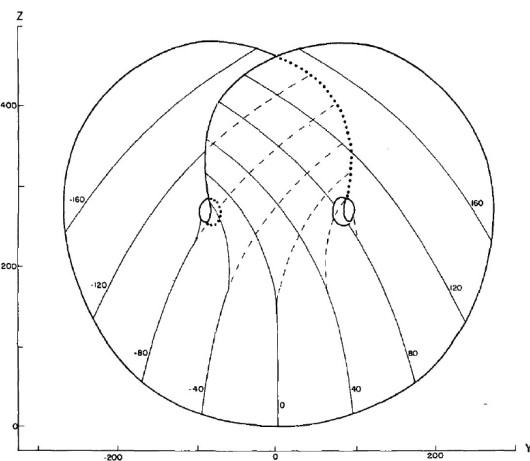

**Figure 3.** Isopleths of $X$ (thin solid curves) as a function of $Y$ and $Z$, based on a single trajectory of length 6 000 time steps of the Lorenz (1963a) attractor, and isopleths of the lower of the two values of $X$ where two values occur (dashed curves) for approximate surfaces formed by all points on nearby trajectories. The heavy solid curve and the extension of the dotted curves indicate natural boundaries of the surfaces. From Lorenz (1963a), published in 1963 by the American Meteorological Society.

branched manifolds have the same topology, are dynamically equivalent. From this discovery, a topologist's dream blooms: can one classify types of dynamics as one classifies the elements in Mendeleev's table?

In the late nineties, it became possible to determine whether or not two three-dimensional (3-D) dissipative dynamical systems are equivalent by using knot theory (Gilmore, 1998; Gilmore and Lefranc, 2003; Natiello et al., 2007; Letellier and Gilmore, 2013). In Sec. 3.1, we address the question of how these authors and many more worked with knots, the difficulties that arose with knot theory, and how the latter were solved, at least in part, using the homology groups of algebraic topology. Sections 3.2–3.4 describe the applications of BraMAH to the Lagrangian analysis of fluid flows in physical space; the introduction of digraphs to complement cell complexes in describing the flow on a branched manifold in phase space; and the extension of the templexes that describe the latter to noise-driven chaotic systems.

## 2   Dynamical System Theory for the Climate Sciences

As we indicated in Sec. 1.1, dynamical systems theory entered the evolution of the climate sciences — at that time consisting mainly of meteorology and oceanography — in the 1960s, in the pioneering papers of Lorenz (1963a, b); Veronis (1963); Stommel (1961) or others. These and other papers of the '60s and early '70s did not necessarily include explicit references to bifurcation theory, although awareness of the fundamental concepts and methods was clearly present, in one form or another. In the 1970s, another set of papers, on energy balance models (EBMs), reported on the possibility of alternative stable steady states, warm and cold, of Earth's climate system (Held and Suarez, 1974a; North, 1975). Ghil (1976a) specifically introduced

the saddle-node bifurcation into this climate setting, as well as numerical methods needed to deal with it in the context of a full partial differential equation model of climate, rather than of low-order or otherwise simplified models.

The contributions of "nonlinear dynamics," as dynamical systems theory tended to be referred to by physicists and other non-mathematicians by training, were presented for the first time in a quadrennial report (1987–1991) of the U.S. geosciences community to the International Union of Geodesy and Geophysics (IUGG) by Ghil et al. (1991). The presentation of elementary bifurcations below is for a broad audience and is based on Boers et al. (2022). It focuses on multistability and the possible transitions between different regimes of behavior: in Sec. 2.1 for systems with time-independent forcing and coefficients, and in Sec. 2.2 for systems in which time dependence is present in either the forcing or the coefficients or both.

## 2.1 Autonomous dynamical systems

Assume that the state of a system of interest can be described by a vector $\mathbf{x} \in \mathbb{R}^d$ and that the time evolution of $\mathbf{x}(t)$ is governed by the following equation of motion, namely a first-order autonomous ODE:

$$\dot{\mathbf{x}} = \mathbf{f}(\mathbf{x}; p); \tag{1}$$

Here $(\cdot)^{\cdot} = \mathrm{d}(\cdot)/\mathrm{dt}$, $\mathbf{f}$ denotes a generally nonlinear, smooth — i.e., continuously differentiable, up to some order — vector field and $p$ a scalar parameter or, in more general cases, a small set of parameters. For clarity, one separates the variables $\mathbf{x}$ from the parameter $p$ by a semi-colon. The term "autonomous" refers to the fact that in Eq. (1) both the coefficients and the forcing are constant in time. This means that changes in $p$ are assumed to be infinitely slow or, at least, very slow compared to the characteristic internal-variability times of the system being modeled.

Points $\mathbf{x}^*$ for which $\mathbf{f}(\mathbf{x}^*; p) = 0$ are called fixed points. Linearizing the equation of motion around a given fixed point $\mathbf{x}^*$ yields, for a small perturbation $\tilde{\mathbf{x}} = \mathbf{x} - \mathbf{x}^*$,

$$\dot{\tilde{x}} = \mathbf{f}'(\mathbf{x}^*; p)\tilde{\mathbf{x}}; \tag{2}$$

here $\mathbf{f}'(\mathbf{x}^*; p)$ is the Jacobian matrix comprised of the elements $\partial f_i / \partial x_j$. For an initial condition $\tilde{\mathbf{x}}_0$, the solution to this linearized equation is given by

$$\tilde{\mathbf{x}}(t) = e^{\mathbf{f}'t}\tilde{\mathbf{x}}_0 . \tag{3}$$

We call a fixed point $\mathbf{x}^*$ linearly stable if all eigenvalues of $\mathbf{f}'$ have negative real part, and linearly unstable otherwise. A scalar example will be given in Sec. 2.1.1.

The bifurcations of a dynamical system that we deal with in this subsection describe the creation and annihilation of fixed points, as well as changes in their linear stability. Further types of bifurcations are considered in the next subsection.

Typically, bifurcations lead to abrupt qualitative changes in the dynamics, explaining why they are often invoked as a mathematical model for abrupt regime shifts or state transitions in real-world systems. Until fairly recently, bifurcations were studied mostly in the context of autonomous dynamical systems. The more realistic situations in which the forcing is allowed to depend explicitly on time are addressed in Sec. 2.2. In this broader context, bifurcations have been called "tippings" in the climate sciences (Lenton et al., 2008; Ashwin et al., 2012; Kuehn, 2011; Ghil, 2019) and elsewhere.

There are at least two different interpretations of "tipping" and "tipping points" in the literature. One of these, emanating from Gladwell (2000) and Lenton et al. (2008), interprets tipping merely as a sudden change, whether due to a well-defined bifurcation or not. In this interpretation, a tipping point is merely a threshold. The other interpretation sees a tipping point as a

generalization to nonautonomous systems of a bifurcation point (Kuehn, 2011; Ghil, 2019).

In the latter case, tipping is necessarily related to a tipping point in phase-parameter space, as opposed to just a threshold in some parameter value; thus, not every jump or critical transition arises from a such a point. Both points of view — pun intended, of course — have their merits, but confusion should be avoided to the extent possible. Clearly, in this review article, we follow the more unambiguously defined mathematical version.

### 2.1.1   The double-well potential as a source of bistability

As an instructive and widely used example, we briefly introduce a prototype model to describe scalar dynamical systems than can occupy either one of two stable fixed points, separated by an unstable one, as plotted in Fig. 4a. The double-well potential $U(x; p) = x^4/4 - x^2/2 - px$ leads to the equation of motion

$$\dot{x} = U'(x; p) = -x^3 + x + p, \tag{4}$$

where $U' = \partial U/\partial x$. For $p < -p^*$ this dynamical system has only the stable fixed point $x_-^*$ and for $p > p^*$ it has only the stable fixed point $x_+^*$, while for $-p^* < p < p^*$ the two stable fixed points $x_\pm^*$ coexist and there is a third, unstable fixed point $x_0^*$ in-between these two.

The two stable fixed points correspond to the two minima of the potential $U$ above, whereas $\pm p^*$ are the two critical tresholds of the system. In this scalar case, the basins of attraction of the two minima are the intervals $-\infty < x < x_0^*$ and $x_0^* < x < +\infty$,

respectively. They are separated by the unstable fixed point $x_0^*$, which is a local maximum of the potential $U(x)$.

Changing $p$ slowly from, say, $p = -1$ to $p = +1$ will lead to a bifurcation-induced critical transition from $x_-^*$ to $x_+^*$ at the critical value $p = p^*$. When $p$ is subsequently changed back from $p = 1$ to $p = -1$, the transition from $x_+^*$ back to $x_-^*$ will only occur at $p = -p^*$. This phenomenon of jumps from one fixed point to the other occurring at distinct parameter values is called *hysteresis*, and it is highly relevant to the practical reversibility of abrupt transitions. It was studied, for instance, in

electromagnetic systems by J. C. Maxwell and by P. Curie, and it is important in the physical, biomedical, engineering, and socio-economic sciences. In the context of the climate sciences, a hysteresis loop like the one seen in Fig. 4a has been described in detail for EBMs by Ghil and Childress (1987, Ch. 10) and by Ghil (1994), using solar insolation as the parameter $p$.

The bifurcation introduced above is called a *double-fold bifurcation*, since it is obtained by combining a *supercritical* fold, with the stable branch reaching forward, to $p \to \infty$, with a *subcritical* one, with the stable branch reaching backward, to

$p \to -\infty$. A more recent version of such a double-fold bifurcation is plotted in Fig. 2 of Von der Heydt et al. (2016) for an energy balance model with respect to carbon dioxide concentration as the parameter $p$. The single-fold bifurcation is often called a saddle-node bifurcation, super- or subcritical, since in two dimensions it corresponds to the merging of a node that is stable in both directions on one branch with a saddle that is stable in one direction and unstable in the perpendicular direction,

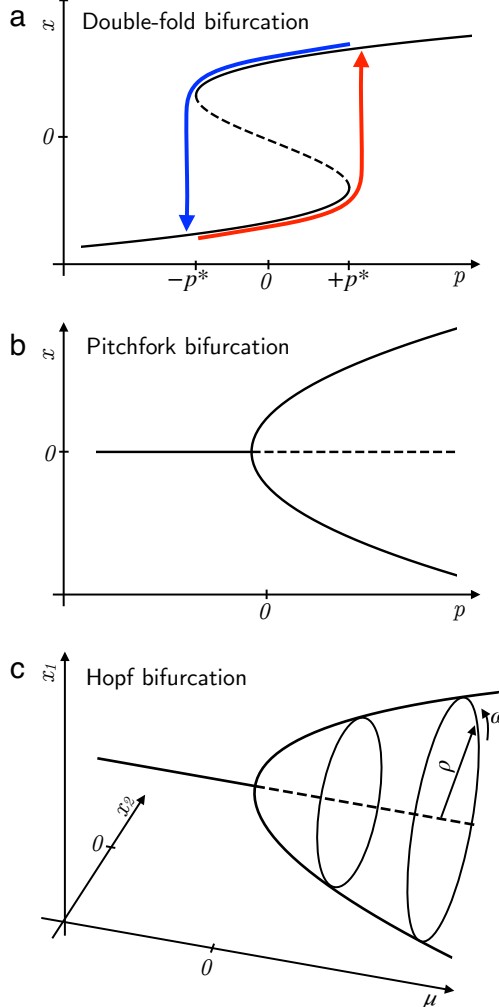

**Figure 4.** Bifurcation diagrams for (a) the double-fold; (b) supercritical pitchfork; and (c) supercritical Hopf bifurcation. Stable fixed-point branches are indicated by solid lines, unstable ones by dashed lines. The colored lines in panel (a) correspond to a hysteresis cycle. See text for details. After Boers et al. (2022) under CC-BY license.

on the other branch; see, for instance, Ghil and Childress (1987, Fig. 12.3) for sketches of the stability of fixed points for a linear autonomous ODE system in two dimensions.

### 2.1.2 Bistability in the presence of symmetry: the pitchfork bifurcation

Another example of bistability is given by a pitchfork bifurcation (Fig. 4b). Its so-called *normal form*, i.e., the simplest ODE that exhibits the change of behavior of interest, is

$$\dot{x} = x(p - x^2). \tag{5}$$

This bifurcation captures bistable behavior in systems in which spatial mirror symmetry prevails for low $p$-values. A well-known example in the climate sciences is symmetry in a meridional plane for an idealized Atlantic Meridional Overturning Circulation (AMOC) at low buoyancy forcing by a weak pole-to-equator temperature and precipitation gradient (Quon and Ghil, 1992; Ghil, 2001; Dijkstra, 2005). In this case, water sinks at both poles and rises on either side of the equator, forming two overturning cells that are symmetric with respect to the equatorial plane.

The solutions of Eq. (5) are $x_0 = 0, x_\pm = \pm\sqrt{p}$. In this normal form, the scalar symmetry of the latter two solutions with respect to $0$ stands for the mirror symmetry of the AMOC's overturning cells with respect to the equator.

The bifurcation occurs as the parameter $p$, which is a normalized form of the thermal and salinity forcing in the AMOC case, crosses over from negative to positive values. It is easy to check that, for $p \leq 0$, $x_0 = 0$ is the unique fixed point, while for $p > 0$ the three fixed points coexist. Their linear stability is given by considering infinitesimal perturbations around a given

steady-state solution $x = x_* + \xi$.

With the scalar version $f = f(x;p)$ of the notation in Eq. (1), we have the scalar version of Eq. (2) in the specific case at hand given by

$$(x_* + \xi)^{\cdot} = f(x_* + \xi;p) = 0 + \left.\frac{\partial f(x;p)}{\partial x}\right|_{x=x_*} \xi + \mathcal{O}(\xi^2) \simeq p - 3x_*^2.$$

Since $\dot{x}_* = 0$, this leaves

$$\dot{\xi} = p - 3x_*^2 \tag{6}$$

to determine linear stability for small $\xi$ (Ghil and Childress, 1987; Dijkstra and Ghil, 2005). Thus it is clear that, for $p < 0$, the unique solution $x_* = 0$ is linearly stable; but, for $p > 0$, this null solution becomes linearly unstable, while the two mutually symmetric solutions $x_* = x_\pm = \pm\sqrt{p}$ are stable, since $p - 3p < 0$. We thus suspect that, for sufficiently strong buoyancy forcing the two-cell AMOC will lose its stability and yield the approximately single-cell AMOC that is currently observed; see Stocker

and Wright (1991) or Quon and Ghil (1992), for instance.

### 2.1.3 Beyond bistability: Hopf bifurcation and limit cycles

Bistability is only the first step up the bifurcation tree that leads from system behavior with the highest degree of symmetry in space and time — possibly as simple as uniform in both — to behavior that has greater and greater complexity (Eckmann, 1981; Ghil and Childress, 1987; Strogatz, 2018). We outline now one further step up this tree, the one leading from fixed points

to stable periodic solutions, called limit cycles in dynamical systems parlance.

In polar coordinates, this normal form is given by

$$\dot{\rho} = \rho(\mu - \rho^2), \quad \dot{\theta} = v, \tag{7}$$

with $v = -1$ for uniform anticlockwise rotation around the origin. The two equations above are decoupled and the one for the radial variable $\rho = x^2 + y^2$ has exactly the same form as the pitchfork normal form for $x$ in Eq. (5). Note, though, that here $\rho$

is necessarily nonnegative and the mirror symmetry of Fig. 4b is replaced by the rotational symmetry of Fig. 4c.

The version shown in Fig. 4 is the supercritical one, which leads to a smooth increase with $\mu$ in the amplitude of an oscillation generated by the Hopf bifurcation. For plots of the subcritical Hopf bifurcation, please see Ghil and Childress (1987, Figs. 12.8 and 12.9). In particular, in the presence of higher-order terms, as in Fig. 12.9b of Ghil and Childress (1987), a sharp jump from no oscillation to a finite-amplitude one occurs as $\mu$ passes a critical threshold, and one can have a hysteresis cycle between no

oscillation and a large-amplitude oscillation. For instance, it is a matter of some debate whether the Mid-Pleistocene Transition — during which both the amplitude and the dominant periodicity of climatic variability changed — might be associated with a sub- or a supercritical Hopf bifurcation; see Riechers et al. (2022, and references therein).

### 2.1.4   Successive bifurcations and routes to chaos

A further step on the route to chaos for deterministic systems with no explicit time dependence (Eckmann, 1981; Ghil and

Childress, 1987; Strogatz, 2018) involves the transition from a one-dimensional limit cycle to a two-dimensional torus in phase space. In the latter case, the motion on the torus is quasi-periodic – i.e., the coordinates of the point on the torus are of the form $(x, y) = (x(t) = f(\upsilon_1 t, \upsilon_2 t), y(t) = g(\upsilon_1 t, \upsilon_2 t))$, where the functions $f(s, t)$ and $g(s, t)$ are arbitrary and the two angular frequencies $\upsilon_1$ and $\upsilon_2$ are incommensurable, i.e., $\upsilon_1 / \upsilon_2$ is not a rational number.

This kind of motion is typical in celestial mechanics (Arnold et al., 2007; Ghil and Childress, 1987) and, in fact, the pe-

riodicities that are associated with the orbital forcing of the glacial-interglacial cycles are of this type, although one usually refers to them by truncated values — such as those in Table 12.1 of Ghil and Childress (1987) — that could suggest that the ratios between these periodicities, like 41 kyr for the obliquity and 19 kyr for the precessional parameter, do have a common denominator. The latter view is clearly an oversimplification but this is not the place to discuss chaos in the solar system, whether Hamiltonian or, more recently, dissipative.

Quasi-periodic motion already looks much more irregular than purely periodic motion. Thus, for instance, the intervals between lunar or solar eclipses are highly irregular. Still, the 14th century scholar Nicole Oresme was already aware of the kinematic consequences of quasi-periodicity for celestial motions (Grant, 1961). He realized that a periodic and a quasi-periodic motion cannot be distinguished from each other during a finite observation interval. Oresme also knew that the motion of a point on a torus will describe a simple closed loop if the two angular velocities are commensurable, while the point's orbit

will never close, but densely cover the surface of the torus if the two velocities are incommensurable (Arnol'd, 2012), i.e., in a way that is visually indistinguishable from painting the whole torus a uniform color.

From quasi-periodic motion to a deterministically chaotic one, there are several routes (Eckmann, 1981), as already mentioned in Sec. 1.1. Some of these routes to chaos were explored numerically in the climate sciences by Lorenz (1963a, b) and described more didactically in Chapters V and VI of Ghil and Childress (1987) for atmospheric motions. For such routes in

the paleoclimatic context, see Ghil and Childress (1987, Ch. XII), as well as Ghil (1994). We shall not go into greater detail herein, but pass instead to the more recent insights from the theory of dynamical systems subject to time-dependent forcing.

## 2.2 Nonautonomous and random dynamical systems

Realistically, the natural systems that we want to describe in terms of dynamical system theory are nonautonomous, meaning that $\mathbf{f}$ in Eq. (1) above has an explicit time dependence, $\partial \mathbf{f}/\partial t \not\equiv \mathbf{0}$. The Earth system as a whole, as well as all its components, is clearly nonautonomous, being affected by time-dependent forcing, such as quasi-periodic variations in solar insolation due to gravitational perturbations in Earth's orbit (Milankovitch, 1920), along with anthropogenic forcing due to rising greenhouse gas concentrations (Arrhenius, 1896; Houghton et al., 1990; Solomon et al., 2007; IPCC, 2014, 2021).

Moreover, there is typically high-frequency forcing, such as cloud processes or weather variability. In a drastic simplification, this type of forcing is often represented by white noise (Hasselmann, 1976). Including both deterministic and stochastic time dependence requires a description of the dynamics in terms of stochastic differential equations of the form

$$\mathrm{d}\mathbf{X} = \mathbf{F}(\mathbf{X}, t; p)\mathrm{d}t + \sigma(\mathbf{X})\mathrm{d}\boldsymbol{\eta}, \tag{8}$$

where $\mathrm{d}\boldsymbol{\eta}$ denotes the infinitesimal increments of a Wiener process, which are stationary and independently distributed according to a normal distribution with mean $\mu = 0$ and variance $\mathbb{E}(|\mathrm{d}\boldsymbol{\eta}|^2) = \mathrm{d}t$. Often, a further simplification is made in assuming that the noise is additive or state-independent, and thus $\sigma = \mathrm{const.}$ above. The possibly time-dependent, but still deterministic term $\mathbf{F}(\mathbf{X}, t; p)$ is called the drift.

Interest in autonomous dynamical systems and their bifurcations started over two centuries ago and can be traced back to L. Euler and the Bernoullis, while that in nonautonomous and random dynamical systems (NDSs and RDSs) only goes back a few decades. We describe some key differences between the two cases next and justify the need for considering pullback attractors (PBAs) in the latter case.

### 2.2.1 NDSs, RDSs and pullback attraction

For the sake of simplicity, we assume that the physical system under consideration is described by a set of ODEs. In the autonomous case, such a set of ODEs can be formally written as

$$\frac{\mathrm{d}\mathbf{X}}{\mathrm{d}t} = \mathbf{F}(\mathbf{X}), \quad \mathbf{X}(t_0) = \mathbf{X}_0; \tag{9}$$

here $t \in \mathbb{R}$, $\mathbf{X} \in \mathbb{R}^d$, $\mathbf{F} : \mathbb{R}^d \to \mathbb{R}^d$, and $d$ is the number of the system's dependent variables.

For the *nonautonomous* case, the brief presentation here follows Caraballo and Han (2017) and the paradigmatic formulation of the initial-value problem is

$$\frac{\mathrm{d}\mathbf{X}}{dt} = \mathbf{G}(t, \mathbf{x}), \quad \mathbf{X}(t_0) = \mathbf{X}_0. \tag{10}$$

As in Eq. (9), $t \in \mathbb{R}$, $\mathbf{X} \in \mathbb{R}^d$, and $\mathbf{G} : \mathbb{R} \times \mathbb{R}^d \to \mathbb{R}^d$ in Eq. (10), and one still assumes that $\mathbf{G}$ has "nice" properties that guarantee the existence, uniqueness and continuous dependence on initial states and on parameters for the solutions of Eq. (10). Furthermore, Caraballo and Han (2017) show that, provided the vector field $\mathbf{G}(t, \mathbf{X})$ is dissipative, solutions of Eq. (10) exist and satisfy the two other properties globally, i.e., for all $t \in \mathbb{R}$. We call such a global solution $\varphi(t, t_0, \mathbf{X}_0)$.

There are two key distinctions between the autonomous case and the nonautonomous one:

(a) In the autonomous setting, solutions cannot intersect, since there is only one trajectory through a given point $\mathbf{X}_0 \in \mathbb{R}^d$, due to uniqueness. Hence, for $d = 2$, the only possible (forward) attracting sets are fixed points and limit cycles, i.e., chaotic behavior and strange attractors can only occur for $d \geq 3$. The NDS setting is different in these respects, i.e., intersections are possible at two times $t_1$ and $t_2 \neq t_1$, and thus chaos can occur for $d = 2$ and periodic forcing, as is the case, for instance, in the Van der Pol oscillator (e.g., Guckenheimer and Holmes, 1983).

(b) In the autonomous setting, solutions depend only on the time $t - t_0$ elapsed since initial time, while in the NDS setting, they depend separately on the initial time $t_0$ and the current time $t$, at which we observe the system. In the former setting, it suffices to consider forward-in-time attraction, which results in attractors that are fixed, time-independent objects, such as fixed points, limit cycles, tori and strange attractors, In the latter case, we need to define pullback attraction and the PBAs that it leads to.

Before proceeding with a more rigorous justification for and definition of a PBA, here it is, in the simplest possible terms: A pullback attractor is a possibly time-dependent object in a system's phase space that exhibits attraction in the sense of convergence at each time $t$ to a set, called a snapshot, to which the system's initial state at time $s$ tends to as $s$ tends to $-\infty$. This is distinct from the forward attractors that can be defined for autonomous systems started at a fixed time $t_0$.

Given the uniqueness and the continuous dependence of the global solutions to Eq. (10) on initial states and on parameters, it is straightforward to verify that a global solution $\varphi$ of (10) satisfies:

(i) the *initial value property* at $t = t_0$, namely $\varphi(t_0, t_0, \mathbf{X}_0) = \mathbf{X}_0$; and

(ii) the *two-parameter semigroup* evolution property,

$$\varphi(t_2, t_0, \mathbf{X}_0) = \varphi(t_2, t_1, \varphi(t_1, t_0, \mathbf{X}_0)) \quad \text{for} \quad t_0 \leq t_1 \leq t_2,$$

which coresponds to the concatenation of solutions; i.e., in order to go from $t_0$ to $t_2$ one can go first from $t_0$ to $t_1$ and then from $t_1$ to $t_2$.

One can then provide the following definition of a *process*.

**Definition 1**. Let $\mathbb{R}_+^2 = \{(t, t_0) \in \mathbb{R}^2 : t \geq t_0\}$. A process on $\mathbb{R}^d$ is a family of mappings

$$\varphi(t, t_0, \cdot)) : \mathbb{R}^d \to \mathbb{R}^d, \quad (t, t_0) \in \mathbb{R}_+^2,$$

which satisfy

(i) the initial value property $\varphi(t_0, t_0, \mathbf{X}) = \mathbf{X}$ for all $\mathbf{X} \in \mathbb{R}^d$ and any $t_0 \in \mathbb{R}$;

(ii) the two-parameter semigroup property for all $\mathbf{X} \in \mathbb{R}^d$ and both $(t_2, t_1) \in \mathbb{R}_+^2$ and $(t_1, t_0) \in \mathbb{R}_+^2$; and

(iii) the continuity property that the mapping $(t, t_0, \mathbf{X}) \mapsto \varphi(t, t_0, \mathbf{X})$ be continuous on $\mathbb{R}_+^2 \times \mathbb{R}^d$.

An alternative NDS formulation to this process formulation is the so-called *skew-product* formulation, which goes back to the work of G. R. Sell, as reviewed in Sell (1971); see also Kloeden and Yang (2020). A process as defined above is also called a two-parameter semigroup on $\mathbb{R}^d$, in contrast with the one-parameter semigroup of an autonomous dynamical system, since the former depends not just on the initial time $t_0$, as in the latter case, but also on the current time $t$.

This difference matters, in particular, in determining the asymptotic behavior of the solutions. In the autonomous case, a global solution is invariant with respect to translation in time, $\varphi(t, t_0, \mathbf{X}_0) = \varphi(t - t_0, 0, \mathbf{X}_0)$. Hence, the usual forward asymptotic behavior for $t \to +\infty$ and $t_0$ fixed is the same as the behavior for $t$ fixed and $t_0 \to -\infty$. This equivalence may no longer hold when the translation invariance is lost, as it is in the NDS case.

To illustrate the effect of this lost invariance, consider the following simple scalar ODE (Caraballo and Han, 2017, Sec. 3.2.1):

$$\frac{\mathrm{d}x}{\mathrm{d}t} = -ax + b\sin t, \quad x(t_0) = x_0, \quad t \geq t_0, \tag{11}$$

for which analytical computations can be carried out explicitly. Individual solutions do not have a forward limit as $t \to +\infty$ for $t_0$ fixed, but the difference between any two solutions vanishes in this limit. The particular solution

$$\mathcal{A}(t) = \frac{b(a\sin t - \cos t)}{a^2 + 1} \tag{12}$$

provides the long-term information on the behavior of all the solutions of Eq. (11). This result is best captured by recognizing that the *pullback* limit

$$\lim_{t_0 \to -\infty} \varphi(t, t_0, x_0) = \mathcal{A}(t) \quad \text{for all } t \text{ and } x_0 \in \mathbb{R} \tag{13}$$

yields $\mathcal{A}(t)$ as the *PBA* of all the solutions of Eq. (11).

One is thus led to the following rigorous definition of a PBA for a forced dissipative dynamical system subject to a time-dependent forcing, where we have generalized $\mathbb{R}^d$ to a finite-dimensional metric space $\mathcal{X}$ and replaced $t_0$ by $s$, for greater symmetry.

**Definition 2**. A PBA $\mathcal{A}$ is an indexed family of invariant sets $(\mathcal{A}(t))_{t \in \mathbb{R}}$ that depend on time and satisfy the following conditions:

1. For all $t$, $\mathcal{A}(t)$ is a compact subset in $\mathcal{X}$ that is invariant with respect to the two-parameter semi-group $\mathcal{F}(t, s)$,

$$\mathcal{F}(t, s)\mathcal{A}(s) = \mathcal{A}(t) \quad \text{for every } s \leq t; \quad \text{and} \tag{14}$$

2. for all $t$, pullback attraction is reached when

$$\lim_{s \to -\infty} D_{\mathrm{H}}(\mathcal{F}(t, s)\mathcal{B}, \mathcal{A}(t)) = 0 \text{ for all } \mathcal{B} \in \mathcal{C} \tag{15}$$

where $D_{\mathrm{H}}(E, D)$ is the Hausdorff semi-distance between two sets, and $\mathcal{C}$ is a collection of bounded sets in $\mathcal{X}$.

In the physical literature, the invariant sets $\mathcal{A}(t)$ at a given $t \in \mathbb{R}$ have been called *snapshots* (Romeiras et al., 1990) and this terminology has been used also in the recent climate literature on the applications of NDSs, RDSs and PBAs (Ghil et al., 2008; Chekroun et al., 2011; Tél et al., 2020).

The finite-dimensional definition above follows Charó et al. (2021b, Appendix A and references therein). In fact, both deterministic and stochastic versions of forcing have been applied, for instance, by Chekroun et al. (2018) in the study of an infinite-dimensional, delay-differential equation model of the El Niño–Southern Oscillation (ENSO). The deterministic forcing corresponded to the purely periodic, seasonal changes in insolation, while the stochastic component represented the westerly wind bursts appearing in various ENSO models by F.-F. Jin and A. Timmermann (e.g., Timmermann and Jin, 2002); see also Chekroun et al. (2011, Sec. 4.3). This ENSO example, among many others, shows that there is great flexibility in the application of the concepts and methods of nonautonomous dynamical systems (NDS and RDS) theory to climate problems.

### 2.2.2 Simple examples of PBAs and RAs

*A straight-line PBA.* An even simpler example of a PBA than the one of Eqs. (11, 12) above is given by

$$\dot{x} = -\alpha x + \sigma t, \tag{16}$$

with both $\alpha$ and $\sigma$ positive. The example was provided by M. D. Chekroun (pers. commun., 2011). The autonomous part of this ODE, $\dot{x} = -\alpha x$, is dissipative, and all solutions $x(t; x_0) = x(t; x(0) = x_0)$ converge to 0 as $t \to +\infty$. What about the nonautonomous, forced ODE?

Here, the time-dependent forcing $\sigma t$ and the state-dependent dissipation $-\alpha x$ will tend to balance. But, again, as in the example of Sec. 2.2.1, there is no forward limit as $t \to +\infty$, and one has to use the pullback limit, i.e., replace $x(t; x_0)$ by $x(s, t; x_0) = x(t; x(s) = x_0)$ and let $s \to -\infty$. Doing so yields the snapshots

$$\mathcal{A}(t) = \frac{\sigma}{\alpha}\left(t - \frac{1}{\alpha}\right). \tag{17}$$

These snapshots are, in the extremely simple case at hand, just the points along the straight line illustrated in Fig. 5, which is the graph of the PBA $(\mathcal{A}(t))_{t \in \Re}$.

*A PBA with periodic forcing.* To further improve the reader's intuition for PBAs, we provide a second illustrative example here. It was worked out in detail by Riechers et al. (2022).

A system defined in polar coordinates by

$$\dot{\rho} = \alpha(\mu - \rho), \quad \dot{\phi} = \upsilon, \quad \text{with } \rho, \mu \in \mathbb{R}^+ \text{ and } \phi \in \mathbb{R}/2\pi, \tag{18}$$

can easily be seen to exhibit a limit cycle in the $(x, y)$-plane with $(x = \rho \cos\phi, y = \rho \sin\phi)$. An initial deviation of $\rho$ from $\mu$ will decay exponentially and the system converges to an oscillation of radius $\mu$ with the angular velocity $\upsilon$. Here, we transform this autonomous dynamical system into a nonautonomous one by modulating the target radius $\mu$ with a sinusoidal forcing

$$\mu \to \mu(t) = \mu + \beta \sin(\nu t), \tag{19}$$

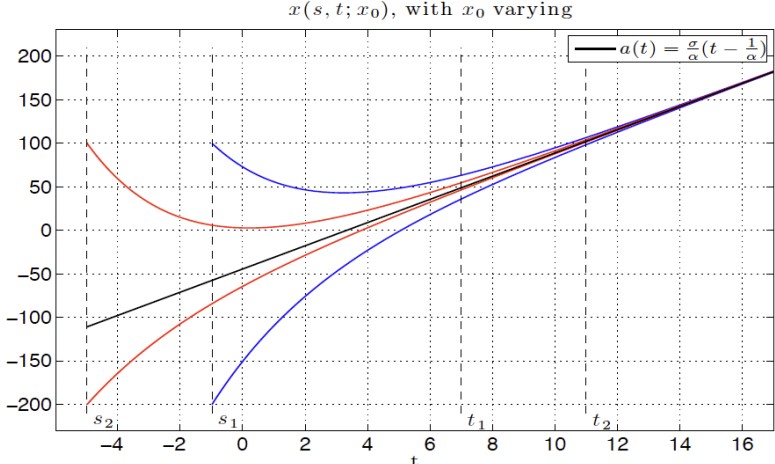

**Figure 5.** The graph of the PBA for the simple NDS example governed by Eq. (16) and given by the indexed family $\left(\frac{\sigma}{\alpha}\left(t - \frac{1}{\alpha}\right)\right)_{t\in\Re}$, along with several trajectories that converge to it from times $s_2 < s_1 < t_1 < t_2$. Here $t_1 < t_2$ are the increasing times at which we are observing the system, while $s_2 < s_1$ are the decreasing times to which we have to pull back in order to get the convergence. Figure provided by Mickaël D. Chekroun.

where the modulation is moderate, so as to guarantee that $\mu + \beta\sin(\nu t) > 0$ for all $t$.

Since the dynamics of the phase $\phi$ and of the radius $\rho$ are decoupled, the corresponding equations can be solved and analyzed separately. While the temporal development of the phase is trivial, the pullback invariant attracting set of the radius for the initial condition $\rho(s) = \rho_0$ is given by

$$\mathcal{A}^{(\rho)}(t;\rho_0) = \lim_{s\to-\infty} \rho(t,s;\rho_0) = \alpha\beta\sin(\nu t + \vartheta) + \mu, \quad \text{with} \tag{20a}$$

$$\vartheta = \arctan(-\nu/\alpha), \tag{20b}$$

as shown in Riechers et al. (2022, Appendix B). Note that, in the limit $s \to -\infty$, the dependence on the initial value $\rho_0$ vanishes and the attracting set $\mathcal{A}_t^{(\rho)}$ performs an oscillation of the same frequency as the forcing. It lags the phase of the time-dependent fixed point by the constant $\vartheta$, while its amplitude is amplified by the factor $\alpha$. Since $\rho$ is restricted to positive values, this solution requires $\alpha\beta < \mu$.

The PBA with respect to the coordinate $\rho$ is comprised of the family of all the sets $\mathcal{A}_t^{(\rho)}$ as defined in (20) and thus reads

$$\mathcal{A}^{(\rho)} = \{\alpha\beta\sin(\nu t + \vartheta) + \mu\}_{t\in\mathbb{R}}. \tag{21}$$

Since the pullback limit for the phase $\phi$ does not exist, no constraints on it other than $\phi \in [0, 2\pi)$ are imposed by the dynamics. Hence, for the system (18) comprised of radius and phase, we find that

$$\lim_{t_0\to-\infty} d_{\mathrm{H}}\left( \big(\rho(t;t_0,\rho_0), \phi(t;t_0,\phi_0)\big), \underbrace{\{\big(\alpha\beta\sin(\nu t + \vartheta) + \mu, \phi\big) : \phi \in [0, 2\pi)\}}_{\mathcal{A}_t} \right) = 0, \tag{22}$$

where $d_{\mathrm{H}}$ denotes the Hausdorff semi-distance. The pullback attracting sets $\mathcal{A}_t$ at time $t$ are circles in the $(x,y)$-plane with oscillating radius, and the system's PBA is given by the family of these circles

$$\mathcal{A} = \{(\alpha\beta\sin(\nu t + \vartheta) + \mu, \phi) : \phi \in [0, 2\pi)\}_{t \in \mathbb{R}}. \tag{23}$$

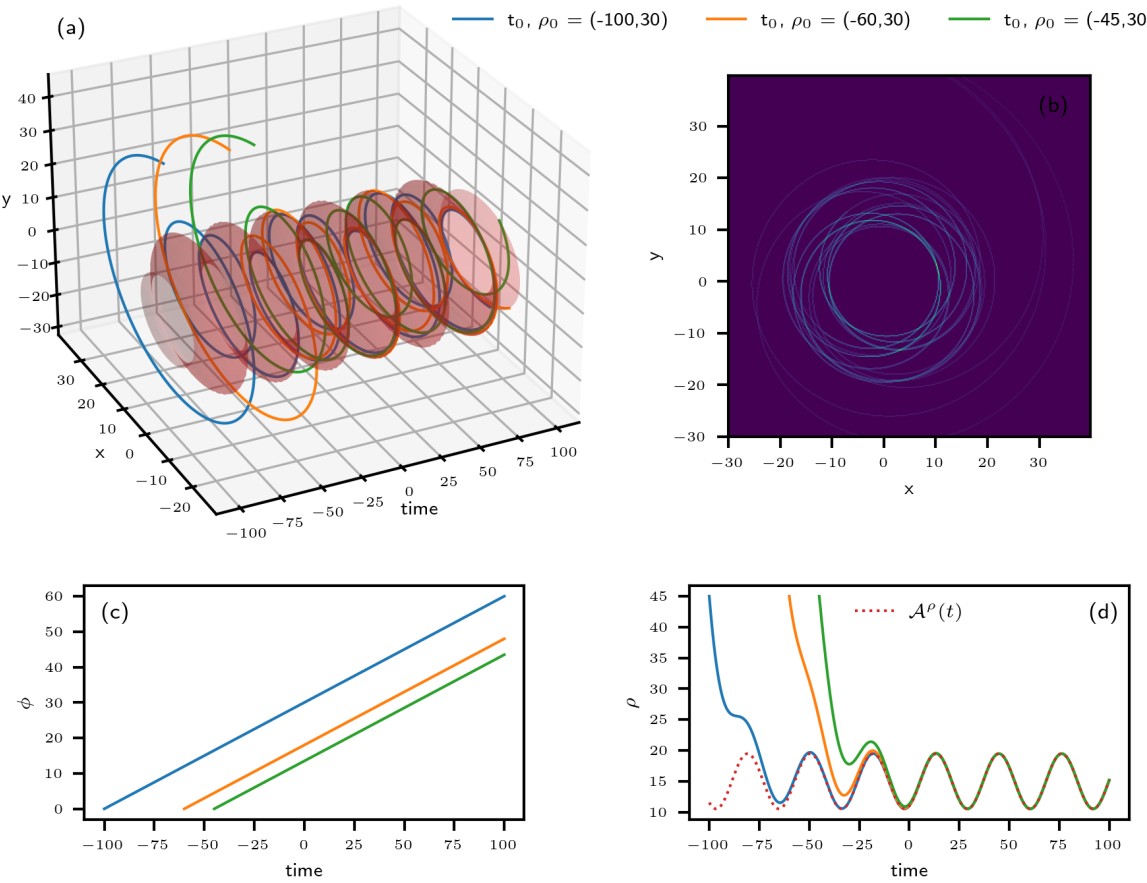

**Figure 6.** Trajectories and PBA of the system defined by Eqs. (18)–(20). (a) Trajectories $(\rho(t), \phi(t))$ of the system starting from different times in the past in the 3-D space spanned by the two cartesian coordinates $(x, y)$ and time $t$; the system's PBA lies on the red-shaded surface. (b) Heat map of the three trajectories' projection onto the $(x, y)$-plane. A video of the heat map filling up, as more and more trajectories with different initial conditions are added, is provided in the Supplementary Material to Riechers et al. (2022). (c) Temporal evolution of the phase. (d) Temporal evolution of the radius (solid colors) together with its PBA (dashed red). From Riechers et al. (2022) with thanks to the coauthors Keno Riechers, Takashita Mitsui and Niklas Boers.

Figure 6 shows trajectories of the system starting from different points in the past. In panel (a) the trajectories are depicted in the 3-D space spanned by the two cartesian coordinates $(x, y)$ and the time $t$, where the usual transformation from polar to

cartesian coordinates was applied. The shaded surface in this panel represents the PBA of the system. Panel (b) shows a heat map (Wilkinson and Friendly, 2009) that approximates a portion of the PBA's invariant measure projected onto the $(x, y)$-plane. For a clean definition of such a measure in NDSs and RDSs, there are several references (e.g., Ghil et al., 2008; Chekroun et al., 2011; Caraballo and Han, 2017; Kloeden and Yang, 2020). Essentially, the heat map here counts the number of times that the trajectories in panel (a) cross small pixels in the $(x, y)$-plane.

Note that the structure of the system's trajectories depends on the ratio $v/\nu$ and three different cases must be distinguished. If the radius is modulated with the same frequency as the oscillation itself, i.e. $v = \nu$, after one period the system practically repeats its orbit. More precisely, the radius of the oscillation does differ from one "roundtrip" to the next, but this difference tends to zero as $\rho(t)$ asymptotically approaches $\mathcal{A}_t^{(\rho)}$. If $v$ and $\nu$ are rationally related, $mv = n\nu$ with $n, m \in \mathbb{N}$, then the same quasi-repetition of the orbit occurs after $n$ periods of the radial modulation and $m$ periods of the system's oscillation. Such a trajectory will appear as an $n$-fold quasi-closed loop. Finally, if $v/\nu \notin \mathbb{Z}$, then the trajectory does not repeat itself but instead covers densely the annular disc $\mathcal{D} = \{(\rho, \phi) : \rho \in [\mu - \alpha\beta, \mu + \alpha\beta]$ and $\phi \in [0, 2\pi)\}$. The trivial evolution of the phase is depicted in panel (c), while the trajectories of $\rho(t)$ and their convergence to $\mathcal{A}_t^{(\rho)}$ are shown in panel (d).

### 2.2.3 Random attractor

Let us return now to the more general, nonlinear and stochastic case of Eq. (8) that includes not only deterministic time dependence $\mathbf{F}(\mathbf{X}, t)$, but also random forcing,

$$d\mathbf{X} = \mathbf{F}(\mathbf{X}, t)dt + \mathbf{G}(\mathbf{X})d\eta; \tag{24}$$

here $\eta = \eta(t, \omega)$ represents a Wiener process, which is taken to be scalar; its independent increments $d\eta$ are commonly referred to as "white noise" and $\omega$ labels the particular realization of this random process. More generally, one can also deal with a vector Wiener process, as in Eq. (8). See, for instance, Wax (1954) for early references on these matters.

When $\mathbf{G} = $ const. the noise is additive, while for $\partial\mathbf{G}/\partial\mathbf{X} \neq \mathbf{0}$ we speak of multiplicative noise. Intuitively, the distinction between $dt$ and $d\eta$ in the stochastic differential equation (24) is necessary since, roughly speaking and following the Einstein (1905) paper on Brownian motion, it is the variance of a Wiener process that is proportional to time and thus $d\eta \propto (dt)^{1/2}$. In Eq. (24), we also dropped for the sake of simplicity the dependence on a parameter $p$ that we had introduced, for the sake of generality, in Eq. (8).

The noise processes may include "weather" and volcanic eruptions when $\mathbf{X}(t)$ is "climate," thus generalizing the linear model of Hasselmann (1976), or cloud processes when we are dealing with the weather itself: one person's signal is another person's noise, as the saying goes. In the case of random forcing, the concepts introduced by the simple example of Eq. (8) above can be illustrated by the *random attractor* $\mathcal{A}(\omega)$ in Fig. 7.

Chekroun et al. (2011) studied a specific case of such a random attractor for the paradigmatic, climate-related Lorenz (1963a) convection model. The authors introduced multiplicative noise into each of the ODEs of the original, deterministically chaotic

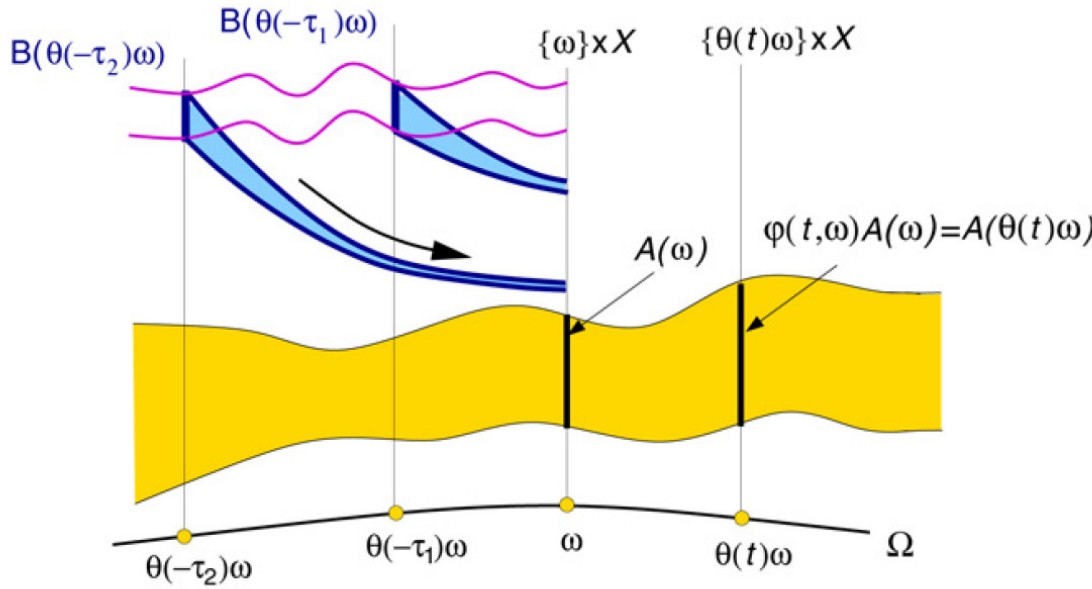

**Pullback attraction to A(ω)**

**Figure 7.** Schematic diagram of a random attractor $\mathscr{A}(\omega)$ and of the pullback attraction to it; here $\omega$ labels the particular realization of the random process $\theta(t)\omega$ that drives the system. We illustrate the evolution in time $t$ of the random process $\theta(t)\omega$ (light solid black line at the bottom); the random attractor $\mathcal{A}(\omega)$ itself (yellow band in the middle), with the *snapshots* $\mathcal{A}_0(\omega) = \mathcal{A}(\omega; t = 0)$) and $\mathcal{A}(\omega; t)$ (the two vertical sections, heavy solid); and the flow of an arbitrary compact set $\mathcal{B}$ from "pullback times" $s = -\tau_2$ and $s = -\tau_1$ onto the attractor (heavy blue arrows). See Appendix A in Ghil et al. (2008) for the requisite properties of the random process $\theta(t)\omega$ that drives the RDS.

After Ghil et al. (2008) with permission from Elsevier.

system, as shown below:

$$\mathrm{d}X = P_r(Y - X)\mathrm{d}t + \sigma X \mathrm{d}\eta, \tag{25a}$$

$$\mathrm{d}Y = (rX - Y - XZ)\mathrm{d}t + \sigma Y \mathrm{d}\eta, \tag{25b}$$

$$\mathrm{d}Z = (-bZ + XY)\mathrm{d}t + \sigma Z \mathrm{d}\eta; \tag{25c}$$

here $r = 28$, $P_r = 10$, $b = 8/3$ are the standard parameter values for chaotic behavior in the absence of noise, and $\sigma$ is a

constant variance of the Wiener process that is not necessarily small. The well-known strange attractor of the deterministic case is replaced by the Lorenz model's random attractor, dubbed LORA by the authors.

Four *snapshots* $\mathcal{A}_t(\omega)$ of LORA are plotted in Fig. 8 here and a video of its evolution in time $\mathcal{A}(\omega) = \{\mathcal{A}_t(\omega)\}_{t\in\mathbb{R}}$ is available as Supplementary Material in Chekroun et al. (2011) at https://doi.org/10.1016/j.physd.2011.06.005.

What is actually plotted, in both the figure reproduced here and in the video, is the approximation of the time-dependent invariant measure $\nu_t(\omega)$ supported on the attractor. A full definition of the sample measures of random attractors would occupy too much space in an already rather long review paper; please see Chekroun et al. (2011, Appendix A) and Charó et al. (2022b, Appendix C), along with the references therein.

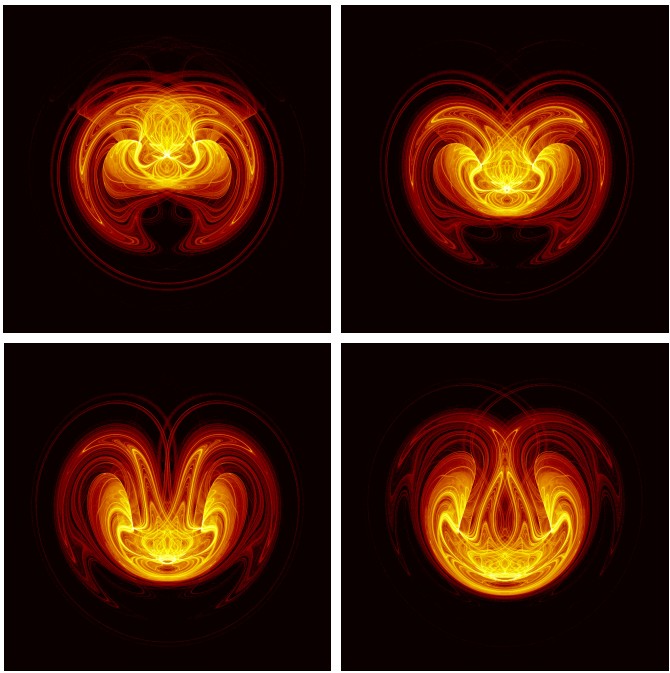

**Figure 8.** Heatmaps of the time-dependent invariant measure $\nu_t(\omega)$ supported on four snapshots $\mathcal{A}_t(\omega)$ of LORA. The values of the parameters $r, s$ and $b$ are the classical ones, while the variance of the noise is $\sigma = 0.5$. The color bar, shown in Chekroun et al. (2011, Fig. 2) for a single snapshot, is on a log-scale and it quantifies the probability of landing in a particular region of phase space; shown is a projection of the 3-D phase space $(X, Y, Z)$ onto the $(X, Z)$-plane. Note the complex, interlaced filament structures between highly populated regions (in yellow) and moderately populated ones (in red); the less populated a small patch, the darker its color. The time interval between the snapshots shown, left to right and top to bottom, is of $\Delta t = 0.0875$ in the nondimensional time units of the deterministic Lorenz (1963a) model. Reproduced from Chekroun et al. (2011, Fig. 3) with permission from Elsevier.

The striking effects of the noise on the nonlinear dynamics that are visible in Fig. 8 here and in the video of Chekroun et al. (2011) motivated much of the work reviewed in Sec. 3 below, starting with LORA's topological study by Charó et al. (2021b). The latter study gathered further insights into the abrupt changes of the snapshots' topology at critical points in time, changes that suggested the possibility of random processes giving rise to qualitative jumps in climate variability.

### 2.2.4 Abrupt transitions in nonautonomous systems

Ashwin et al. (2012) have proposed three classes of abrupt transitions in systems that can be described by Eq. (24): (i) bifurcation-induced transitions, (ii) noise-induced transitions, and (iii) rate-induced transitions. An example of the first class has already been given in Sec. 2.1 and Fig. 4a above.

For an example of the second class, assume that the control parameter $p$ remains constant in the drift term of Eq. (8), which is taken again to correspond to a double-well potential, as in Eq. (4). Noise-induced transitions occur when the noise amplitude is sufficiently high for the system to switch occasionally, and unpredictably, from one potential well to the other. Moreover, when $p$ varies so as to push the system toward a bifurcation point, the noise will cause it to transition before – and, in certain cases, long before — the critical parameter value $p^*$ of the corresponding deterministic system is reached.

Finally, the third class of rate-induced transitions arises when there is no strong separation between the system's intrinsic time scales and those at which the control parameter changes. So far, we implicitly assumed that, for each change in $p$, the system has sufficient time to adapt to the new equilibrium position; this type of slow change in $p$ is sometimes called quasi-adiabatic. If this is not the case, the fixed point attracting the system may change its position so quickly that the system cannot follow and eventually loses track of the basin of attraction in which it started and falls into the other one (Ashwin et al., 2012).

Ashwin et al. (2012) have called these transitions *tippings* and refer to the three types described above as B-tipping, N-tipping and R-tipping. Thus, aside from the rhetorically striking character of tipping points, tippings are the mathematically well defined generalization of the bifurcations treated in the autonomous dynamical systems of Sec. 2.1 to the nonautonomous and random setting addressed herein. In fact, the first two types, B- and N-tipping, are not totally novel inasmuch as they only add deeper insight to what happens when a parameter $p$ changes at a slow but finite, rather than infinitely slow rate. The biggest surprises occur for R-tipping (Wieczorek et al., 2011; Feudel et al., 2018; Ghil, 2019; Pierini and Ghil, 2021), but we will not deal explicitly with this form of tipping herein.

We illustrate in Fig. 9 the N-tipping of a system governed by $dx = U'(x; p)dt + \sigma dW$, with $U$ as in Eq. (4) and $dW$ as in Eq. (24), but $\sigma = $ const. For example, simple EBMs (Ghil and Lucarini, 2020) exhibit a double-fold bifurcation of this kind, as described already in Sec. 2.1 above. The upper stable branch corresponds in this case to the current climate state, while the lower one corresponds to the Snowball Earth state (Held and Suarez, 1974b; Ghil, 1976b; Ghil and Lucarini, 2020).

To simulate the system's trajectory, the control parameter $p$ is varied slowly from $+1$ to $-1$ and back to $+1$, causing the system to transition first from the upper stable branch to the lower one, and then, at a considerably higher $p$-value, back to the upper stable branch. Note that due to the noise driving the system, transitions typically occur earlier than expected from the corresponding deterministic dynamics governed by Eq. (4).

Note also that in the generalization from autonomous bifurcations to nonautonomous tippings, the phrase "tipping point" — aside from its threatening implication — is somewhat misleading: a bifurcation point is a point in phase-parameter space, like $(\pm x^*, \pm p^*)$ for the double well of Eq. (4) and Fig. 4a. While the meaning attached to it by Gladwell (2000) in general and by Lenton et al. (2008) in the climate sciences refers only to the value of the forcing, like $\pm p^*$ in the case above.

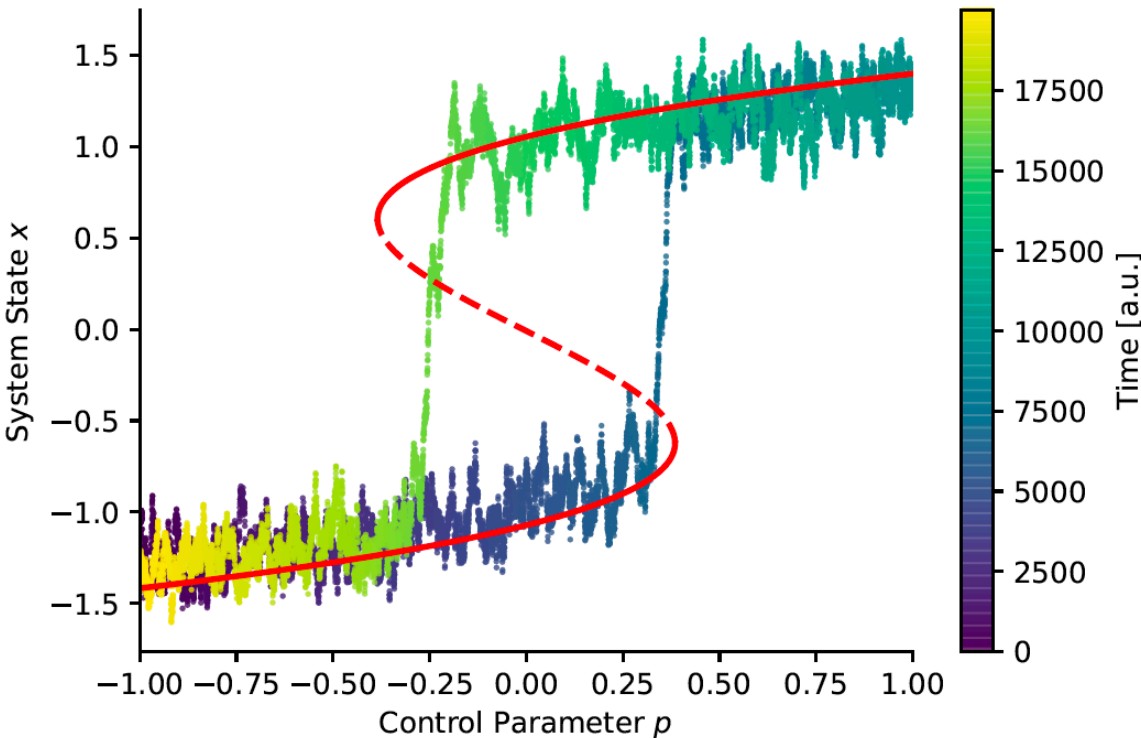

**Figure 9.** Sketch of a double-fold bifurcation and how it leads to abrupt transitions and hysteresis in the temporal evolution of a system in a double-well potential with slowly changing parameter, $p = p(\epsilon t)$, where $\epsilon \ll 1$, driven by additive white noise. The stable branch of fixed points is indicated by solid red, the unstable one by dashed red. Compare with Fig. 4a. After Boers et al. (2022).

## 3 Topological Structure of Flows in Phase Space and in Physical Space

At the end of Sec. 1.2, we mentioned that knot theory provided a first approach towards unveiling the topological structure of a flow in a 3-D phase space. In this case, the term "flow" does not refer to a fluid flow in physical space, but to a family of solution curves of ODEs or other evolution equations (Arnol'd, 2012; Coddington and Levinson, 1955; Guckenheimer and Holmes, 1983). Of course, a flow in phase space may — as we will see later in Sec. 3.2 — refer to a particle in the Lagrangian description of a fluid flow in physical space. There is a strong link between the two situations, but the keywords refer to different motivations and objectives.

Clarifying the difference between these two kinds of flow, in physical space and in phase space, is relevant here because, in the community involved in the work been reviewed here, the phrase "topological chaos" is used when studying how fluid-particle trajectories are entangled in physical space during a mixing experiment. A noteworthy example is the motion induced

by spatially periodic obstacles in a two-dimensional flow in order to form nontrivial braids (Gouillart et al., 2006; Thiffeault and Finn, 2006b), as shown in Fig. 10. Such motion generates exponential stretching of material lines, and hence efficient mixing.

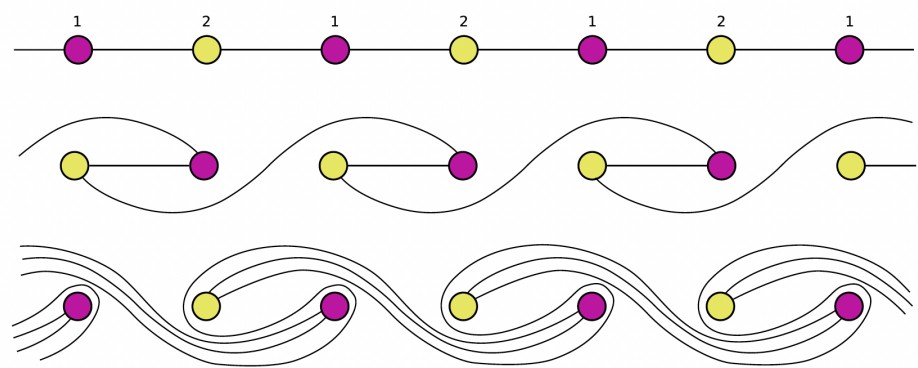

**Figure 10.** Topological chaos emerges in stirring or mixing experiments. Here we see stylized streamlines induced by pairs of rods on a periodic lattice and we see how these streamlines are stretched in physical space. From Thiffeault and Finn (2006a), under CC-BY license.

On the other hand, "topology of chaos" or "chaos topology," for short, considers the problem of how multi-dimensional point clouds or trajectories are topologically structured in phase space. Such a study in phase space is not equivalent to the type of study illustrated in Fig. 10. Working with the topology of real fluid-flow trajectories in physical space requires working in no more than three dimensions, for example. The topological structure we will always be referring to in the present work is defined in phase space, even when studying how such a topological structure is related to the motion of fluid particles in physical space. In 3-D phase space, deterministic flows can be characterized by topological invariants, and, therefore, in terms of knots.

Mathematically, a knot is an embedding of a circle in 3-D Euclidean space $\mathbb{R}^3$. We can imagine a knot as a thin tangled rope in 3-D space whose ends are glued together (Prasolov and Sossinsky, 1997). Two mathematical knots are equivalent if one can be transformed into the other via a deformation of $\mathbb{R}^3$ into itself, known as an ambient isotopy; these transformations correspond to manipulations of a knotted string that do not involve cutting it or passing it through itself. The knot approach — i.e., extracting the knot content of hyperbolic attractors — is based on a geometrical construction that was named template or knot-holder. The first way of applying this approach consisted in computing certain knot invariants — such as linking numbers or Conway polynomials — by starting from a set of trajectories (Gilmore, 1998; Gilmore and Lefranc, 2003; Natiello et al., 2007; Letellier and Gilmore, 2013).

There are in fact three steps in this knot-theoretical approach, and the aim of each one is carried out in a particular way:

(i) approximate the neighboring UPO around which the flow is evolving with an orbit or closed curve;

(ii) find a topological representation for the orbit structure; and

(iii) obtain an algebraic description for the topological representation.

The first step is rooted in Henri Poincaré's observation that one can always choose a model's periodic solution as a first approximation of an aperiodic one (Poincaré, 1892, 1893, 1899). To achieve the first step, one thus applies a close returns method (Mindlin and Gilmore, 1992; Boyd et al., 1994). If the trajectories being studied have been obtained from a data-driven method rather than a model simulation — using, for instance, time-delay embedding (Takens, 1981) — this step requires long, well sampled time series that are noise free for orbits to be reconstructed accurately.

Knot theory comes in the procedure's second step and computing the identified knot invariants closes the procedure. Another possibility, instead of using knots, is resorting to braids, as discussed by Natiello et al. (2007). A braid is a collection of strands crossing over or under each other. The braid approach is based on results due to Thurston on the classification of two-dimensional diffeomorphisms and on the braid content of a given diffeomorphism (Fathi, 1979). The spirit of the procedure is the same, because when connecting the ends of a braid, one ends up with a knot. In the Letellier and Gilmore (2013) Festschrift for Robert Gilmore's 70th birthday, Mario Natiello's Chapter 7 is titled "A braided view of a knotty story." The reason is that knots dissolve into trivial objects in dimensions higher than three …

In "How topology came to chaos," Gilmore (2013b, p. 175) explains that metric and dynamical invariants do not provide a way to distinguish among the different types of chaotic attractors and that a tool of a different nature was needed to create a dictionary of processes and mechanisms underlying a chaotic system. While Gilmore, Lefranc and co-workers were "mulling over implementing a program [based on building tables of linking numbers and relative rotation rates between trajectories, a] better solution became available." Joan Birman and Robert Williams had shown that the dissipative nature of a flow in phase space allows projecting the points along the direction of the stable manifold by identifying all the points with the same future.

"Suppose [continues Gilmore, that] we have a dissipative [chaotic] flow in three dimensions: There is one positive Lyapunov exponent $\lambda_1 > 0$ [for the unstable direction,] one negative Lyapunov exponent $\lambda_3 < 0$ [for the stable direction], and one zero exponent $\lambda_2 = 0$ 'along the direction of the flow'. The dissipative nature of the flow requires $\lambda_1 + \lambda_2 + \lambda_3 = 0$. Then it is possible to project points in the phase space 'down' along the direction of the stable manifold. This is done by identifying all the points with the same future:

$$x \simeq y \text{ if } \lim_{t \to +\infty} |x(t) - y(t)| = 0,$$

[where] $x(t)$ is the future in phase space of the point $x = x(0)$ under the flow. This Birman-Williams identification effectively projects the [3-D] flow down to a two-dimensional set that is a manifold almost everywhere," except at the points where the flow splits into branches heading towards distinct parts of phase space, or at the points where two branches are squeezed together. These mathematical structures were called branched manifolds.

A branched manifold, in the strict sense of the two words that make up the term, can in fact be defined mathematically without reference either to a flow or to the Birman-Williams projection mentioned above. Following Kinsey (1993, p. 64), an $n$-dimensional manifold is a topological space such that every point has a neighborhood topologically equivalent to an $n$-dimensional open disc with center $\mathbf{x}$ and radius $r$. Such a manifold is said to be Hausdorff if and only if any two distinct

points have disjoint neighborhoods. The second condition is not satisfied precisely at the junction between branches, i.e., at the locations that describe stretching and squeezing of a flow in phase space.

A branched manifold is, therefore, a manifold that is not required to fulfill the Hausdorff property. We prefer this more general definition, instead of the one related to the Birman-Williams projection, for several reasons, including the possibility of extending the concept of branched manifold to the structure of instantaneous snapshots of random attractors, as we shall see in Sec. 3.4. This mathematical definition of a branched manifold will also let us extend the procedure to cases in which the hypotheses of the Birman-Williams theorem — in which the dynamical system must be hyperbolic, 3-D, and dissipative — are not valid. In most geoscientific applications, for instance, uniform hyperbolicity does not apply.

As the topological structure of a branched manifold is closely related to the stretching and squeezing mechanisms that constitute the fingerprint of a certain chaotic attractor, its properties can be used to distinguish among different attractors. This is how one can justify the two-way correspondence between topology and dynamics. This correspondence remains valid in the case of four-dimensional semi-conservative systems (Charó et al., 2019, 2021a), for which the hypotheses of the Birman-Williams theorem do not hold.

The terms "branched manifold" and "template" have often been used interchangeably. We do not consider them as synonyms, for technical reasons that will be important in the development of the concept of templex in Sec. 3.3. A branched manifold is just a particular type of manifold that can be reconstructed from a set of points in $\mathbb{R}^n$, by approximating subsets of points by disks of local dimension $d \leq n$. Now, to describe branched manifolds immersed in $n \geq 4$, we still need a different tool. This tool is in fact provided by homology theory.

## 3.1 Branched Manifold Analysis through Homologies

Homologies provide an algebraization of topology by building compressed representations of a certain object through cell complexes and by computing essential signatures of the object's shape through homology groups that do not depend on the the particular representation used to compute them. Homology groups enable the analysis of $n$-dimensional manifolds or point clouds, with $n$ as high as desired. This procedure can handle time-delay embeddings produced with shorter and reasonably noisy time series, since the method no longer relies on orbit reconstruction in phase space. In Natiello's terms (Letellier and Gilmore, 2013, Ch. 7), homologies are knotless and orbit-less, and the topological program can be extended to deal with higher dimensional systems and with real, noisy data.

Other approaches that characterize aspects of dynamical chaos in arbitrary dimensions (e.g., Lefranc, 2006) are somewhat similar to cell complexes. These approaches so far only address estimating the entropy of the flow, which is still an important issue in and of itself.

To illustrate how homologies work, let us take as an example a point cloud obtained by integration of the deterministic (Lorenz, 1963a) model. Here too the methodology has three steps, but they differ in their tasks and their objectives:

(a) approximate the points as lying on a branched manifold;

(b) find a topological approximation of the branched manifold; and

(c) obtain an algebraic description for the topological structure.

Essentially, the passage through the closed orbits is replaced by passing through the branched manifold.

A branched manifold is a generalization of a differentiable manifold that may have singularities of a very restricted type, which correspond to the branching, and admits a well-defined tangent space at each point. In other words, such a manifold has the property that each point has a neighborhood that is homeomorphic to either a full 2-ball or a half 2-ball, and which is locally homeomorphic to Euclidean space or locally metrizable, but not globally so, because of the branching (Williams, 1974). A typical branching line is the one that joins the "pair of surfaces which appear to merge in the lower portion of Fig. 3."

As points in our cloud are assumed to lie on a branched manifold, we can classify the points into subsets that constitute a good local approximation of a $d$-disk, where $d$ is the local dimension of the branched manifold and $n$ is the dimension of phase space ($d \leq n$). In the case of the Lorenz attractor, $d = 2$ and $n = 3$. The topological representation is obtained if we convert each subset of points into an individual cell of a cell complex. This complex is sort of a skeleton of the object of interest, namely the Lorenz (1963a) attractor in the case at hand.

Here we use polygons for the cells that pave the attractor's branched manifold. These cells must be correctly glued to each other in order to retain the topological features of the original point cloud. Once the cell complex is constructed, homologies can be computed to yield an algebraic description of the approximating structure. In this review paper, we will not go into the mathematical definitions and theorems required to fully and correctly understand cell complexes and homology theory, but only give a taste of the theoretical framework via challenging applications. The reader is referred to Kinsey (1993) for the full mathematics at a comfortable level and to Sciamarella (2019) for a more detailed explanation of the geoscientific applications.

The keypoint here is that the homology groups represent essential information about the branched manifold, while being independent of the number of cells used to construct the complex (Poincaré, 1895; Siersma, 2012). The topological structure describing the manifold can thus be identified, higher dimensions can be handled, and relatively short and noisy data can be sufficient for this purpose, too.

When Michael Ghil visited the University of Buenos Aires in Fall 2018 and got acquainted with this methodology, whose first results were published two decades ago (Sciamarella and Mindlin, 1999, 2001), he suggested one should give it a name that identifies and distinguishes it from other methods that had become popular in the meantime in topological data analysis, in particular that of persistent homologies (PH: Zomorodian and Carlsson, 2004; Edelsbrunner et al., 2008). The PH methodology has been enormously successful in problems of shape recognition and classification from large but incomplete data sets.

In dynamic problems, and especially in chaotic dynamics, the PH approach has to contend with the difficulty of finding robust criteria for the degree to which a cell complex does represent a manifold that underlies a point cloud (Carlsson and Zomorodian, 2007). Instead of insisting on the improved approximation of such a manifold, PH chooses to display and evaluate the properties of a sequence of cell complexes constructed with a cell creation rule, called a *filtration*, which depends on a filtration parameter, such as the size of the balls used to approximate the original space around each point of the point cloud. The problem with filtrations is that it is perfectly possible that none of the complexes created by a dynamics-independent rule correctly approximates the branched manifold whose topology is to be described.

For this reason, the Buenos Aires group chose to establish special rules for the construction of a complex, namely rules that do take into account that the objective of the reconstruction is not just any arbitrary shape, but a branched manifold in phase space. Michael Ghil's suggestion led to using Branched Manifold Analysis through Homologies (BraMAH) for this method, a name that says it all and simultaneously recalls the Hindu god of creation and knowledge, which seems very auspicious. The precursors of this technique are four researchers of the Nonlinear Systems Laboratory of the Mathematics Institute at the

University of Warwick, who extracted Betti numbers from time series (Muldoon et al., 1993). Betti numbers define the rank of the homology groups, and they can be seen as the number of "holes" in a point cloud. This method served as a guide to construct a cell complex from a point cloud, using singular value decomposition.

We review here briefly the improvements that Sciamarella and Mindlin (1999, 2001) brought to the Warwick approach. The information that was obtained as output by Muldoon et al. (1993) is useful but incomplete if one wishes to identify a branched

manifold. As observed in the concluding remarks of the latter paper, the examples used therein involve boundaryless manifolds traversed by a dense orbit, but they suggest potential applications to a wider class of objects including branched manifolds. In order to identify a branched manifold from a point cloud through homologies, it is important to realize that there is much more information contained in a cell complex than just the Betti numbers, and that much of this information is relevant to describing the underlying topology.

Sciamarella and Mindlin (1999) were able to show that the branched manifold could be reconstructed with all its features, including torsions and branch locations, from a noisy dataset. The example used was a time series associated with a voice signal, that of a Spanish speaker articulating the word "casa." The topological analysis was carried out on the first vowel, showing that a 3-D time-delay embedding of the acoustic pressure yielded a point cloud with an organization that is typical of a branched manifold. The authors used this dataset to show that the BraMAH method could be applied to reconstruction

from a noisy time series, where identifying unstable periodic orbits would have been very difficult or even impossible. They succeeded in characterizing the topology of this dataset, but also in showing that their approach and its underlying principles had been fruitful.

In their follow-up paper, Sciamarella and Mindlin (2001) described the algorithm in detail, coded in Wolfram Mathematica, and presented an example of a four-dimensional dynamical systems having chaotic solutions of the Shilnikov type. The flow

generated by the set of ODEs considered threin was such that any 3-D projection contained self-intersections, stressing the truly four-dimensional nature of the dataset. Sciamarella and Mindlin (1999, 2001) thus showed that their approach could overcome the two main obstacles in the topological analysis of dynamical systems, namely the limitations of dimensionality imposed by the knot-theoretical approach and the noise.

BraMAH can also detect the presence of a Klein bottle in the data, like the one discovered by Mindlin and Solari (1997).

Recall that a Klein bottle is a a one-sided surface that is formed by passing the narrow end of a tapered tube through the side of the tube and flaring this end out to join the other end. Immersed in three dimensions — as usually shown in the drawings we are used to — a Klein bottle presents self-intersections, and this is why it is a paradigmatic example of a structure that is inherently four-dimensional. In phase space, self-intersections violate uniqueness, and this is why projections may be not only inconvenien but also misleading. Returning to the Muldoon et al. (1993) algorithm, the Betti numbers alone that it computes

do not distinguish a Klein bottle from a Möbius strip. The moral is that the topological description of nonlinear dynamical systems in phase space should not only count the holes — as done today by many available topological toolkits — but should be carried out more fully, as in BraMAH. The method is illustrated in Fig. 11, where it is is applied to the strange attractor of the deterministic Lorenz (1963a) model, according to Charó et al. (2022a) and Charó et al. (2022b).

The topological-analysis program has been applied to many fields of science: voice production (Sciamarella and Mindlin, 1999), ocean color (Tufillaro, 2013), biological motor patterns (Mindlin, 2013), financial economics (Gilmore, 2013a), nano-oscillators (Gilmore and Gilmore, 2013), and so on. What is the purpose? To quote Robert Gilmore: "Topological methods can be used to determine whether or not two dynamical systems are equivalent; in particular, they can determine whether a model developed from time-series data is an accurate representation of a physical system. Conversely, it can be used to provide a model for the dynamical mechanisms that generate chaotic data." The topological program can hence be harnessed for multiple purposes, including but not restricted to

(i) validate or refute models – simulations vs. observations;

(ii) comparing models – time series generated by different models;

(iii) comparing datasets – e.g., in situ versus satellite data.

(iv) characterizing and labeling chaotic behaviors – towards a systematic classification; and

(v) classifying sets of time series according to their main dynamical traits – e.g., in Lagrangian flow analysis.

## 3.2  Lagrangian Coherence in Fluid Flows

In fluid mechanics, two viewpoints are possible. In the Eulerian viewpoint, fluid motion is observed at specific locations in space, as time passes. In the Lagrangian viewpoint, instead, the observer follows individual fluid particles as they move through the fluid domain. The Eulerian description is more often used, for prediction and other purposes. Lagrangian analysis, though, is a powerful way to analyze fluid flows when tracking and understanding the origins and fates of individual particles are important (Bennett, 2006). The fluid envelopes of the Earth System, for instance, exhibit a wide variety of dynamical motions that can act quite differently on mixing and transport. In the ocean, for instance, fluid particles carry tracers such as nutrients, plankton, heat, salt or marine debris (Van Sebille et al., 2018). Hence, in the climate sciences, we are often interested in how particles in the ocean or the atmosphere move and how this motion affects tracer transport.

The oft observed formation of ordered patterns in fluids with complex behavior has led to the search for a theory that could explain Lagrangian coherence in terms of an underlying skeleton responsible for structuring the pathways of sets of fluid particles. These structures may have a finite lifetime and so, one refers to them as finite-time coherent sets e.g. (Williams et al., 2015). Sensitivity to initial conditions makes Lagrangian fluid motion inherently unstable, calling for methods from nonlinear dynamical systems theory (Haller, 2015). In this section, we show how algebraic and chaos topology can help one understand transport in fluid flows (Charó et al., 2020, 2021a) and, more specifically, we demonstrate BraMAH's potential in this setting.

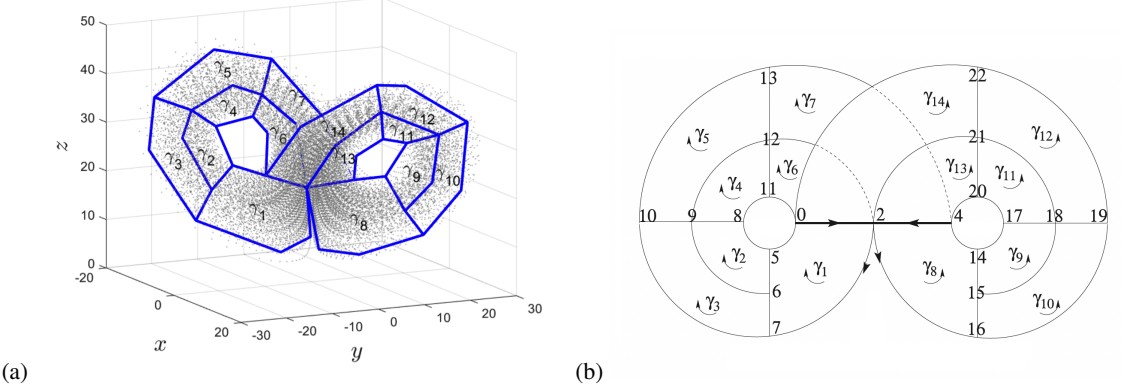

**Figure 11.** BraMAH analysis of the Lorenz (1963a) attractor. (a) Cell complex with the cells constructed as an approximation to subsets of points in a point cloud with $N_0 = 25\,000$ points; reproduced from Charó et al. (2022b) under CC-BY license. (b) A diagram showing the labels of the vertices forming the polygonal 0-cells of the complex, as well as the orientation of the 2-cells $\{\gamma_i : i = 1, 2, \ldots, 14\}$. The heavy horizontal line in panel (b) indicates the singular line that unites the two branches; reproduced from Charó, G. D., Letellier, C., and Sciamarella, D.: Templex: A bridge between homologies and templates for chaotic attractors, Chaos: An Interdisciplinary Journal of Nonlinear Science, 32, 083 108, 2022 with the permission of AIP Publishing.

The Unsteady or Driven Double Gyre (DDG) system is an analytic model, often used to show how much Lagrangian patterns may differ from patterns in Eulerian fields. Shadden et al. (2005) introduced the DDG model to mimic the motion of two adjacent oceanic gyres enclosed by land and, since the work of Sulalitha Priyankara et al. (2017), it is known to present chaotic transport between the two counter-rotating laterally oscillating vortices. The Lagrangian model is defined by the set of
720 ODEs

$$\dot{\mathbf{x}}(t; t_0, \mathbf{x}_0) = \mathbf{v}(t; \mathbf{x}(t; t_0, \mathbf{x}_0)), \tag{26a}$$

$$\mathbf{x}(t_0; t_0, \mathbf{x}_0) = \mathbf{x}_0. \tag{26b}$$

Here the initial conditions $\mathbf{x}_0$ lie in a rectangular domain $\Omega = [0, 2] \times [0, 1]$ and $\mathbf{v} = (u, v)$ is the Eulerian velocity field, which is derived from the streamfunction $\psi = \psi(x, y, t)$ given by

725 $\quad \psi(x_1, x_2, t) = A \sin(\pi f(x_1, t)) \sin(\pi x_2), \tag{27a}$

$$f(x_1, t) = a(t)x_1{}^2 + b(t)x_1, \quad a(t) = \eta \sin(\omega t), \quad b(t) = 1 - 2a(t). \tag{27b}$$

The usual parameter values are $A = 0.1$, $\eta = 0.1$ and $\omega = \pi/5$. Note that $u = -\partial\psi/\partial y, v = \partial\psi/\partial x$ and hence the flow is nondivergent at all times, $\partial u/\partial x + \partial v/\partial y = 0$.

Clearly, this DDG model is nonautonomous for $\eta \neq 0$, since the coefficients $a, b$ are periodic in time. Note, however, that the
730 streamfunction $\psi(x_1, x_2, t)$ given by Eqs. (27a, 27b) would not correspond to a solution of the Navier-Stokes equations in two dimensions: it is a synthetic example that (i) exhibits somewhat familiar oceanic flow patterns; and (ii) chaotic behavior within

certain subsets of the induced particle motion (Shadden et al., 2005). In fact, more realistic Eulerian flows that are solutions of the so-called quasi-geostrohic equations governing the wind-driven oceanic circulation subject to rotation (Pedlosky, 1987; Dijkstra, 2005) are themselves chaotic, rather than periodic in time, for realistic parameter values (Jiang et al., 1995; Dijkstra and Ghil, 2005).

From the Eulerian perspective, the DDG has a time-periodic and simple behavior, a snapshot of which is shown in Fig. 12 (a). What happens, though, if there is an "oil spill" in the middle of the DDG domain? When injecting a passive tracer, as in Fig. 12(b), blank regions appear, i.e. zones of particles in motion that are never reached by the oil spill, and present circular or triangular shapes. The system being conservative, particle behavior depends on the initial particle position being integrated. The oil spill spreads in a chaotic sea surrounding regular islands containing particles where behavior is quasi-periodic. Between the regular islands and the chaotic sea, there are hermetic transport barriers, inhibiting particle to move from one region to another one. This simple, synthetic example demonstrates therewith that flow patterns can effectively differ depending on whether the system is observed in Eulerian or Lagrangian terms. The transport barriers are not even visible in the Eulerian perspective. For further details on particle behavior, the reader is referred to Charó et al. (2019).

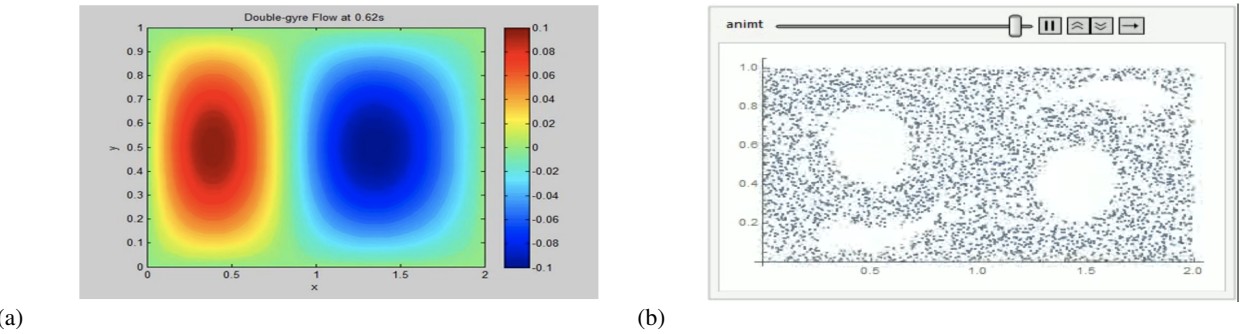

(a)                                                                                   (b)

**Figure 12.** Eulerian and Lagrangian perspectives for a fluid flow in the case of the Shadden et al. (2005) Driven Double Gyre (DDG). (a) Vorticity field; and (b) passive tracer injected in the middle of the domain $(1, 0.5)$. The blank regions are not static: they describe closed orbits within the right and left hemispheres of the domain and deform as they move. The video can be seen at https://youtu.be/W1yndTsvR0g. Particles in these four blank regions are trapped within them and exhibit a regular, nonchaotic behavior, while particles in the region visited by the tracer do exhibit Lagrangian chaos. The DDG is known to have an embedded horseshoe near the point $(1, 1)$ (Sulalitha Priyankara et al., 2017). Subsets of particles behaving alike — and therefore sharing the same topology — swarm together robustly.

How can BraMAH help us in Lagrangian analysis? The interesting cases, as shown by the DDG example, correspond to dynamical systems that are nonautonomous. But in such systems, some processes involved in the particle dynamics derived from the Eulerian streamfunction are not explicitly described in the two-dimensional space spanned by the particle positions' coordinates. Many authors choose to work in an "extended phase space," in which time is added as a phase space coordinate.

But such an extended phase space is in fact deceptive, since it assigns a double status to the time variable, which should not play the role of both an independent and a dependent variable. Due to this double status, some tools from autonomous

dynamical systems theory do not apply (Charó et al., 2019). The importance of this point in Topology of Chaos should not be neglected. In fact, one of the fundamental hypotheses in writing a dynamical system as a set of ODEs is that time is the only independent variable, while all state variables are time dependent.

Working in a space whose dimension is increased by one due to introducing the extra ODE $\dot{t} = 1$ leads to certain difficulties in using the tools borrowed from nonlinear dynamical systems theory — for instance, the state space is no longer bounded. In this extended phase space, a periodic orbit is no longer a closed curve, simply because when the system returns to the same state, it does not return to the same point. The very definition of phase space in which a point represents one-to-one a state of the system is no longer valid in the extended phase space.

Many of the properties that are valid in a well-defined phase space are altered in an extended phase space, and topology is one of them. In the case of the DDG model discussed by Charó et al. (2019), the starting point is a nonautonomous system of two ODEs. The extended phase space — with a third ODE written as $\dot{t} = 1$ — is three-dimensional. But the paper shows that a fourth dimension is needed to rewrite the system as an autonomous set of ODEs without using the standard extension trick. The genuine phase space of the autonomously written driven double gyre has four ODEs: two additional variables are required, $u$ and $v$.

Such a transformation gets rid of the explicit time dependence with a legitimate procedure that does not run into the previously explained inconsistency. In this four-dimensional phase space, and for certain initial conditions, the topological structure that is obtained is a Klein bottle. A Klein bottle cannot be immersed into a 3-D space without self-intersections: the role of the fourth dimension that is required to rewrite the system in an autonomous form is, therefore, highly relevant here. Thus, to use topological tools self-consistently, one must be prepared to work in a well-defined phase space, and with as many dimensions as required.

In the fluid-flow problem, the four-dimensional phase space complements the Lagrangian variables by an indirect representation of the Eulerian variables. A knotless approach like BraMAH does allow one to work in such a space, which was previously out of reach for a topological analysis. As we shall see, though, in Sec. 3.4, a more general approach to the topological study of NDS and RDS problems is to extend the time-independent BraMAH of Sec. 3.1 and the associated templexes of Sec. 3.3 to the corresponding time-dependent cases.

When applied to time series describing particle trajectories in fluid flows, BraMAH falls within a family of methods that measure complexity of individual trajectories to identify coherent regions, i.e., regions with qualitatively different dynamical trajectory behavior. Rypina et al. (2011), for instance, use correlation dimension as a measure of complexity. Correlation dimension, though, is a metric invariant, which does not provide information on how to model the system's dynamics. Charó et al. (2020) applied BraMAH to Lagrangian trajectories $\mathbf{x}(t; t_0, \mathbf{x}_0)$ and obtained the topology of the associated branched manifold in the full four-dimensional phase space of the DDG equations in their Lagrangian form (26). This result is achieved by deriving the recipes that knead the DDG model's dynamical behavior in phase space, without having to look into the geometrical complexity of individual particle trajectories.

Returning now to the "oil spill" in the middle of the DDG system's domain $\Omega$, Charó et al. (2020) applied BraMAH to

8528 fluid particles in a four-dimensional reconstructed phase space. Only five distinct topological classes emerge, and their characteristic cell complexes are plotted in Fig. 13.

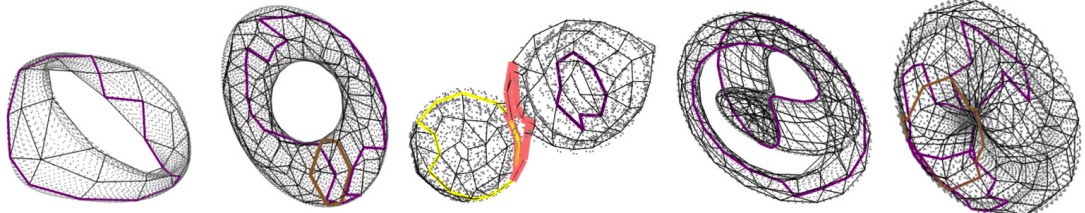

**Figure 13.** The analysis of a set of 8528 particles advected by the DDG flow field yields five topological classes. These five classes are obtained by applying BraMAH to four-dimensional point clouds; the plots in the figure are three-dimensional projections of representative cell complexes for each of the five classes. Four of them involve quasi-periodic particle motion, and only one of them, which is represented by the third cell complex, points to a branched manifold that refers to the so-called chaotic sea (colored in blue in Fig. 14 below). The 1-cells, i.e., the lines that are highlighted in color, indicate the generators of the homology group $H_1$ of holes in each cell complex. From Charó et al. (2021a) with permission with permission from Cambridge University Press.

From left to right, Class I corresponds to a strip (green), Class II to a torus (magenta), and Class III to a branched manifold with three 1-holes — i.e., with a Betti number $\beta_1 = 3$ — and a torsion that is indicated by the orientability chain (blue). The remaining complexes are of Class IV, with the topology of a Klein bottle (red), and of Class V, which is a very peculiar kind

of torus that involves a torsion and a weak boundary (orange). The colors assigned to each topological class in this list are used in Fig. 14 to tag the particles in motion, and thus identify distinct particle sets that stay coherent while moving and being distorted. The frontiers between differently colored regions will be called *separators*. Such flow separators are associated with LCSs that are known to separate dynamically distinct regions in fluid flows (Kelley et al., 2013).

The presence of the Klein bottle as Class IV among the five classes in Fig. 13 stresses the importance of being able to work

in a sufficiently high-dimensional phase space that guarantees an autonomous setting: as mentioned in Sec. 3.1 before, the Klein bottle cannot be immersed in three dimensions without self-intersections.

Charó et al. (2021a) further emphasized that BraMAH can identify and describe LCSs in a fluid flow from a sparse set of particles, and achieve this without inspecting relative particle positions. The method differs from previous ones because it describes transport by how particles behave without looking at where they go. Such a dynamical analysis ends up pointing to

Finite-Time Coherent Sets, thanks to the property that particles sharing equivalent dynamics tend to stay together. The same authors have also successfully used BraMAH to study numerically generated fluid particle behavior in the wake behind a rotary oscillating cylinder (Charó et al., 2021a).

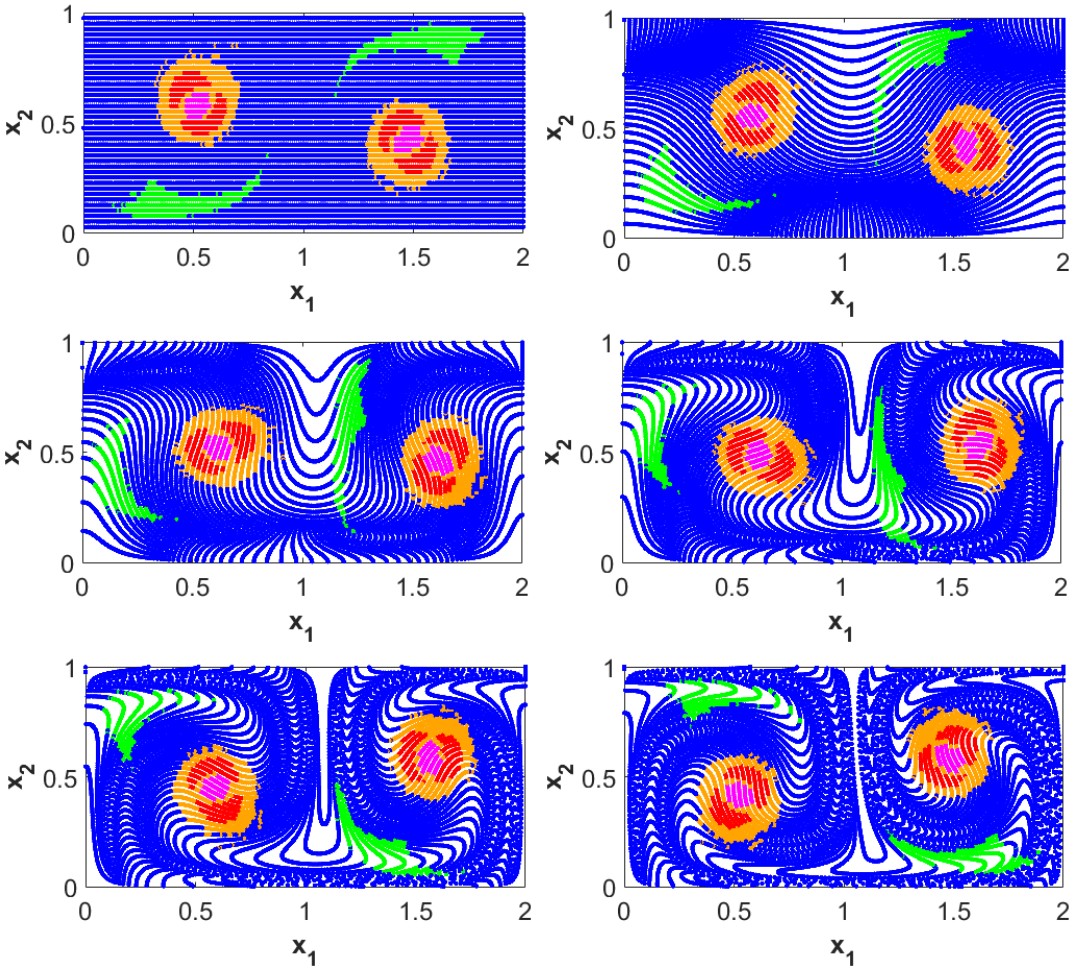

**Figure 14.** Coloring of 8528 particles in motion in a DDG field, with colors corresponding to the topological structure of the particle trajectories in phase space. The boundaries between distinct colors are fairly well defined, displaying the existence of transport barriers that separate non-mixing regions, like the green, orange, red or magenta, vs. the chaotic sea (blue). A direct correspondence is found between the regions identified by the topological BraMAH analysis and those observed dynamically using a Poincaré section, as in Charó et al. (2019, Fig. 5), or a finite-time Lyapunov exponent study, as in You and Leung (2014, Fig. 12). From Charó et al. (2021a), with permission from Cambridge University Press.

The BraMAH applications reviewed in this subsection demonstrate substantial progress in Lagrangian analysis, by providing a method that enables one to identify coherent sets without previous knowledge of the flow field. This particular set of results also shows methodological progress in chaos topology, since it appears that BraMAH can help describe the topological structure of nondissipative, Hamiltonian systems. Recall, as a stepping stone in this direction, the analogy between the nondivergence of a fluid flow in physical space, like the DDG model, and the Hamiltonian character of a dynamical system's flow conserving volume in phase space, like the equations of celestial mechanics (Poincaré, 1892, 1893, 1899; Arnold et al., 2007).

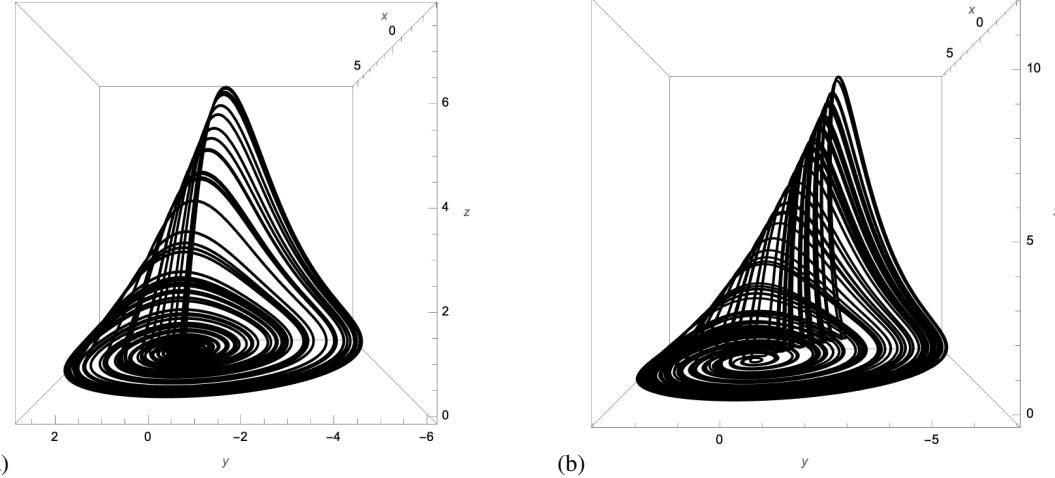

**Figure 15.** Solution trajectories for (a) the spiral type ($a = 0.343295, b = 2, c = 4$); and (b) the funnel type ($a = 0.492, b = 2, c = 4$) attractors of the Rössler (1976) model.

### 3.3 Templexes for Dynamical Systems

Structures in phase space are special, because they are not just spatial objects: they are associated with a semi-flow on them, which is sometimes represented by arrows. A cell complex can effectively encapsulate the properties of a branched manifold in standard space, but it will not convey the fact that, when the cells in a complex represent a semi-flow on a spatial object, they can be traversed in an arbitrary order only at the expense of forgetting about the semi-flow. In other words, time is absent from the description. Including the arrow of time into the description calls for a more refined mathematical object, in which

the topological properties of a flow in phase space come into light through the combined analysis of both the spatial structure of the underlying branched manifold and of the semi-flow upon it.

Charó et al. (2022a) introduced such a novel type of mathematical object and called it a *templex*, a word obtained from the contraction between "template" and "complex." A template in dynamical systems theory is a synonym for a knot-holder (Birman and Williams, 1983a; Tufillaro et al., 1992; Ghrist et al., 1997). Since Mindlin and Gilmore (1992), templates have

been used to describe three-dimensional flows from experimental data in many fields: to study a three-species food chain model in ecology (Letellier and Aziz-Alaoui, 2002); to forecast the time series of sunspot numbers (Aguirre et al., 2008), or to better understand delayed interactions between cancer cells and the micro-environment (Ghosh et al., 2017). Albeit limited to three dimensions, a template provides a description of an attractor at a level of detail that homologies alone cannot achieve.

The Rössler (1976) model,

$$\dot{x} = -y - z, \tag{28a}$$

$$\dot{y} = x + ay, \tag{28b}$$

$$\dot{z} = b + z(x - c). \tag{28c}$$

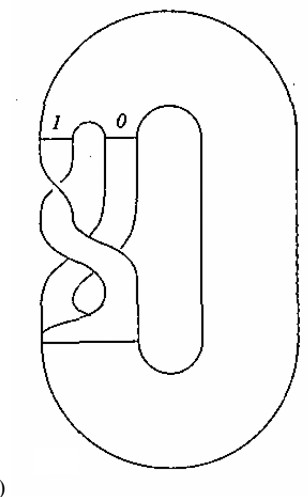 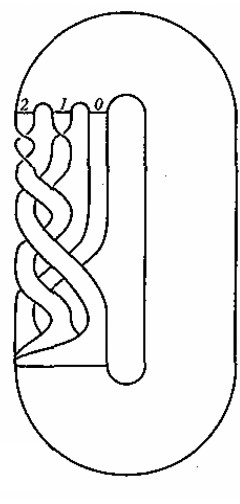

(a)             (b)

**Figure 16.** Templates for the (a) spiral type and (b) funnel type attractors for the Rössler (1976) model. Reproduced from Letellier, C., Dutertre, P., and Maheu, 3: Unstable periodic orbits and templates of the Rössler system: Toward a systematic topological characterization, Chaos, 5, 271–282, 1995 with the permission of AIP Publishing.

provides a simple example. Changing two of the parameter values in the governing equations (28b, 28c), one can produce two distinct chaotic attractors, shown in Fig. 15: (a) the spiral case, with $(a = 0.343295, b = 2, c = 4)$; and (b) the funnel case, with $(a = 0.492, b = 2, c - 4)$. As discussed by Charó et al. (2022a), these two structures can be approximated by cell complexes that are homologically equivalent. But templates are able to discriminate between the two cases using the concept of strip.

For strongly dissipative systems, like the Rössler attractor, the number of monotone branches of the first-return map provides the number of strips required to construct the corresponding template. Strips are cylinders, in topological terms, but one must beware that the meaning of strip in a template is not introduced to refer to a topological class, but to discriminate between the different paths followed by the flow along the branched manifold. A strip is typically defined between a splitting chart and a joining chart, in which the strips are split and joined respectively. Thus, in the template terminology, the spiral attractor has two strips, while the funnel attractor has three strips, as shown in Figs. 16(a) and 16(b), respectively.

Strips in a template are associated with a tearing of the flow. They are sometimes split in a fictitious manner, introducing false holes into the branched manifold, even if these strips are not necessarily delimited by boundaries or associated to holes in the sense of homologies. Their number can be obtained, for strongly dissipative systems, by computing the number of monotone branches of the first return map. But where are these strips in a cell complex? As mentioned above, they cannot be directly identified with holes in the latter. Can they be identified all the same from some other properties of the cell complex? The short answer is yes, but not without the information that is contained in the flow on the cell complex, rather than just in the cell complex itself.

The templex thus combines all the essential information that is relevant to the topology of the branched manifold and to the flow on it. The flow on the cell complex is represented by a directed graph (digraph) (e.g., Bang-Jensen and Gutin, 2008), whose

nodes are the highest dimensional cells and whose edges, or arcs, are provided by the cell connections that are consistent with the flow. In a templex, the cell complex and the digraph are interrelated. Computations carried out on the two complementary objects yield a description of the branched manifold and of the permitted nonequivalent paths around it.

Algebraic computations on a templex provide, on the one hand, the already known properties of the cell complex — such as the homology groups, torsion groups, and weak boundaries — that describe the branched manifold; on the other hand, they provide the properties of the flow on this structure. The topology of a templex is described in terms of a set of sub-templexes that will be called stripexes, since they play the same role as strips in a template. This is no longer done at the price of introducing false holes or boundaries to separate the strips. It is achieved through a set of well defined operations that include

flow-orienting the cell complex; minimizing the cell structure at the joining loci, where the tearing of the flow takes place, to obtain a generating templex; calculating the cycles of the digraph; and checking for local twists, since uneven torsions in a strip correspond to a local twist in a stripex. The reader is referred to the steps in Charó et al. (2022a) for further details. This dissection of the cell complex into stripexes provides the information that enables one to distinguish the topological properties of the two Rössler attractors from each other. In order to see how, consider Fig. 17 that illustrates the templexes for the two

types of Rössler (1976) attractor.

    The cell complex of a templex can be seen as a dynamic kirigami, or cut-out paper model, made of pieces that fit together; in this case, the pieces are polygons. Notice that points or segments with the same label must be glued together when constructing the paper model. The digraph can be seen as a map of the flow-compatible connections between the pieces. Combining the cell complex and the digraph, we can define and algebraically compute the stripexes. For details on this procedure, the reader

is again referred to Charó et al. (2022a). The stripexes for the spiral attractor are given by two paths along the cell complex, indicated by the two cycles below, the first of which is twisted.

$$\gamma_1 \to \gamma_2 \to \gamma_4 \to \gamma_6 \to \gamma_1, \tag{29a}$$

$$\gamma_1 \to \gamma_2 \to \gamma_3 \to \gamma_5 \to \gamma_1. \tag{29b}$$

There are three stripexes for the funnel attractor, as shown below, and only the middle one presents a local twist:

$$\gamma_1' \to \gamma_2' \to \gamma_4' \to \gamma_6' \to \gamma_1', \tag{30a}$$

$$\gamma_1' \to \gamma_2' \to \gamma_3' \to \gamma_5' \to \gamma_1', \tag{30b}$$

$$\gamma_1' \to \gamma_2' \to \gamma_3' \to \gamma_6' \to \gamma_1'. \tag{30c}$$

The description in terms of stripexes provided by the two templexes in Eqs. (29, 30) is equivalent to the strips in the templates of the spiral and the funnel case of the Rössler (1976) attractor, as shown in Fig. 16. Let us recall that templates are knot-

holders, and can therefore only be obtained for three-dimensional flows, while templexes can be computed for four- or higher-dimensional dynamical systems, as shown in Charó et al. (2022a, Sec. IV).

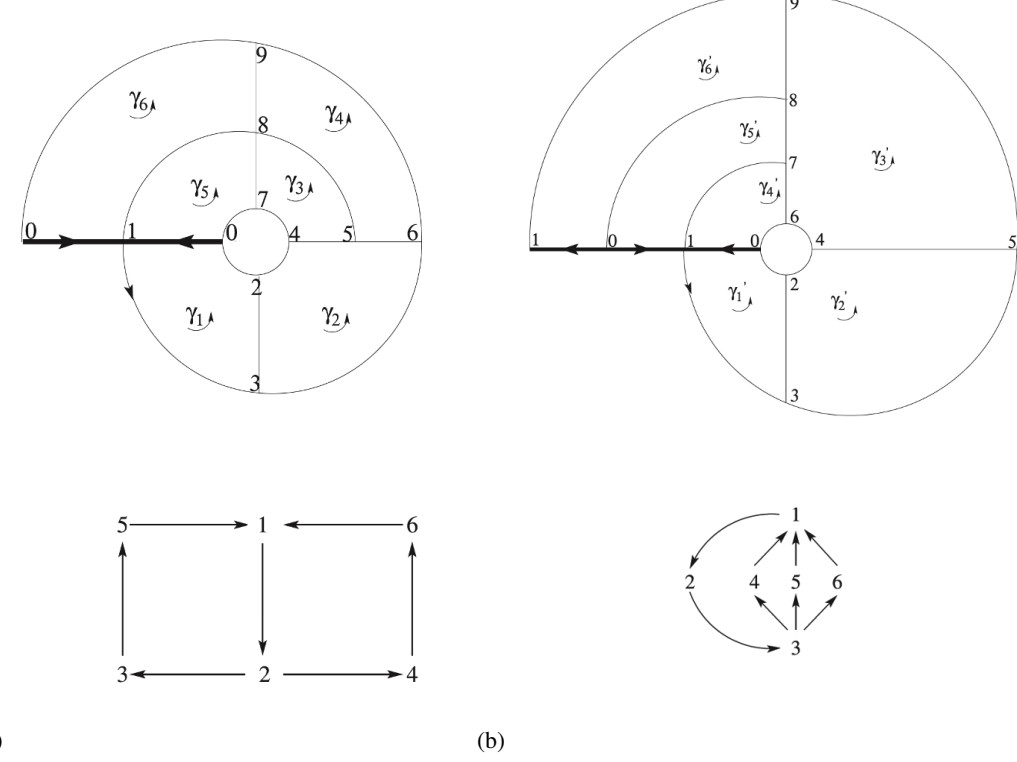

(a)            (b)

**Figure 17.** Templexes for (a) the spiral and for (b) the funnel attractors corresponding to the structures in phase space shown in Fig. 15(a) and (b), respectively. Cell complexes (above) are shown as planar diagrams, with the convention that points (0-cells) and segments (1-cells) with identical labels must be glued to each other. The digraphs provide the allowed connections between the polygons (2-cells) labeled $\gamma_i$ for the spiral case, and $\gamma_i'$ for the funnel case, with $i \in \mathbb{N}$. The two cell complexes are homologically equivalent. Reproduced from Charó, G. D., Letellier, C., and Sciamarella, D.: Templex: A bridge between homologies and templates for chaotic attractors, Chaos: An Interdisciplinary Journal of Nonlinear Science, 32, 083 108, 2022 with the permission of AIP Publishing.

## 3.4 Algebraic Topology and Noise-driven Chaos

BraMAH and the associated templexes, as presented so far, provide a topological description that holds within an autonomous and deterministic framework. As discussed in Sec 2.2 regarding dynamical system theory for the climate sciences, the question that naturally arises, though, is whether we can take one step beyond; namely extend the topological perspective to NDSs and RDSs, which provide the appropriate mathematical framework to tackle the effects of time-dependent forcing on intrinsic climate variability (Ghil, 2019; Ghil and Lucarini, 2020; Tél et al., 2020). Of the two forms of time-dependent forcing, it is the random one that is more challenging. Moreover, the topological characterization of noise-driven chaos is crucial in the understanding of complex systems in general, where part of the dynamics remains unresolved and is modeled as noise.

An example involving not only deterministic time dependence but also random forcing was presented in Eqs. (25) and Fig. 8 of Sec. 2.2. In the stochastically perturbed Lorenz (1963a) model's random attractor, termed LORA (Chekroun et al., 2011), the stretching and folding mechanisms shape the flow in phase space yielding a time-evolving branched manifold, which must be analyzed accordingly. Nothing prevents one from applying BraMAH to successive point clouds, each of which corresponds to a single snapshot, and comparing the topological properties of these instantaneous cell complexes, as done for the first time
by Charó et al. (2021b).

    Such an analysis was performed by Charó et al. (2021b) for a fixed realization of the driving noise $d\eta$ at different instants in time. In order to construct the cell complexes, these authors first sieved the LORA point clouds to retain the most populated regions in phase space. The deterministic concept of branched manifold (Williams, 1974) was extended to the stochastic framework by redefining it locally as an integer-dimensional set in phase space that robustly supports the point cloud associated
with the system's invariant measure at each time instant. The numerical results show that BraMAH captures LORA's time-evolving homologies (Charó et al., 2021b), as shown here in Fig. 18. The topologies differ from the deterministic Lorenz model's strange attractor, and the noise-driven model's branched manifold exhibits sharp topological transitions in time.

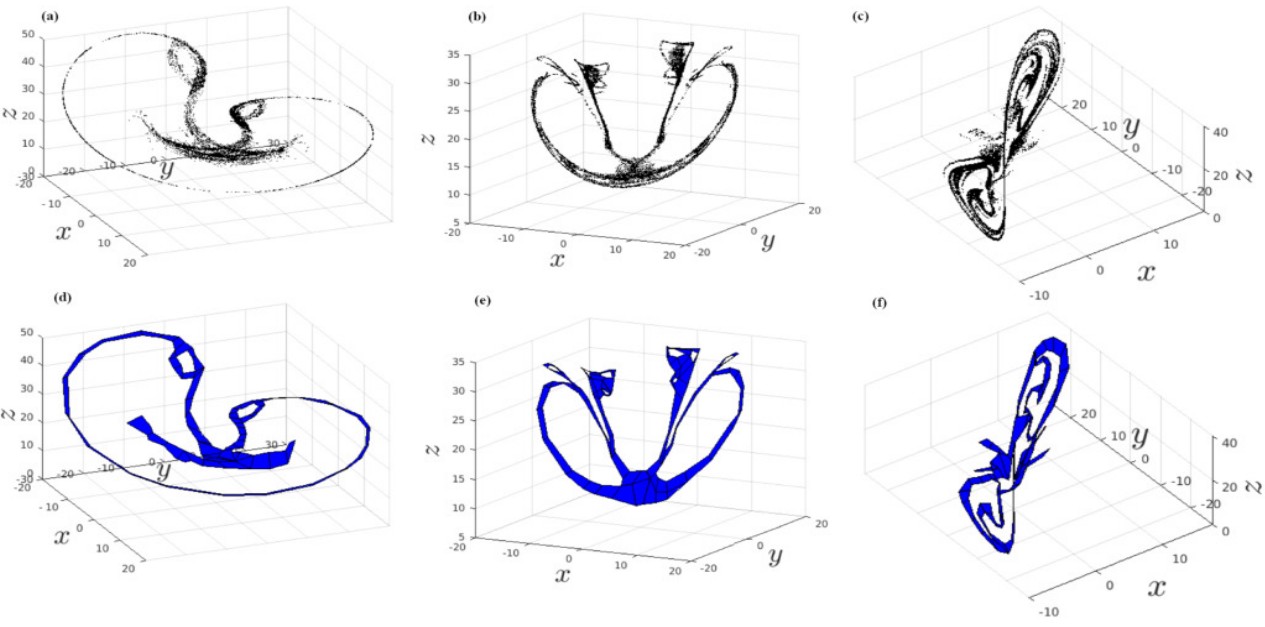

**Figure 18.** Three LORA snapshots with the noise variance $\sigma = 0.3$ and cloud size $N_0 = 10^8$. Sieved point clouds (a)–(c) and cell complexes (d)–(f); (a,d) $t = 40.09$; (b,e) $t = 40.18$; and (c,f) $t = 40.27$. The cell complexes are not homologically equivalent from one snapshot to another: their Betti numbers are $\beta_1 = 3, 10$ and $4$ for $t = 40.09, 40.18$ and $40.27$, while the Betti number for the deterministic strange attractor in Fig. 11 is $\beta_1 = 2$, stemming from the two holes around the two convective fixed points on either "wing" of the butterfly. Reproduced from Charó, G. D., Chekroun, M. D., Sciamarella, D., and Ghil, M.: Noise-driven topological changes in chaotic dynamics, Chaos, 31, 103 115, 2021 with the permission of AIP Publishing.

The stochastic branched manifold, characterized by a single cell complex for each snapshot, does not contain any information about the future or the past of the invariant measure. The flow in a cell complex representing the invariant measure on a random attractor can no longer be represented within that cell complex, as done when using a deterministic templex, like the one described in Sec. 3.3 for the Rössler (1976) model. Incorporating time into this formalism requires establishing a link between the cell complexes of distinct snapshots.

But how can one track changes between different cell complexes without using specific individual cells? Let us recall that the number of cells and their distribution in a cell complex are arbitrary, and that homology groups are conceived so as to cancel out the extraneous information in the cells and to only retain the essential properties of the topological space. Homologies will thus provide the key to connect a cell complex of a random attractor at a given instant to a cell complex corresponding to another instant. For a random attractor, we will endow a set of cell complexes with a digraph that does not connect cells within a single complex, as in Fig. 17, but holes of cell complexes at distinct instants of time. This is the key idea that led Charó et al. (2022b) to construct their *random templexes*.

Tracking holes requires some caveats, though. Homology groups and the associated Betti numbers are independent of the particular set of cells forming a cell complex. Hence, the holes or generators of a homology group can be expressed in terms of one of several representative cycles that need not strictly follow the boundary of the holes, as shown in Fig. 19. A representative cycle may wander around a hole, without tightly encircling the empty space. Still, the boundaries of the holes can be retrieved algebraically, from the cell complex itself, as shown by Charó et al. (2022b). We can thus define a *random templex* as an indexed family of BraMAH cell complexes hanging together by a digraph. In this digraph, each node is a minimal hole of a given cell complex and the edges, or arcs (Bang-Jensen and Gutin, 2008), denote the connections between minimal holes occurring at successive time instants.

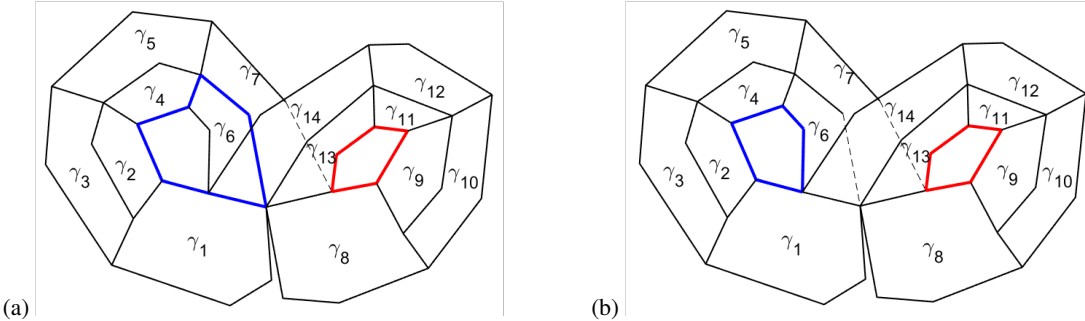

**Figure 19.** Cell complex of the deterministic Lorenz (1963a) model's attractor, as shown in Fig. 11(a) here. Emphasized in color in this figure are (a) the holes obtained in the homology computation; and (b) the tight or minimal holes. From Charó et al. (2022b) under CC-BY license.

What does the random templex, thus defined, encode? In the life of a random attractor, there may be time intervals within which the branched manifold evolves geometrically but maintains its homological properties. Topology can be said to change when the holes that are being tracked from a snapshot to the next are created or destroyed. Some of them can be found to split or merge. Such changes are associated with what we call hereafter a *Topological Tipping Point (TTP)* (Charó et al.,

2021b, 2022b). Since the Betti numbers are integers, any changes in them must be sudden. In fact, these sudden changes could already be noticed visually in the LORA video published by Chekroun et al. (2011) at https://vimeo.com/240039610.

To confirm this further, Charó et al. (2021b, Fig. 4) showed that the time intervals over which the Betti numbers changed drastically were quite short, i.e., no longer than $\delta t = 0.09$, as reproduced in Fig. 18 herein. This time interval is very short indeed, compared to the characteristic time to switch wings for a trajectory of the deterministic Lorenz (1963a) model, which is of the order of units.

Charó et al. (2022b, Fig. 5) further showed that the numerically observed transition intervals over which Betti numbers in LORA change can be even shorter, with $\delta t \leq 0.065$. More interestingly, these authors demonstrated that TTPs can be identified and classified using the digraph of a random templex.

Figure 20 here shows the "story" of two holes in a finite time window $T_w = 40.065 \leq t \leq 40.110$ of LORA's life in the form of two tree plots; the two holes, 73 and 74, lie on opposite wings of the LORA butterfly at the window's initial time. For the sake of simplicity, we kept only two of the fifteen connected components of the complete finite-time random templex of LORA for $T_w$; see Charó et al. (2022b, Fig. 6) for the complete picture. Square nodes correspond either to an initial or to a final node for a given time window. A splitting TTP occurs where two or more edges emerge and a merging node receives two or more edges. Similarly, there is a creation or annihilation TTP where an initial or a terminal node in a connected component of the digraph does not correspond to the boundaries of the time window: square nodes cannot be TTPs since the preceding or following instant in time is outside the inspected time window.

The indices in Fig. 20 label a hole at a certain instant. Tracking enables one to connect, for instance, hole 73 with 91, which will split into holes 108 and 109; this is why hole 91 is colored in red. A symmetric splitting event can be found on the other wing of the animated butterfly, where hole 74 becomes hole 96, which splits into holes 112 and 116. All these holes can be located in phase space using the coordinates of each hole's barycenter. Plotting the position of the barycenters of all the holes present in the analysis in phase space, we obtain a *constellation set*, as shown in Fig. 21. Each constellation contains the immersed nodes and edges forming a connected component in the digraph and transforms the tree plots into actual paths in phase space. In other words, embedding the digraph of the random templex into phase space, one can represent parsimoniously the evolution of LORA's topology over a given time interval. Such a representation might provide access to a more detailed description of the flow dynamics in a random attractor.

## 4  Concluding Remarks

The purpose of this paper was to provide an account of the convergence between two strains of Henri Poincaré's heritage — dynamical systems theory (Poincaré, 1892, 1893, 1899, 2017) and algebraic topology (Poincaré, 1895; Siersma, 2012) — and their joint applications to the climate sciences.

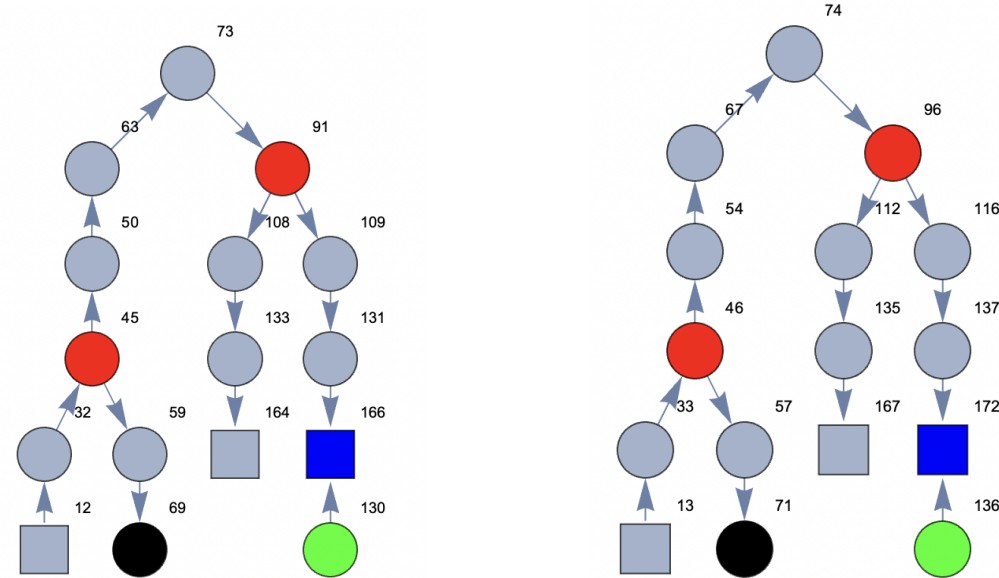

**Figure 20.** A "day in the life" of two mutually symmetric holes of LORA for a fixed noise realization and noise intensity $\sigma = 0.3$. The time window is $T_w = [40.065, 40.11]$. The nodes are highlighted in different colors according to the type of event: creation in green, destruction in black, splitting in red, and merging in blue.

## 4.1 Summary

In Sec. 1, we provided a bird's eye view of the evolution of these two strains of research since the mid-20th century and how they started to be applied to issues related to fluid flows at both engineering and planetary scales. Sections 2 and 3 developed next in greater detail (a) the concepts and methods associated with dynamical systems and their applications to the climate sciences, and (b) those associated with algebraic topology and their applications first to engineering fluid dynamics and then to the climate sciences. Notice that the pioneering references mentioned in Secs. 1.1 and 2 date back to the early 1960s, while those of Secs. 1.2 and 3 start in the early 1980s. It is clear that, on the whole, algebraic topology has started playing a noticeable role in the climate sciences about two decades later than dynamical systems theory.

Section 2.1 covered autonomous dynamical systems, in which neither the forcing nor the coefficients depend explicitly on time. A very extensive and thorough mathematical theory exists and certain aspects of it are well known to a substantial fraction of climate scientists; see, for instance, Ghil and Childress (1987) and Dijkstra (2013). The contents of this section emphasized elementary bifurcations — saddle-node and fold, pitchfork, and Hopf, summarized in Fig. 4 — ending with bifurcation trees and routes to deterministic chaos.

The material in Sec. 2.2 refers to systems with explicit time dependence in the forcing or the coefficients and it is much newer. The theory of NDSs and RDSs only started in the 1960s — with George Sell, followed by Ed Ott and colleagues and by

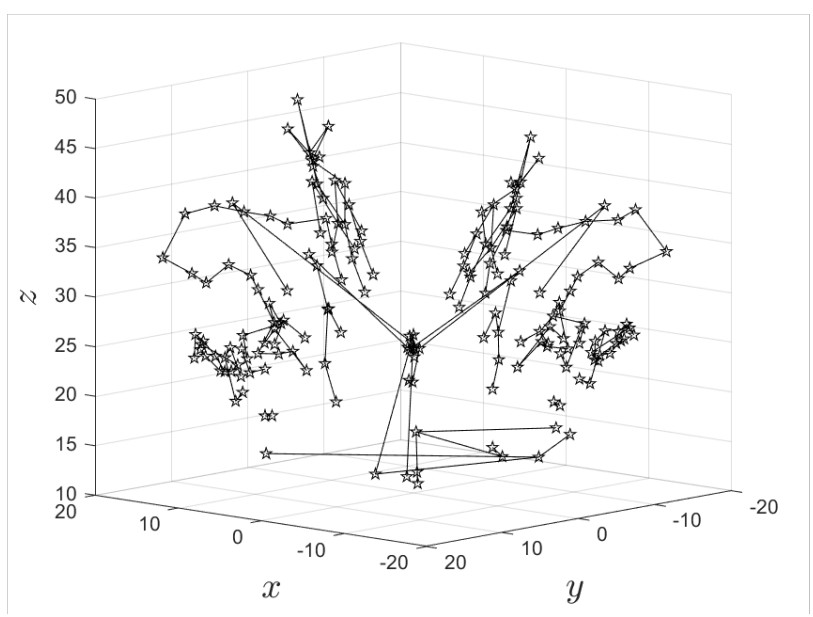

**Figure 21.** A particular constellation out of the set that represents the essence of the evolution of LORA's 1-holes within a time window $T_w = [40.065, 40.11]$. The plot shows the embedding into the Lorenz (1963a) model's phase space of a connected component in the digraph of LORA's random templex, by using the coordinates of the barycenters of the nodes. Regular nodes in a constellation are marked by open stars, while nodes associated with TTPs — such as splitting or merging of holes — are marked by filled stars. From Charó et al. (2022b) under CC-BY license.

Ludwig Arnold, Hans Crauel and Franco Einaudi (Sell, 1971; Romeiras et al., 1990; Crauel and Flandoli, 1994; Arnold, 1998; Caraballo and Han, 2017; Kloeden and Yang, 2020) — and its applications to the climate sciences started merely a little over a dozen years ago (Ghil et al., 2008; Chekroun et al., 2011; Tél et al., 2020).

We first explained in this section the essential difference between forward and pullback attraction, i.e., between convergence in time of single-parameter and two-parameter semi-groups of solutions to the governing equations. Simple examples of pullback attractors (PBAs) were given to familiarize newcomers with the appropriate concepts and methods; see again Figs. 5 and 6. The sequence of examples was concluded with the striking random attractor of the stochastically perturbed Lorenz model, as introduced and studied by Chekroun et al. (2011); see Figs. 7 and 8. Finally, tipping points were introduced as the proper

generalization to NDSs and RDSs of the elementary bifurcations for autonomous systems described in Sec. 2.1 (Fig. 9).

    In Sec. 3, we presented topological methods in a dynamical systems perspective. We reviewed the advantages of working with homology theory in order to overcome the limitation of a three-dimensional space imposed by using knot theory, since knots simply disentangle in higher dimensions. In Sec. 3.1, we showed that homologies provide a knotless method, and that a BraMAH cell complex can be used to describe the spatial structure of a flow in phase space by using homology group

generators, weak boundaries and torsion groups (Fig. 11). We described an application of these concepts and methods to

Lagrangian analysis in Sec. 3.2, by showing how to define and detect localized coherent sets (LCSs) for fluid flows in physical space; see again Figs. 12–14.

In Sec. 3.3, we dealt with the fact that BraMAH alone does not provide a robust skeleton of the flow in phase space on its branched manifold, even for an autonomous, deterministic system. To obtain such a robust and parsimonious flow description in phase space, we introduced a directed graph (digraph), whose nodes are the cells, while the edges point from one cell to another, in a way that is consistent with the flow on the branched manifold. The mathematical object that combines such a digraph with the underlying cell complex is called a templex. Homologically equivalent attractors — such as the spiral and funnel versions of the Rössler (1976) attractor — can be distinguished using a templex, given its digraph's properties; see Figs. 15–17.

Finally, in Sec. 3.4, we discussed how a digraph, and hence a templex, can be generalized from the autonomous and deterministic version of Sec. 3.3 to nonautonomous and random dynamical systems; see Figs. 18–20. To define a random templex, one needs to shift the perspective from defining a digraph on the single cell complex of an autonomous system to an indexed family of cell complexes at successive instants in time; and the vertices pointing from one cell at time $t = t_j$ to the corresponding one at time $t = t_{j+1}$.

The fact that the change in the Betti numbers $\beta_k$ of a cell complex at $t = t_j$ to the next one is sudden, since they are integers — in particular for the number of holes $\beta_1$ — allowed us to rigorously define topological tipping points (TTPs) as happening at an instant at which such a sudden change does occur. It is these TTPs that are a matter of particular interest for future work in the overlap of the two fields that we considered in this review, namely dynamical systems and algebraic topology.

## 4.2 Perspectives

As usual, when stumbling upon some striking findings, there are two kinds of paths that one might wish to pursue: (i) more general or stronger theoretical results; and (ii) interesting applications. Clearly we have some rather striking findings and we will outline some intriguing paths to pursue, of both kinds, as well as connections between the two kinds of paths.

TTPs in the templex of an NDS or RDS are obviously connected with a lot more detailed information in phase space about the system under investigation than one might suspect from the usual kinds of bifurcation-induced, noise-induced and rate-induced transitions — or B-tipping, N-tipping and R-tipping — discussed by Ashwin et al. (2012) and mentioned here toward the end of Sec. 2.2. But what does that say about changes of the flow in physical space? Could this localization in phase space say something about association with localized sudden changes of the flow in physical space, i.e., with the "tipping elements" of Lenton et al. (2008)? The use of systematically derived reduced-order models (Kondrashov et al., 2015, 2018; Gutiérrez et al., 2021), for which both TTPs and the better understood dynamical tipping points can be computed fairly easily, could help clarify such relationships and the associated precursors of critical transitions.

An interesting example, among many, of localized changes in Earth's physical space is that of persistent anomalies (Dole and Gordon, 1983) or flow regimes (Legras and Ghil, 1985; Ghil and Childress, 1987, Ch. 6) or weather regimes (Hannachi et al., 2017). Strommen et al. (2023) have recently applied a multiparameter PH method (Carlsson and Zomorodian, 2007; Vipond et al., 2021) to decide more objectively the much-debated existence of distinct regimes in the large-scale atmosphere's

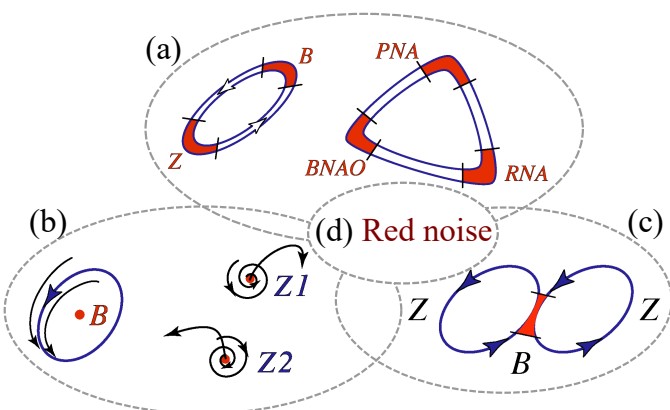

**Figure 22.** Schematic overview of atmospheric LFV mechanisms. From Ghil et al. (2018) with the permission of Elsevier.

phase space (Ghil and Robertson, 2002; Hannachi et al., 2017; Robertson and Vitart, 2018). Their findings certainly strengthen the affirmative reply to the quandary. Before proceeding to the next quandary, though, let us consider briefly the issues that are still open in applying this approach to regime identification.

In applying their multiparameter PH method to the classical Lorenz (1963a) convection model, Strommen et al. (2023) essentially equate the existence of distinct regimes to the existence of two holes in its branched manifold. Doing so, however,
is not quite enough. To explain why, we revisit the example of the Rössler attractor discussed in our Sec. 3.3. This attractor's spiral form, shown in Fig. 15(a), has a single hole, but it has two strips: one strip related to the system's slow branch and the other strip to its fast branch. These two branches, however, are not separated by a hole in the sense of homology groups, and this is why homologies alone cannot distinguish between them. Instead, the templex introduced in the same subsection captures these two ways of circulating around the attractor in terms of two stripexes, despite the fact that the branched manifold
is single-holed.

The existence of multiple regimes in a dynamical system is certainly associated with its attractor's nontrivial topological structure, as Strommen et al. (2023) state, but this nontrivial topology is not necessarily captured by homologies alone. As explained throughout this work, the templex — with its stripexes and digraph — contributes additional tools to accurately describe the phase-space topology of a flow and of its single or multiple regimes.

Ghil and Robertson (2002), however, asked a more subtle question: is the so-called low-frequency variability (LFV) of the atmosphere — which is closely related to the rapidly growing interest in subseasonal-to-seasonal (S2S) predictability (e.g., Robertson and Vitart, 2018) — oscillatory, i.e., wavelike, or episodic and intermittent, i.e., particle-like. These authors, with an obvious nod to the classical problem of quantum mechanics, formulated the question as "waves" vs. "particles." Two decades later, this question is still far from settled, as discussed quite recently by Ghil et al. (2018) and by Ghil and Lucarini (2020).

Which type of phenomena dominate atmospheric LFV? There are two apparently contradictory descriptions: oscillatory, wavelike flow features or geographically fixed, particle-like, episodic flow features; e.g., blocking of the westerlies (particle-

like) or intraseasonal oscillations (wavelike), with periodicities of 40–50 days (Ghil et al., 2018). In fact, these two are by now accompanied by several more key dynamical mechanisms of mid-latitude LFV variability, summarized in Fig. 22.

The simplest approach to persistent anomalies in mid-latitude atmospheric flows on 10–100-day time scales is to consider them as due to slowing down of Rossby waves or to their linear interference (Lindzen et al., 1982; Lindzen, 1986). This approach is illustrated in the sketch labeled (c) within the figure: zonal flow $Z$ and blocked flow $B$ are simply slow phases of an harmonic oscillation, like the neighborhood of $t = \pi/2$ or $t = 3\pi/2$ for a sine wave $\sin(t)$; or else they are due to an interference like that occurring for a sum $A\sin(t) + B\sin(3t)$ near $t = (2k+1)\pi/2$. A more versatile, quasi-linear version of this approach is to study long-lived resonant wave triads between a topographic Rossby wave and two free Rossby waves (Egger, 1978; Trevisan and Buzzi, 1980; Ghil and Childress, 1987, Section 6.2). Neither version of this approach, though, explains the anomalies' organizing into distinct flow regimes.

Rossby et al. (1939) initiated a different, genuinely nonlinear approach by raising the possibility of multiple equilibria as an explanation of preferred atmospheric flow patterns. These authors drew an analogy between such equilibria and hydraulic jumps, and formulated simple models in which similar transitions between faster and slower atmospheric flows could occur. This multiple-equilibria approach was then pursued quite aggressively in the 1980s (Charney and DeVore, 1979; Charney et al., 1981; Legras and Ghil, 1985; Ghil and Childress, 1987, Sections 6.3–6.6) and it is illustrated in Fig. 22 by the sketch labeled (a): one version of the sketch illustrates models that concentrated on the $B$–$Z$ dichotomy (Charney and DeVore, 1979; Charney et al., 1981; Benzi et al., 1986), the other on models (e.g., Legras and Ghil, 1985) that allowed for the presence of additional clusters, like those found by Kimoto and Ghil (1993a) or Smyth et al. (1999), viz. opposite phases of the North Atlantic Oscillation (NAO) and the Pacific North American (PNA) anomalies — dubbed RNA for Reverse PNA and BNAO for Blocked NAO in sketch (a) of Fig. 22. The LFV dynamics in this approach is given by the preferred transition paths of a Markov chain between the two or more regimes.

A third approach is associated with the idea of oscillatory instabilities of one or more of the multiple fixed points that can play the role of regime centroids. Thus, Legras and Ghil (1985) found a 40-day oscillation arising by Hopf bifurcation off their blocked regime $B$, as illustrated in sketch (b) of the figure. An ambiguity arises, though, between this point of view and a complementary possibility, namely that the regimes are just slow phases of such an oscillation, caused itself by the interaction of the mid-latitude jet with topography. Thus, Kimoto and Ghil (1993b) found, in their observational data, closed paths within a Markov chain whose states resemble well-known phases of an intraseasonal oscillation. Kondrashov et al. (2004) confirmed the likelihood of such a scenario in the intermediate-complexity model of Marshall and Molteni (1993). Furthermore, multiple regimes and intraseasonal oscillations can coexist in a two-layer model on the sphere within the scenario of "chaotic itinerancy" (Itoh and Kimoto, 1996, 1997).

Finally, sketch (d) in the figure refers to the role of stochastic processes in LFV variability and S2S prediction, whether it be noise that is white in time — as in Hasselmann (1976) or in Linear Inverse Models (Penland, 1989, 1996; Penland and Ghil, 1993; Penland and Sardeshmukh, 1995) — or red in time, as in Empirical Model Reduction and Multilayer Stochastic Models (Kravtsov et al., 2005, 2009; Kondrashov et al., 2013, 2015; Gutiérrez et al., 2021) or even non-Gaussian (Sardeshmukh and Penland, 2015). Stochastic processes may enter into models situated on various rungs of the modeling hierarchy, from the sim-

plest conceptual models to high-resolution global climate models. In the latter, they may enter via stochastic parametrizations of subgrid-scale processes (e.g., Palmer and Williams, 2009, and references therein), while in the former they may enter via stochastic forcing, whether additive or multiplicative, Gaussian or not (e.g., Kondrashov et al., 2015; Gutiérrez et al., 2021, and references therein). Dorrington and Palmer (2023) recently drew attention to yet another mechanism of interaction between stochastic forcing and nonlinear regime dynamics that might modify the picture.

How might topological data analysis contribute to clarify this thicket of apparently contradictory descriptions of LFV? One hint is the work of Lucarini and Gritsun (2020), who showed that blocking can be studied by extracting from the complex high-dimensional dynamics of a model its essential building blocks, given by truly nonlinear modes. In this work, they abandoned the classic identification of weather regimes with fixed points, as in Charney and DeVore (1979), and directly considered the chaotic nature of the atmosphere, using the unstable periodic orbits (UPOs) that are a key component of the Gilmore (1998) topological analysis of chaos program.

This UPO-based approach did confirm certain theoretical results of Legras and Ghil (1985) and the laboratory findings of Weeks et al. (1997) — about the relative stability and persistence of blocked and zonal flows — as well as providing further insights into the waves-vs-particles quandary (Ghil et al., 2018). UPOs can be very useful in characterizing a chaotic system, since the information about them can be obtained in a finite time, which is particularly useful in nonstationary systems, and because a single UPO can already provide substantial information (Amon and Lefranc, 2004). Still, Lucarini and Gritsun (2020) found it quite hard to carry out the necessary computations of very numerous UPOs even for the relatively simple Marshall and Molteni (1993) model.

As explained here in Secs. 1.2 and 3.1, BraMAH is crucially inspired by the Gilmore (1998) program. Yet it is more powerful than the knots-and-braids methodology, which is limited by the dimensionality of the phase spaces that it can be applied to. Likewise, it is more computationally efficient than the UPO methodology and it provides, as shown at the beginning of this subsection, considerably more information than the PH methodology for chaotic dynamics.

It is thus conceivable, although it remains to be demonstrated, that the additional tools brought to the table by the mathematical object we called templex — namely the digraph and stripexes — could help explore in a highly simplified setting issues like the existence and multiplicity of regimes, as well as of the presence of oscillatory features in the dynamics. As explained in the Rössler (1976) attractor context, stripexes can greatly help, beyond counting holes, to determine regime multiplicity.

Finally, as stated at the end of Sec. 3.4, minimal or tight holes in the cell complex of a random templex can be located in phase space using the coordinates of each hole's barycenter. Plotting the position of the barycenters of all the holes present in the analysis in phase space, yields a constellation set as shown in Fig. 21. The topology of this constellation set deserves further exploration. It might lead, quite conceivably, to the generalization of a stripex for a random templex, and therefore to the extraction of the nonequivalent paths that a nonlinear system follows when driven by multiplicative noise. Random stripexes should provide us with the stretching, squeezing, folding and tearing mechanisms that knead, mold and alter the topological structure of a noise-driven flow in phase space.

The extension of the templex from autonomous and deterministic systems (Charó et al., 2022a) to nonautonomous and stochastic ones (Charó et al., 2022b) opens the way to the exploration of key aspects of the LFV quandaries associated with

Fig. 22. More broadly, it can facilitate exploring a plethora of climate problems that are strongly affected by time-dependent forcing, such as anthropogenic greenhouse gas and aerosol emissions, and stochastic components, such as cloud microprocesses. One can imagine, for instance, applying methods from network theory (Bang-Jensen and Gutin, 2008; Coluzzi et al., 2011; Colon and Ghil, 2017) to investigate the presence of cyclicity in a given model's or dataset's digraph, as well as issues of multimodality or multistability.

More broadly, complex networks (Zou et al., 2019) have found numerous applications in the climate sciences in recent years and could provide other links between topology and the multivariate time series analysis of nonlinear phenomena. The field of complex networks shares many of the challenges that are faced by the topology of chaos. Algebraic topology is not mentioned in the cited review, but there have been some papers applying PH methods to complex networks (De Silva and Ghrist, 2007; Horak et al., 2009; Petri et al., 2013). The network approach is used to reconstruct the phase space, which is a preliminary and certainly necessary step for the analysis of the topological structure of flows from data.

The PH framework to obtain families of nested cell complexes from point clouds has only been mentioned in passing in this review article for the sake of brevity; it should be taken into account, though, as an important of branch of computational topology that is continuously providing us with solutions to algorithmic problems being faced in chaos topology and the climate sciences. So far, the complex network community seems to be lacking a dual object such as the templex to deal with nonstationarity. Finding such an object that captures the spatial structure and is, in addition, endowed with another object that captures the flow structure on the spatial object, appears to be a worthwhile challenge.

*Code/Data availability*. Not relevant for a review paper. Please consult papers cited, if relevant.

*Author contributions*. The authors have contributed equally to the work on this review paper and to its writing. Their names are in alphabetical order.

*Competing interests*. The authors do not have any competing interests.

*Acknowledgements*. Sections 3.2–3.4 of this article rely to a great extent on recent joint work with Guillermo Artana, Gisela D. Charó, Mickaël Chekroun and Christophe Letellier (Charó et al., 2019, 2021a; Charó et al., 2021b; Charó et al., 2022a; Charó et al., 2022b). It is a pleasure to thank them for their respective inputs into the work published in these cited articles and the valuable discussions concerning this line of work that are continuing. We are grateful for the comments by Valerio Lembo and Paul Pukite, as well as the two anonymous reviews and the unpublished correspondence with Joshua Dorrington that have helped improve the original version of this review article. The article is TiPES Contribution No. 218; this project has received funding from the European Union's Horizon 2020 research and innovation program under Grant Agreement No. 820970 (M.G.).

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
