# Peer review of "Review Article: Dynamical Systems, Algebraic Topology, and the Climate Sciences"

_EGUsphere, 2023_

## Author Comment (AC4)

**Reply to RC2 - egusphere-2023-216**

[The reviewer's input is in italic font, while our responses are in regular font.]

*It has been a great opportunity for me to read this review article. The two authors have made a great attempt to bring together – to my knowledge for the first time in a kind of systematic manner – the recent developments regarding applications of the two commonly rather disparate fields of dynamical system theory and algebraic topology in the context of climatology. Both authors are well known for their enormous work on both topics over the last decades, so it is not surprising that the resulting manuscript presents great educational material on both topics.*

Thank you very much for this supportive and encouraging review.

*There is practically only one, very minor general comment that I may raise regarding this impressive work, and I need to say that this is a very subjective one based on my own knowledge of both fields, which is far from complete. Eventually, this paper may benefit (although I am not sure if that would indeed make up for an improvement) if the authors could highlight the specific potentials for further integration of both mathematical "disciplines" for future climate (or climate-related) studies even a bit more prominently. There might be a kind of "grey zone" between a review and perspectives paper. The way the material is presented is maybe not a classical review in the sense of attempting a complete coverage of the addressed field(s), which I feel absolutely comfortable with when reading this work. In such situation, I may envision this manuscript to become even more impactful when outlining somewhat more transparently the authors' perspective on a possible future research agenda, or at least parts of it. Please take this just as a suggestion, not a definite request of mine.*

We agree that there is a grey zone between a review and perspectives paper, and we will add a few paragraphs to improve the paper in its addressing future perspectives.

*All other comments I may have regarding certain parts of the manuscript are rather specific and/or technical and listed below:*

**Specific comments/suggestions**

*- p.3, ll.68-71: The authors essentially mention codimension-1 bifurcations (is "transverse" the same as "transcritical", the term that I am familiar with?); it might be useful to mention (here or later) the existence (and possible relevance for climate problems?) of bifurcations of codimension 2 or higher.*

Thank you for catching this typo. In fact, "transcritical" is a particular kind of codimension-1 bifurcation, in which a single solution branch preserves its existence but changes its stability across the bifurcation point: stable on one side of it and unstable on the other. "Transverse" refers to a particular condition for the intersection of two manifolds. Actually, we meant "transcritical" and will correct this typo, inherited from the Ghil et al (1991) paper.

Thank you also for suggesting that we mention the existence and relevance of codimension-2 bifurcations — such as Shilnikov, which is actually mentioned later, in Sec. 3.1, or Bogdanov-Takens. We will add one or two paragraphs on this important matter in the revised text.

*- In general, Section 1.1 could benefit from a few more "tutorial" references on dynamical system theory. The choice of references in this initial overview appears partially a bit "Ghil-centric", which is fine for the more specific discussions in Section 2 focusing on contributions of the first author.*

Good point, thank you. There is a rich literature on autonomous dynamical systems, in both mathematical and physical texts, although less so in the climate sciences. The main references in the latter field of applications, due to Henk Dijkstra aside from Ghil, have already been mentioned, as has a considerable amount of NDS and RDS literature in Secs. 2 and 4. But we will be happy to help the reader even more in citing additional good references.

*- p.4, ll.94-95: "This complementary view of the way that dynamics and topology interact is a main motivation of the present article." I fundamentally agree with the authors' emphasis on this point. There is a lot of algebraic topology tools in classical as well as modern dynamical system theory. One recent field that seems to provide another link between the two topics, which has also found vast applications in climate science in recent years, would be complex networks. I would leave it to the authors' choice whether or not to elaborate a bit (maybe one brief paragraph in the end) on corresponding recent developments and their potentials. The authors mention this very briefly on p.43, ll. 993-995, which emphasis on a rather specific problem, but I think there might be more to that.*

We acknowledge the impressive work being pursued in complex networks and their relevance in time series analysis [Zou et al, Physics Reports, Volume 787, 2019, Pages 1-97]. Algebraic topology is not mentioned in this review, but there have

been some papers applying persistent homologies to complex networks [Horak et al J. Stat. Mech. (2009) P03034; Petri et al (2013). PloS one, 8(6), e66506; De Silva & Ghrist (2007). Algebraic & Geometric Topology, 7(1), 339-358].

Horak et al. construct simplicial complexes from graphs (networks) to evaluate the robustness of a network and to distinguish different network types. Petri et al. use persistent homologies to detect particular nonlocal structures, akin to weighted holes within the link-weight network fabric. De Silva & Ghrist propose a method using persistent homologies for nonlocalized sensor networks with ad hoc wireless communications.

Despite the rapid development of computational topology and data science, the triple combination between algebraic topology, time series analysis and complex networks seems to be untouched so far. The network approach is used, however, to reconstruct the phase space, which is a prior and certainly necessary step for the analysis of the topological structure of flows from data.

The prospective directions in the field of complex networks enumerated by Zou et al [2019] share many of the challenges that are also faced by the topology of chaos. We are grateful to the reviewer for having directed our attention to this point and will expand upon it in the revision.

*- Somewhat related to the previous point, I might also suggest mentioning the framework of persistent homology and its potential applications, maybe even at the end of the introduction section (it appears only be briefly mentioned on p.24, l.559, before quickly focusing on the BraMAH methodology in the following).*

We will follow the reviewer's suggestion in this as well.

*- Figure 1: Since it seems not to be further explained, I would find it helpful (for non-experts) if the term "isopleth" could be briefly explained.*

The etymology of "isopleth" combines "iso" plus the ancient Greek word plêthos, "a great number". It is generically used to refer to a curve of points sharing the same value of some quantity. In his 1963 paper, Edward N. Lorenz plots the isopleths (isolines) of $X$ as a function of $Y$ and $Z$ of his attractor, to approximate surfaces formed by all points on limiting trajectories. We will add this explanation to our manuscript.

*- p.8, ll.194-195: I recommend adding a brief clarification that any bifurcation presents a kind of tipping ("B-tipping" in the language nowadays used by many authors, probably going back to Ditlivsen and co-workers?), while not every tipping behaviour in a complex system originates from an underlying bifurcation.*

There are at least two different interpretations of "tipping" and "tipping points" in the literature. One of these, emanating from Gladwell (2000) and Lenton et al (2008), interprets tipping merely as a sudden change, whether due to a well-defined bifurcation or not. In this interpretation, a tipping point is merely a threshold.

The other interpretation sees a tipping point as a generalization to nonautonomous systems of a bifurcation point (Ghil, 2019; Kuehn, C., 2011. A mathematical framework for critical transitions: Bifurcations, fast–slow systems and stochastic dynamics. Physica D, 240(12), 1020–1035). In this case, tipping is necessarily related to a tipping point in phase-parameter space and not every jump or critical transition arises from a such a point.

We will clarify this in the revised version since both points of view have their merits, but confusion should be avoided to the extent possible.

*- p.11, ll.272-273: In fact, the orbital cycles emerge from chaotic motion, but have contributions with relatively narrow (yet not exactly fixed) frequency and strongly varying amplitudes. So in reality, one would not assume that 19 kyr and 41 kyr variability components are exactly fixed (and, hence, would not have a simple integer ratio), but may lead to more complex dynamical phase locking-unlocking processes. This is far beyond the scope of the present work, but maybe the specific sentence here could be a bit reshaped to clarify what the authors actually attempt to focus on.*

We are not exactly sure whether the reviewer refers to work on the presence of chaos in the planetary system or not (e.g., Varadi, F., M. Ghil, and W. M. Kaula, 1999: Jupiter, Saturn and the edge of chaos, Icarus, 139, 286–294). We will try to clarify this point further, too.

*- In Section 2.1, state vectors are denoted in bold face. In Section 2.2, however, vectorial quantities are not written in bold face anymore (e.g. x in Eq. (9), l.307 and following). I strongly suggest revising the appearance of mathematical terms for self-consistency between the different (sub)sections.*

Thank you for this correction. We will use boldface vectors throughout.

*- p.13, l.320: Please clarify that this refers to chaotic trajectories in the Lagrangian chaos sense, not chaos of the underlying field g(t,x) itself.*

Will be clarified, thank you.

*- p.13, l.325: Please explain the term "pullback attraction" in a few lay words.*

Good idea, thank you. We will use simple language, like "A pullback attractor is a possibly time-dependent object in a system's phase space that exhibits attraction in the sense of convergence at each time $t$ to a set, called a snapshot, to which the system's initial state at time $s$ tends to as $s$ tends to $-\infty$. This is distinct from the forward attractors that can be defined for autonomous systems started at a fixed time $t_0$."

*- p.15, l.375: There is no $\beta$ in Eq. (16), only $\sigma$. Is that one meant here?*

Yes, thank you for the correction.

*- Section 2.2.3: The symbol $\omega$ used here has been previously used for a frequency in Section 2.2.2. Please consider using a different symbol.*

Thank you for noticing. We will use two different symbols in Secs. 2.2.2 and 2.2.3 for a frequency and for a realization of the driving noise, respectively.

*- p.18, ll.434-437: Could you add a brief note on suitable "mathematical" types of noise distributions?*

We will define more precisely the connection between Brownian motion $d\eta$ and a Wiener process $W$, including a reference.

*- p.20, Fig. 8: The figure caption refers to a color bar that is missing in the figure.*

We will explain the color bar in words in the caption.

*- p.23, l.517: The term "braids" might be unfamiliar to many readers, so a brief explanation might be helpful. Similar for p.29, l.671 ("templexes", only introduced in the following subsection) and p.31, l.714 ("knot-holder").*

We can imagine a knot as a thin tangled rope in three-dimensional space whose ends are glued together [Prasolov et al. Knots, links, braids and 3-manifolds: an introduction to the new invariants in low-dimensional topology. No. 154. Amer. Math. Soc., 1997.], while a braid is a collection of strands crossing over or under each other. Both concepts became central around 1987 in the attempt to classify low dimensional (3-D) systems using topological orbit organization.

The knot approach — i.e., extracting the knot content of hyperbolic attractors — is rooted in results from Birman-Williams-Holmes, through a geometrical construction that was named template or knot-holder. The braid approach is based on results due to Thurston on the classification of 2-D diffeomorphisms and the braid content of the diffeomorphism [Natiello, M. A., 2007. The User's Approach to Topological Methods in 3d Dynamical Systems. World Scientific].

*- p.41, ll.918-926: The discussion on "wave-like" vs. "particle-like" behaviour (drawing upon a quantum mechanical analogy) reminds me a bit of the traffic jam analogy of atmospheric blocking situations by Nakamura and Huang (Science, 361 (6397), 42-47, 2018). I would wonder if the authors would see some link to this very active field of studies in climate science (persistent atmospheric wave trains, blocking, and extreme events) from their more fundamental "mathematical" (conceptual) perspectives.*

Well, the authors are familiar with the very interesting Nakamura and Huang (2018) paper and one of them is working in a separate collaboration on applying extremal length theory (Ahlfors, L., 1973. Conformal Invariants: Topics in Geometric Function Theory, American Mathematical Society) to midlatitude flow diagnostics. But the paper at hand is getting quite long and persistent anomalies have been a "very active field of studies in climate science" for over four decades, so adding yet another approach to Fig. 22 does not seem imperative at this stage. Thank you, though, for finding inspiration in our quantum mechanical analogy.

**Technical suggestions:**

*- p.1, l.3: "in the 1960s"?*

Yes, at least in the U.S. spelling (Chicago Manual of Style), which is used throughout this paper, that is correct, and not the more common "1960's." After the first mention, the '60s and '70s are correct, too.

*- p.4, l.110: typo "current"*
*- p.6, l.141: typo "the way"*
*- p.6, l.142: citation style of Williams (1974) should be adjusted*

Right, thanks.

*- p.6, l.150: "in the 1990s"? (this would be consistent with l.3 and others... )*
*- p.6, l.160: "in the 1960s and early 1970s"*
*- p.8, ll.188 and 190: I suggest replacing "in the next subsection" by "in Section 2.1.1"*
*- p.8, l.211: It might be too much of a request, but citing some original work by Maxwell might be a quite unique thing for a paper in this journal.*

Given the early Poincaré references, adding James Clark Maxwell or Pierre Curie would seem quite appropriate. Thank you for sharing our taste for early citations.

*- p.9, l.212; p.10, l.220; p.11, ll.258-259: citation style of Ghil and Childress (1987) should be adjusted*

Thanks for noticing: this will be done, of course.

*- p.11, l.257: replace "in the above figure" by "in Fig. 4"*

Thank you, will be done.

*- p.11, l.262: "periodicity of glacial cycles"*

Not sure what exactly this refers to: "glacial cycles" suggest a unique periodicity, "climatic variability" does not. We beg to differ.

*- p.13, l.319: the condition should read "$t_0 \leq t_1 \leq t_2$" (subscript 1 is missing)*

Correct, thanks for noticing; will be fixed.

*- p.13, Definition B.1 (why are definitions enumerated as B.1, B.2, etc.?): You introduce the set $\mathbb{R}_+^2$, but refer to $\mathbb{R}_\geq^2$ in the definition of the mappings.*

The B's will be removed; thanks for noticing their being redundant herein. The difference of notation for the nonnegative real numbers is a typo and will be fixed.

*- p.14, l.341: "translation in time"*
*- p.14, l.344: I would start the sentence with "As an example, analytical computations. . . "*
*- p.14, l.353: I am not sure if the abbreviation PBA had been defined before; I would suggest to avoid it.*
*- p.15, Eq. (17): use large brackets*
*- p.16, Eqs. (22) and (23) (and also a bit of the text in the remainder of Section 2.2.2 mixes up the symbols $\phi$ and $\varphi$. Please keep consistency.*
*- p.22, l.497: replace "While. . . " with "By contrast. . . " or something similar*
*- p.27, ll.638-639: Why is the set of ODEs (26a,26b) infinite?*
*- p.28, l.649: The solution of the Navier-Stokes equations would be a velocity field, not a streamfunction. Better write: ". . . the streamfunction. . .  would not correspond to a solution. . . "*
*- p.32, l.725: "distinct chaotic attractors"*
*- p.38, l.852: type "known"*
*- p.40, l.886: In my understanding, Betti numbers are integers, so any change in this property must be necessarily discontinuous. Or do the authors want to emphasize something different here?*

Yes, they are integers: the phrase "can be quite sudden" will be changed to "is quite sudden, since they are integers."

*- p.42, l.943: "Pacific North America (PNA) pattern"*
*- p.43, l.964: omit abbreviation "TDA"*
*- p.43, l.971: ". . . and provided further. . . "*
*- p.45, l.1038: volume missing in Carlsson & Zomorodian (2007)*
*- p.46, l.1060: remove "20 pp."*
*- p.47, ll.1081-1082: this seems to be a duplicate reference*
*- p.48, l.1123: remove "41 pages"*
*- p.48, ll.1132-1133: if this is a book chapter, add page numbers; otherwise if this is the full book title, adjust citation style accordingly*
*- p.49, l.1168: there is something odd with the style of this citation, please check/correct*

There is a blank space missing between "Basis." and "Contribution"; will be fixed.

*- p.51, ll.1232 and 1262: please give full (non-abbreviated) journal names*
*- p.52, l.1295: update reference with volume and page numbers, or provide doi if still only "online first"*
*- p.52, ll.1204-1306: this seems to be another duplicate reference*
*- p.53, l.1307: replace page numbers by proper article ID*
*- p.53, l.1327: volume missing*

All the items that are not specifically addressed in our replies will be taken care of as well. Thank you for your extremely careful reading and very constructive comments.

---

## Author Response (AR1)

**Replies to Comments and Reviews - egusphere-2023-216**

**Reply to CC1**

[The comments are in italic font, while our responses are in regular font.]

*Thank you for this review article for it's mix of perspectives on autonomous versus non-autonomous systems, and of natural versus forced (pullback attractors) responses. It's known that non-linear otherwise chaotic systems can also deterministically follow the forcing applied. This is where the forced response overrides the natural response. Doesn't mean that it's easy to figure out what the response is (based partly on "hawkmoth" structural uncertainty ), but like other forced responses, the dependence on initial conditions becomes irrelevant once it synchronizes with the forcing applied. This means that there may be hope in predicting dynamical climate once the patterns of forcing and responses are better understood.*

*The excerpt attached from "Synchronization in Oscillatory Networks", Osipov et al (Springer, 2007)*

*Upload of image excerpt didn't work in the last reply comment so here is a link to the image https://imagizer.imageshack.com/img923/3113/FbOxei.png*

Thank you, Dr. Pukite, for your interesting comment on our review paper. Your comment raises two questions: (i) that of structural uncertainty or instability, sometimes labeled the "hawkmoth effect"; and (ii) that of the connection between the theory of nonautonomous and random dynamical systems (NDSs and RDSs) and synchronization theory (e.g., Osipov et al., 2007; Duane et al., 2017).

(i) Structural stability refers to the stability of a system's behavior under perturbations of its parameters or, more generally, of its governing equations. It is distinct from and complementary to the usual stability of steady states, in particular, or, more generally, to that of other invariant solutions, such as periodic (limit cycles) or quasi-periodic (tori) ones to perturbations in initial conditions. Hence the term hawkmoth effect for the former, which parallels the term "butterfly effect" for the latter. Ghil (1976) referred to these two types of stability as external vs. internal.

Structural stability was introduced into dynamical systems theory by Andronov and Pontryagin (1937) and it is by now well understood for autonomous dynamical systems (Arnold, 1983; Guckenheimer and Holmes, 1983). Similar results are

available for NDSs and RDSs (e.g., Caraballo and Han, 2017; Kloeden and Rasmussen, 2011). Our review article is already quite long and we could not, at this stage, add material on this important but somewhat technical topic. The concept of pullback attraction — as opposed to the forward attraction of autonomous systems, which we do present and discuss in our paper — plays a key role in the NDS and RDS case of structural stability, too.

(ii) Synchronization, in its simplest form, is a particular manifestation of an oscillator's frequency becoming entrained by that of a forcing. A well-known example is that of circadian rhythms in humans and other animals, as well as in plants (Winfree, 1980). Ghil and Childress (1987/2012, Ch. 12) have discussed the more general case of quasi-periodic forcing of a climatic oscillator during the Quaternary glaciation cycles. Riechers et al. (2022) have considered this case from the point of view of NDS theory.

Mutual synchronization between two or more oscillators is another form of this widespread phenomenon in the physical and life sciences. More recently, the limit of this case to large networks and continuous media has been actively considered (Duane et al., 2017, and references therein). It would be of substantial interest to study the connection between synchronization in this limit and NDS theory. Again, such considerations go well beyond what's feasible in the present paper.

**References**

1. Andronov, A. A., and L. Pontryagin. Systémes grossiers. Dokl. Akad. Nauk. SSSR, 14, 247–251, 1937.
2. Arnold, V. I., Geometrical Methods in the Theory of Ordinary Differential Equations. Springer, New York, 1983.
3. Caraballo, T. and Han, X., Applied Nonautonomous and Random Dynamical Systems: Applied Dynamical Systems. Springer, 2017.
4. Duane, G., C. Grabow, F. Selten, and M. Ghil, Introduction to focus issue: Synchronization in large networks and continuous media—data, models, and supermodels', Chaos, 27, 126601 (9 pp.), doi: 10.1063/1.5018728 , 2017.
5. Ghil, M.: Climate stability for a Sellers-type model, J. Atmos. Sci., 33, 3–20, 1976.
6. Ghil, M., and S. Childress, Topics in Geophysical Fluid Dynamics: Atmospheric Dynamics, Dynamo Theory and Climate Dynamics, Springer Science & Business Media, doi:10.1007/978-1-4612-1052-8, ISBN 978-F0-387-96475-l, pp. xv + 485. 1987; reissued as an eBook by Springer, ISBN 978-1-4612-1052-8, 2012.

7. Kloeden, P. E., and M. Rasmussen. Nonautonomous Dynamical Systems. American Mathematical Society, 2011.

8. Osipov, G.V., Kurths, J. and Zhou, C., Synchronization in Oscillatory Networks. Springer Science & Business Media, 2007.

9. Riechers, K., T. Mitsui, N. Boers, and M. Ghil, Orbital insolation variations, intrinsic climate variability, and Quaternary glaciations, Clim. Past, 18(1), 863–893, doi:10.5194/cp-18-863-2022, 2022.

10. Winfree, A. T., 1980: The Geometry of Biological Time, Springer, New York, 530 pp.

**Reply to RC1**

[The reviewer's comments are in italic font, while our responses are in regular font.]

*In this very interesting and stimulating review, the authors provide the readers with an introduction to how algebraic topology can cast insight into the behavior of dynamical systems, after recalling how dynamical system theory is very relevant to geophysical models in general and to climate models in particular (it is quite telling that Henri Poincaré took the example of weather as an example of chaos).*

Thank you very much for this insightful and supportive review.

*The algebraic topology approach outlined in the review is itself based on a topological approach whose defining concepts (e.g., branched manifolds) were laid out by Birman and Williams and whose application to the natural sciences was pioneered by Gilmore and co-workers. The exposition follows a historical perspective where the identification of branched manifolds through cell complexes and homologies characterizing these complexes is first recalled before presenting the latest developments in the fields (templexes and stripexes) which not only take into account the skeleton on the attractor, but how the flow explores it by inferring the underlying semi-flow. Fundamental nonlinear phenomena such as bistability, appearance of self-sustained oscillations through a Hopf bifurcation are clearly explained.*

*An interesting feature of the review is how it integrates different aspects very relevant to geophysical and climate applications : noise, variability due to non-stationarity, eulerian vs lagrangian description which will make the review very useful for the readers of Nonlinear Processes in Geophysics. How these aspects increase the complexity of characterizing geophysical systems is well illustrated.*

*The paper is quite succcessful in convincing the readers that the approaches advocated have very promising perspectives to tackle the challenges presented by of geophysical problems.*

*I found the perspective section very rich and interesting.*

*Thus, I strongly recommend this manuscript for publication in NPG, given that the following minor remarks are taken into account by the authors.*

Your strong recommendation is much appreciated, and the authors have done their very best to take into account your remarks and thoughtful suggestions.

*- in the introduction, it would be nice to shortly discuss a geophysical model of interest to make the discussion more illustrative.*

Unfortunately, there is no canonical geophysical model on which all the concepts and methods discussed in this review could be illustrated. But we now give separate examples for all the major ones.

*- around Figure 1 and line 135, the authors allude to the stretching, squeezing, folding etc. mechanisms that build a chaotic attractor, but in my opinion, they do not provide sufficient information for an uninformed reader to grasp what these mechanisms are nor do they explain how and why the topological approach is an elegant and natural way to capture these mechanisms.*

*- they introduced branched manifolds without describing them very much nor giving a precise definition of them. Recall that branched manifolds are obtained by identifying points along a given segment of the stable manifold, so that it is a kind of projection. How many stable directions must be taken into account will matter very much, in particular. This is important to understand what is the semi-flow that the authors invoke.*

Answers to the two previous comments:

In "How topology came to chaos," Gilmore (2013) explains that metric and dynamical invariants do not provide a way to distinguish among the different types of chaotic attractors and that a tool of a different nature was needed to create a dictionary of processes and mechanisms underlying a chaotic system.

"Listening more closely to Poincaré, it was clear that this new tool ought to involve the periodic orbits 'in' a chaotic attractor. A chaotic trajectory winds around in

phase space arbitrarily close to any unstable periodic orbit, so it ought to be possible to use segments of a chaotic trajectory as good approximations (surrogates) for UPOs. [...] It was clear that UPOs could also serve as the skeleton of the strange attractor."

While Gilmore, Lefranc and co-workers were "mulling over implementing a program based on building tables of linking numbers and/or relative rotation rates between trajectories, a better solution became available. Joan Birman and Robert Williams had shown that the dissipative nature of a flow in phase space allows projecting the points along the direction of the stable manifold by identifying all the points with the same future."

"Suppose we have a dissipative chaotic flow in three dimensions: there are three Lyapunov exponents (for the unstable direction, for the flow direction and for the stable direction). The dissipative nature of the flow requires . Then it is possible to project points in the phase space down in the direction of the stable manifold. This is done by identifying all the points with the same future:

$$x \sim y \text{ iff } \lim_{t \to +\infty} |x(t) - y(t)| = 0,$$

where $x(t)$ is the future in phase space of the point $x = x(0)$ under the flow. This Birman-Williams identification effectively projects the flow down to a manifold almost everywhere, except at the points where the flow splits into branches heading towards distinct parts of phase space, or at the points where two branches are squeezed together. These mathematical structures were called branched manifolds."

A branched manifold can in fact be defined mathematically without reference to a flow, or to the Birman-Williams projection mentioned above

**Definition** (from Kinsey page 64). An $n$-dimensional manifold is a topological space such that every point has a neighborhood topologically equivalent to an n-dimensional open disc with center $x$ and radius $r$. Such a manifold is said to be Hausdorff iff any two distinct points have disjoint neighborhoods.

The second condition is not satisfied precisely at the junction between branches, i.e., at the locations that describe stretching and squeezing of a flow in phase space. A branched manifold is therefore a manifold that is not required to fulfill the Hausdorff property.

We prefer this more general definition, instead of the one related to the Birman-

Williams projection, for several reasons, including the possibility of extending the concept of branched manifold to the structure of instantaneous snapshots of random attractors. This mathematical definition of a branched manifold will also let us extend the procedure to cases in which the hypotheses of the Birman-Williams theorem – in which the dynamical system must be hyperbolic, three-dimensional, and dissipative – are not valid. In most geoscientific applications, for instance, uniform hyperbolicity does not apply.

As the topological structure of a branched manifold is closely related to the stretching and squeezing mechanisms that constitute the fingerprint of a certain chaotic attractor, its properties can be used to distinguish among different attractors. This is how the two-way correspondence between topology and dynamics can be justified. This correspondence remains valid in the case of four-dimensional semi-conservative systems [Charó et al, 2019; Charó et al, JFM, 2021], for which the hypotheses of the Birman-Williams theorem do not hold.

The terms "branched manifold" and "template" have often been used interchangeably. We do not consider them as synonyms, for technical reasons that will be important in the development of the concept of templex. A branched manifold is just a particular type of manifold that can be reconstructed from a set of points in Rn, by approximating subsets of points by cells, which are glued to form a cell complex. The dimension of the cell complex d coincides, by construction, with the local dimension of the branched manifold approximating the point cloud, but there is no restriction in the value of n or of d. Both values are computed directly from the dataset, using successive singular value decompositions. The number of eigenvalues scaling linearly with the number of points grouped in a cell provide the value of d for that cell, and this computation is done on matrices that contain the n coordinates of the points, without performing projections of any kind. These computations construct a cell complex from the point cloud without involving the flow. The information carried by the flow is not contained in the cell complex but will be contained in the digraph of the templex.

We have incorporated these clarifications into the text of our paper; please see the paper's attached latexdiff between the original submission and the revised version submitted herewith.

*- it is not entirely correct to write that systems whose branched manifolds are topological equivalent are dynamically equivalent. Their orbit content, or the associated symbolic dynamics could differ. But it is true that they cannot be equivalent if the branched manifolds differ.*

On page 175 of "How topology came to chaos," we read:

"Branched manifolds are useful constructions for distinguishing among different mechanisms that generate strange attractors. Topological equivalence between branched manifolds is by isotopy. Two things are isotopic if it is possible to mold one into the other without tearing or gluing it. As a result, identifying the branched manifold that describes a strange attractor is a powerful tool for distinguishing one (class of) strange attractors from the other."

It is in this sense that we speak of dynamical equivalence, while the metric or dynamical invariants describing the orbit content are not being considered. These points are now clarifyied in the text; please see attached latexdiff.

*- In definition B.1, is $R_>^2 =$ or $R_+^2$ ?*

Thank you for noticing the typo in the second line of the definition. Both notations are used but they should not be mixed up. The single $R_>^2 =$ in the definition has been replaced by $R_+^2$.

*- I find the discussion about pull-back attractors (please define PBA when it first appears !) interesting however I missed the motivation for introducing the concept, which became clearer later in the text. Please explain from the beginning why the concept is interesting and useful. It seems to me that it is particularly relevant for noisy systems, am I right ?*

Good point. We have now defined the PBA acronym the first time it appears, on p. 13, l. 328, and have paid even more attention to motivation, although several concepts are introduced to a broader audience and the best sequence in which to do this is not always obvious. The pullback concept is relevant for all systems that have time-dependent forcing, whether stochastic or deterministic. This point is now mentioned in the Abstract, p. 1, l. 14.

*- line 375, beta or sigma*

It should be sigma. Corrected!

*- I would distinguish between nonautonomous systems and non-stationary systems. The former can have some sort of regularity (e.g., periodic driving). The latter could experience any type of slow drift. It seems to me that the authors switch from one type of system to the other without prior notice.*

The definition of nonautonomous system that we are using is unambiguous and stated as the explicit appearance of time in the mathematical expression of the governing equations of the system under consideration. Stationarity is a property of solutions, not of the system of equations. An autonomous system can have stationary, periodic or chaotic solutions. A nonautonomous system cannot, as far as we can tell, have stationary solutions, unless the forcing tends to zero.

*- there is a strong link between topological chaos as studied by Thiffeault and Gouillart and what the authors call "topology of chaos". In a fluid, you have a flow taking particles from points in space to other points, and the same occurs in phase space. Chaos is due to mixing processes in the two situations.*

There is a strong link between the two situations, but the keywords refer to different motivations and objectives. Working with the topology of real fluid flow trajectories in physical space implies working in no more than three dimensions, for example. On the other hand, investigating fluid flows in phase space and in physical space is not equivalent. Physical space for fluid flows is most often a plane projection of a higher-dimensional state space.

*- it is not entirely true that periodic orbits approximate actual trajectories in the topological approach. Rather, it estimates the neighboring UPO which the flow is evolving around. That is the opposite perspective (l. 508).*

In the methodology that approximates trajectories with knots, trajectories must be closed into a knot that approximates the neighboring UPO around which the flow is evolving.

We have modified the sentence.

*- l. 660 I do not quite understand what the problem with the standard approach of considering time as a state variable for non-autonomous systems, I feel that the authors should elaborate.*

This discussion is central to the Charó et al. [2019] paper. When a dynamical system is governed by a set of equations where the time explicitly occurs, some processes involved in the dynamics are not explicitly described and the state space is not completely determined. In fact, one of the fundamental hypotheses in writing a dynamical system as a set of ODEs is that time is the only independent variable, while all state variables are time dependent. Making time play a double role – that of the only independent variable and that of a state variable – is misleading.

Working in a space whose dimension is increased by one due to introducing the extra ODE leads to certain difficulties in using the tools borrowed from nonlinear dynamical systems theory—for instance, the state space is no longer bounded. When time t is added as an extra coordinate to the phase space, we get an 'extended phase space'. In this extended phase space, a periodic orbit is no longer a closed curve, simply because when the system returns to the same state, it does not return to the same point. The very definition of phase space in which a point univocally represents a state of the system is no longer valid in the extended phase space. Many of the properties that are valid in a well-defined phase space are altered in an extended phase space, and topology is one of them.

In the case of the driven double gyre discussed by Charó et al. [2019], the starting point is a nonautonomous system of two ODEs. The extended phase space (with a third ODE written as $\dot{t} = 1$) is three-dimensional. But the paper shows that a fourth dimension is needed to rewrite the system as an autonomous set of ODEs without using the trick. The phase space of the autonomized driven double gyre has four ODEs: two additional variables are required, $u$ and $v$. Such a transformation gets rid of the explicit time dependence with a legitimate procedure that does not run into the previously explained inconsistency. In this four-dimensional phase space, and for certain initial conditions, the topological structure that is obtained is a Klein bottle. A Klein bottle cannot be immersed into a three-dimensional space without self-intersections: the role of the fourth dimension that is required to rewrite the system in an autonomous form is, therefore, highly relevant here.

Thus, to use topological tools self-consistently, one must be prepared to work in a well-defined phase space, and with as many dimensions as required. This point is now clarified in the text; see latexdiff.

*- the authors should mention other approaches to characterize dynamical chaos in arbitrary dimensions. In particular, the method proposed by Lefranc (Phys. Rev. E 035202, 2006) is based on a triangulation of periodic points that is very similar to cell complexes, and characterizes how facets of the triangulation are transformed between themselves, which is a description of the semi-flow. It only considers the problem of estimating the entropy of the flow, but this is also a challenge in geophysics. Similary, it would interesting to mention applications of the Conley index by Mischaikow and collaborators.*

*- at several places, there is a discussion of using UPO or not utilizing them, but it could be useful to mention that UPO can be very useful to characterize a chaotic system because the information about them can be obtained in a finite time, which can be useful in non-stationary systems and because a single UPO can bring much*

*information (see Amon and Lefranc, Phys. Rev. Lett. 094101, 2004).*

We have added these references to the text and explain how constructing a cell complex and computing its homology groups differs from the methodologies considered in them. Please see the attached latexdiff of the paper.

*I am convinced that the authors can easily address these minor issues.*

Thank you again for having raised these very useful points.

**References**

Gilmore, R.: How topology came to chaos, in: Topology and Dynamics of Chaos in Celebration of Robert Gilmore's 70th Birthday, edited 1275 by Letellier, C. and Gilmore, R., pp. 63–98, World Scientific Publishing, 2013.

**Reply to CC3**

[The reviewer's input is in italic font, while our responses are in regular font.]

*I appreciated the reading of manuscript "Dynamical Systems, Algebraic Topology and the Climate Sciences", a review article submitted for consideration on Nonlinear Processes in Geophysics by Michael Ghil and Denisse Sciamarella. The contribution is part of an invitation-only special issue, "Perspectives on Climate Sciences: from historical developments to research frontiers", meant to be a follow-up of a webinar series organized under the auspices of the European Geosciences Union between 2020 and 2021.*

*The review article is aimed at bringing together authors' work and views on recent developments in dynamical systems, especially non-autonomous ones, and the usage of features of branched manifolds defined in algebraic topology for the characterization of an attractor's behavior. The article is well written in most of its parts, and it is a very enjoyable reading, especially for those not familiar with the specific topics addressed. I wish to share here a few remarks, and also some views on how these ideas can be expanded and find applications in climate sciences.*

We are grateful for your careful reading and for helping us improve this manuscript.

**SPECIFIC COMMENTS**

*- ll. 146 and elsewhere: a non-expert introduction to branched manifolds is missing, and I think it would be very welcome, given the potentially broad audience to whom the contribution is directed. Not only a mathematical definition of these objects (some idea of that is given at ll. 537-540), but rather expanding on the potential advantage of adopting this approach in the field of dynamical systems would be maybe helpful;*

This point has been also raised by the first reviewer. An introduction to the concept of branched manifold is given below. We also comment on how this definition has been tailored to suit the different aspects of the approach that we present.

In "How topology came to chaos," Gilmore (2013) explains that metric and dynamical invariants do not provide a way to distinguish among the different types of chaotic attractors and that a tool of a different nature is needed to create a dictionary of processes and mechanisms underlying a chaotic system.

"Listening more closely to Poincaré, it was clear that this new tool ought to involve the periodic orbits 'in' a chaotic attractor. A chaotic trajectory winds around in phase space arbitrarily close to any unstable periodic orbit, so it ought to be possible to use segments of a chaotic trajectory as good approximations (surrogates) for UPOs. [...] It was clear that UPOs could also serve as the skeleton of the strange attractor."

While Gilmore, Lefranc and co-workers were "mulling over implementing a program based on building tables of linking numbers and/or relative rotation rates between trajectories, a better solution became available. Joan Birman and Robert Williams had shown that the dissipative nature of a flow in phase space allows projecting the points along the direction of the stable manifold by identifying all the points with the same future."

"Suppose we have a dissipative chaotic flow in three dimensions: there are three Lyapunov exponents ($\lambda_1 > 0$ for the unstable direction, $\lambda_2 = 0$ for the flow direction and $\lambda_3 < 0$ for the stable direction). The dissipative nature of the flow requires $\lambda_1 + \lambda_2 + \lambda_3 = 0$. Then it is possible to project points in the phase space down in the direction of the stable manifold. This is done by identifying all the points with the same future via the relation

$$x \sim y \ \text{ iff } \lim_{t \to +\infty} |x(t) - y(t)| = 0,$$

where $x(t)$ is the future in phase space of the point $x = x(0)$ under the flow. This Birman-Williams identification effectively projects the flow down to a manifold almost everywhere, except at the points where the flow splits into branches heading towards distinct parts of phase space, or at the points where two branches are squeezed together. These mathematical structures were called branched manifolds."

A branched manifold can in fact be defined mathematically without reference to a flow, or to the Birman-Williams projection mentioned above. Definition (from Kinsey page 64). An $n$-dimensional manifold is a topological space such that every point has a neighborhood topologically equivalent to an n- dimensional open disc with center $x$ and radius $r$. Such a manifold is said to be Hausdorff if and only if any two distinct points have disjoint neighborhoods.

The second condition is not satisfied precisely at the junction between branches, i.e., at the locations that describe stretching and squeezing of a flow in phase space. A branched manifold is therefore a manifold that is not required to fulfill the Hausdorff property.

We prefer this more general definition, instead of the one related to the Birman-Williams projection, for several reasons, including the possibility of extending the concept of branched manifold to the structure of instantaneous snapshots of random attractors. This mathematical definition of a branched manifold will also let us extend the procedure to cases in which the hypotheses of the Birman-Williams theorem – in which the dynamical system must be hyperbolic, three-dimensional, and dissipative – are not valid. In most geoscientific applications, for instance, uniform hyperbolicity does not apply.

As the topological structure of a branched manifold is closely related to the stretching and squeezing mechanisms that constitute the fingerprint of a certain chaotic attractor, its properties can be used to distinguish among different attractors. This is how the two-way correspondence between topology and dynamics can be justified. This correspondence remains valid in the case of four-dimensional semi-conservative systems [Charó et al, 2019; Charó et al, JFM, 2021], for which the hypotheses of the Birman-Williams theorem do not hold.

The terms "branched manifold" and "template" have often been used interchangeably. We do not consider them as synonyms, for technical reasons that become important in the development of the concept of templex. A branched manifold is just a particular type of manifold that can be reconstructed from a set of points in $\mathbb{R}^n$, by approximating subsets of points by cells, which are glued to form a cell

complex. The dimension $d$ of the cell complex coincides, by construction, with the local dimension of the branched manifold approximating the point cloud, but there is no restriction on the value of either $n$ or $d$. Both values are computed directly from the dataset, using successive singular value decompositions.

Since the number of eigenvalues scales linearly with the number of points grouped in a cell, this number provides the value of $d$ for the given cell, and this computation is carried out on matrices that contain the $n$ coordinates of the points, without performing projections of any kind. These computations construct a cell complex from the point cloud without involving the flow. The information carried by the flow is not contained in the cell complex but will be contained in the digraph of the templex.

Please see latexdiff of the paper for the corresponding text; blue paras. at bottom of p. 31 and top of p. 32.

*- Figure 2: the label is not very self-explanatory and should be better detailed;*

These three-dimensional point clouds are obtained by integrating the Lorenz equations using coordinate transformations for some of the variables. The butterfly is deformed but the topological structure of the butterfly is maintained. The caption has been expanded.

*- ll. 366-371: This paragraph seems to be missing a take-home message;*

Good point, thank you. The paragraph now reads:

"The finite-dimensional definition above follows Charó et al. (2021b, Appendix A and references therein). In fact, both deterministic and stochastic versions of [time-dependent] forcing have been applied, for instance, by Chekroun et al. (2018) in the study of an infinite-dimensional, delay-differential equation model of the El Niño–Southern Oscillation (ENSO). The deterministic forcing corresponded to the purely periodic, seasonal changes in insolation, while the stochastic component represented the westerly wind bursts appearing in various ENSO models by F.-F. Jin and A. Timmermann (e.g., Timmermann and Jin, 2002); see also Chekroun et al. (2011, Sec. 4.3). This ENSO example, among many others, shows that there is great flexibility in the application of the concepts and methods of nonautonomous dynamical systems (NDS and RDS) theory to climate problems."

The take-home message is in the paragraph's last sentence.

*- Figure 8: the label here is also a bit ambiguous, as the invariant measure nu is not defined anywhere in the text;*

A full definition of invariant measures would occupy too much additional space in an already rather long review paper. A simple definition in lay words is given in the discussion of the heat map of Fig. 6b, ll. 443–448, along with suitable references. A similar effort has been made in the caption of Fig. 8 on p. 21.

*- ll. 569-619: the authors present here an extensive list of possible applications of the BraMAH approach, but this has not been yet described in the manuscript. I think this part shall be significantly reduced;*

We beg to differ. In fact, the comments of the second solicited reviewer, RC2, request us to expand the discussion on future perspectives. We prefer to listen to the advice in RC2.

*- l. 671: the authors imply that the method has been adopted and described in Sect. 3.1, but it is not the case (see my point above);*

Line 671, which is part of Sec. 3.2, is a bit confusing, since BraMAH for autonomous systems was, in fact, described in the preceding Sec. 3.1 but the templex is described in the subsequent Sec. 3.3. The sentence has been modified to clarify this point.

*- l. 681: it would be nice to see how these classes emerge in the DDG model;*

These clouds are obtained by integrating Shadden's ordinary differential equations from different initial conditions. Further details have been provided in the text to save the reader the trouble of having to refer to the source article. Please see latexdiff.

*- Figure 13: not clear to me what the colors refer to here, as the label refers to colors that do not appear to be present in the figure;*

*- ll. 683-689: are the authors discussing Figure 13 or 14 here;*

Information and clarifications have been added to the captions of Figs. 12, 13 & 14 and to the text discussing them, to make this review article as self-contained as possible; please see the attached latexdiff. Thank you for these remarks.

*- ll. 727-728: to this point, it is not clear to me what a "strip" is in a topological sense. Given that a reader might not be familiar with algebraic topology, I think that some qualitative description might be provided here or elsewhere;*

Thank you for this remark, the definition of strip is part of the template terminology. We have added this explanation to the text. Please see the last full para. on p. 35 of the latexdiff.

**TECHNICAL CORRECTIONS**

*- l. 141: "they" → "the"; - l. 187: "eingenvalues" → "eigenvalues";*

Done, thank you.

**OUTCOMES**

*Overall, I think that the approach of random temples on random attractors is very promising, and I see potential applications on several aspects that are of interest for climate sciences. I list here a number of possible topics to develop:*

- *Design of optimal ensembles for climate predictions: given that in the range of problems related to climate prediction we are not constrained about initial conditions as in NWP, but even if we are in a genuinely non-autonomous dynamical system with a possibly random forcing, we are reasonably confident that the evolution of the attractor will preserve its homologies. This said, an efficient mapping of the ensemble initial conditions on the cloud of trajectories around the fixed point of the attractor, selected according to their homological properties, might help increasing the reliability of ensemble prediction with a reduced number of members. This is well inside the scope of reconciling the different flavors and approaches to Low Frequency Variability, as outlined in the text;*

- *Investigation of precursors of tipping points: given the rigorous definition of "topological tipping points", it would make sense, as outlined in the concluding remarks, to discuss to what extent these tipping points are representative of tipping points in a climatic sense. In order to do so, idealized conceptual models of key tipping elements might be useful tools, as they*

*would bear a relatively known attractor, at the same time allowing to explain the physics behind the described processes and to identify precursors of critical transitions;*

We are grateful to the reviewer for raising these two broader points. Both of them are related to the information provided by random templexes computed for random attractors in the investigation of the actual climate system and of its prediction. We have added a few sentences that might withstand the test of time. Particular attention has been given to the possible connections between TTPs, on the one hand, and the better understood tipping points associated with the description and prediction of the climate system by differential systems rather than by homologies, on the other.

Please see last sentence of the 2nd para. of **Sec. 4.2 Perspectives**, p. 44 of the latexdiff.

**Reply to RC2**

[The reviewer's input is in italic font, while our responses are in regular font.]

*It has been a great opportunity for me to read this review article. The two authors have made a great attempt to bring together – to my knowledge for the first time in a kind of systematic manner – the recent developments regarding applications of the two commonly rather disparate fields of dynamical system theory and algebraic topology in the context of climatology. Both authors are well known for their enormous work on both topics over the last decades, so it is not surprising that the resulting manuscript presents great educational material on both topics.*

Thank you very much for this supportive and encouraging review.

*There is practically only one, very minor general comment that I may raise regarding this impressive work, and I need to say that this is a very subjective one based on my own knowledge of both fields, which is far from complete. Eventually, this paper may benefit (although I am not sure if that would indeed make up for an improvement) if the authors could highlight the specific potentials for further integration of both mathematical "disciplines" for future climate (or climate-related) studies even a bit more prominently. There might be a kind of "grey zone" between a review and perspectives paper. The way the material is presented is maybe not*

*a classical review in the sense of attempting a complete coverage of the addressed field(s), which I feel absolutely comfortable with when reading this work. In such situation, I may envision this manuscript to become even more impactful when outlining somewhat more transparently the authors' perspective on a possible future research agenda, or at least parts of it. Please take this just as a suggestion, not a definite request of mine.*

We agree that there is a grey zone between a review and a perspectives paper. We have added a few paragraphs to improve the paper in its addressing future perspectives. Please see in particular the last two paras. of **Sec. 4.2 Perspectives**, pp. 47-48 of the latexdiff.

*All other comments I may have regarding certain parts of the manuscript are rather specific and/or technical and listed below:*

**Specific comments/suggestions**

*- p.3, ll.68-71: The authors essentially mention codimension-1 bifurcations (is "transverse" the same as "transcritical", the term that I am familiar with?); it might be useful to mention (here or later) the existence (and possible relevance for climate problems?) of bifurcations of codimension 2 or higher.*

Thank you for catching this typo. In fact, "transcritical" is a particular kind of codimension-1 bifurcation, in which a single solution branch preserves its existence but changes its stability across the bifurcation point: stable on one side of it and unstable on the other. "Transverse" refers to a particular condition for the intersection of two manifolds. Actually, we meant "transcritical". We have corrected this typo, inherited from the Ghil et al (1991) paper.

The suggestion that we mention the existence and relevance of codimension-2 bifurcations — such as Shilnikov, which is actually mentioned later, in Sec. 3.1, or Bogdanov-Takens — has merit. But we eventually decided that adding information on such bifurcations would break the flow of the presentation in Sec. 2.1.4 without much connection to the rest of the paper.

*- In general, Section 1.1 could benefit from a few more "tutorial" references on dynamical system theory. The choice of references in this initial overview appears partially a bit "Ghil-centric", which is fine for the more specific discussions in Section 2 focusing on contributions of the first author.*

Good point, thank you. There is a rich literature on autonomous dynamical systems, in both mathematical and physical texts, although less so in the climate sciences. The main references in the latter field of applications, due to Henk Dijkstra aside from Ghil, have already been mentioned, as has a considerable amount of NDS and RDS literature in Secs. 2 and 4. But we have added a full paragraph and several references at the head of p. 3 in the revision.

*- p.4, ll.94-95: "This complementary view of the way that dynamics and topology interact is a main motivation of the present article." I fundamentally agree with the authors' emphasis on this point. There is a lot of algebraic topology tools in classical as well as modern dynamical system theory. One recent field that seems to provide another link between the two topics, which has also found vast applications in climate science in recent years, would be complex networks. I would leave it to the authors' choice whether or not to elaborate a bit (maybe one brief paragraph in the end) on corresponding recent developments and their potentials. The authors mention this very briefly on p.43, ll. 993-995, which emphasis on a rather specific problem, but I think there might be more to that.*

We acknowledge the impressive work being pursued in complex networks and their relevance in time series analysis [Zou et al, Physics Reports, Volume 787, 2019, Pages 1-97]. Algebraic topology is not mentioned in this review, but there have been some papers applying persistent homologies to complex networks [Horak et al J. Stat. Mech. (2009) P03034; Petri et al (2013). PloS one, 8(6), e66506; De Silva & Ghrist (2007). Algebraic & Geometric Topology, 7(1), 339-358].

Horak et al. [2009] construct simplicial complexes from graphs (networks) to evaluate the robustness of a network and to distinguish different network types. Petri et al. use persistent homologies to detect particular nonlocal structures, akin to weighted holes within the link-weight network fabric. De Silva & Ghrist [2007] propose a method using persistent homologies for nonlocalized sensor networks with ad hoc wireless communications.

Despite the rapid development of computational topology and data science, the triple combination between algebraic topology, time series analysis and complex networks seems to be untouched so far. The network approach is used, however, to reconstruct the phase space, which is a prior and certainly necessary step for the analysis of the topological structure of flows from data.

The prospective directions in the field of complex networks enumerated by Zou et al [2019] share many of the challenges that are also faced by the topology of chaos.

We are grateful to the reviewer for having directed our attention to this point and have added two full paras. on this matter at the very end of the revision.

*- Somewhat related to the previous point, I might also suggest mentioning the framework of persistent homology and its potential applications, maybe even at the end of the introduction section (it appears only be briefly mentioned on p.24, l.559, before quickly focusing on the BraMAH methodology in the following).*

We have followed the reviewer's suggestion in this matter by expanding on the discussion of the Strommen et al. [2023] paper's application of a multiparameter PH method to weather regimes, in two paragraphs of Sec. 4.2; please see latexdiff.

*- Figure 1: Since it seems not to be further explained, I would find it helpful (for non-experts) if the term "isopleth" could be briefly explained.*

The etymology of "isopleth" combines "iso" with the ancient Greek word plêthos, "a great number," as in the modern English word 'plethora'. It is generically used to refer to a curve of points sharing the same value of some quantity. In his 1963 paper, Edward N. Lorenz plots the isopleths (isolines) of $X$ as a function of $Y$ and $Z$ of his attractor, to approximate surfaces formed by all points on limiting trajectories. We have added this explanation to our manuscript.

*- p.8, ll.194-195: I recommend adding a brief clarification that any bifurcation presents a kind of tipping ("B-tipping" in the language nowadays used by many authors, probably going back to Ditlivsen and co-workers?), while not every tipping behaviour in a complex system originates from an underlying bifurcation.*

There are at least two different interpretations of "tipping" and "tipping points" in the literature. One of these, emanating from Gladwell (2000) and Lenton et al. (2008), interprets tipping merely as a sudden change, whether due to a well-defined bifurcation or not. In this interpretation, a tipping point is merely a threshold.

The other interpretation sees a tipping point as a generalization to nonautonomous systems of a bifurcation point (Kuehn, 2011; Ghil, 2019]. In this case, tipping is necessarily related to a tipping point in phase-parameter space and not every jump or critical transition arises from a such a point.

We have clarified this in the revised version since both points of view have their

merits, but confusion should be avoided to the extent possible. Please see the two full paras. at the top of p. 9 in the latexdiff.

*- p.11, ll.272-273: In fact, the orbital cycles emerge from chaotic motion, but have contributions with relatively narrow (yet not exactly fixed) frequency and strongly varying amplitudes. So in reality, one would not assume that 19 kyr and 41 kyr variability components are exactly fixed (and, hence, would not have a simple integer ratio), but may lead to more complex dynamical phase locking-unlocking processes. This is far beyond the scope of the present work, but maybe the specific sentence here could be a bit reshaped to clarify what the authors actually attempt to focus on.*

We are not exactly sure whether the reviewer refers to work on the presence of chaos in the planetary system or not (e.g., Varadi, F., M. Ghil, and W. M. Kaula, 1999: Jupiter, Saturn and the edge of chaos, Icarus, 139, 286–294). We have clarified this point further, too, with a sentence at the end of the para. in question, now p. 12, ll. 303-304 of the latexdiff.

*- In Section 2.1, state vectors are denoted in bold face. In Section 2.2, however, vectorial quantities are not written in bold face anymore (e.g. x in Eq. (9), l.307 and following). I strongly suggest revising the appearance of mathematical terms for self-consistency between the different (sub)sections.*

Thank you for this correction. We are now using boldface vectors throughout.

*- p.13, l.320: Please clarify that this refers to chaotic trajectories in the Lagrangian chaos sense, not chaos of the underlying field g(t,x) itself.*

This might be a misunderstanding: the text refers to trajectories in phase space, not to fluid flow in physical space.

*- p.13, l.325: Please explain the term "pullback attraction" in a few lay words.*

Good idea, thank you. We now use simple language, as follows, "A pullback attractor is a possibly time-dependent object in a system's phase space that exhibits attraction in the sense of convergence at each time $t$ to a set, called a snapshot, to which the system's initial state at time $s$ tends to as $s$ tends to $-\infty$. This is distinct

from the forward attractors that can be defined for autonomous systems started at a fixed time $t_0$."

This highly simplified definition has been inserted in **Sec. 2.2.1 NDSs, RDSs and pullback attraction**, on p. 14, ll. 360–363 of the latexdiff.

*- p.15, l.375: There is no β in Eq. (16), only σ. Is that one meant here?*

Yes, thank you for the correction.

*- Section 2.2.3: The symbol ω used here has been previously used for a frequency in Section 2.2.2. Please consider using a different symbol.*

Thank you for noticing. We are now using the symbol $\upsilon$ in Secs. 2.2.2 and 2.2.3 for a frequency and keep the symbol $\omega$ for a realization of the driving noise, respectively.

*- p.18, ll.434-437: Could you add a brief note on suitable "mathematical" types of noise distributions?*

We have defined more precisely the connection between Brownian motion $d\eta$ and a Wiener process $\eta$, including a reference, in the discussion of Eqs. (8) and (24). Please note that we have also eliminated the redundant notation **W** in Eq. (8), replacing it by the notation $\boldsymbol{\eta}$ used later.

*- p.20, Fig. 8: The figure caption refers to a color bar that is missing in the figure.*

We now explain the color bar in words in the caption.

*- p.23, l.517: The term "braids" might be unfamiliar to many readers, so a brief explanation might be helpful. Similar for p.29, l.671 ("templexes", only introduced in the following subsection) and p.31, l.714 ("knot-holder").*

We can imagine a knot as a thin tangled rope in three-dimensional space whose ends are glued together [Prasolov and Sossinsky, Amer. Math. Soc., 1997; reference added], while a braid is a collection of strands crossing over or under each other. Both concepts became important around 1987 in the attempt to classify low dimensional (3-D) systems using topological orbit organization.

The knot approach — i.e., extracting the knot content of hyperbolic attractors — is rooted in results from Birman-Williams-Holmes, through a geometrical construction that was named template or knot-holder. The braid approach is based on results due to Thurston on the classification of 2-D diffeomorphisms and the braid content of the diffeomorphism [Natiello, M. A., 2007. The User's Approach to Topological Methods in 3d Dynamical Systems. World Scientific].

*- p.41, ll.918-926: The discussion on "wave-like" vs. "particle-like" behaviour (drawing upon a quantum mechanical analogy) reminds me a bit of the traffic jam analogy of atmospheric blocking situations by Nakamura and Huang (Science, 361 (6397), 42-47, 2018). I would wonder if the authors would see some link to this very active field of studies in climate science (persistent atmospheric wave trains, blocking, and extreme events) from their more fundamental "mathematical" (conceptual) perspectives.*

Well, the authors are familiar with the very interesting Nakamura and Huang (2018) paper and one of them is working in a separate collaboration on applying extremal length theory (Ahlfors, L., 1973. Conformal Invariants: Topics in Geometric Function Theory, American Mathematical Society) to midlatitude flow diagnostics. But the paper at hand is getting quite long and persistent anomalies have been a "very active field of studies in climate science" for over four decades, so adding yet another approach to Fig. 22 does not seem imperative at this stage. Thank you, though, for finding inspiration in our quantum mechanical analogy.

**Technical suggestions:**

*- p.1, l.3: "in the 1960s"?*

Yes, at least in the U.S. spelling (Chicago Manual of Style), which is used throughout this paper, that is correct, and not the more common "1960's." After the first mention, the '60s and '70s are correct, too.

*- p.4, l.110: typo "current"*
*- p.6, l.141: typo "the way"*

Both fixed, thank you.

*- p.6, l.142: citation style of Williams (1974) should be adjusted*

Right, thanks.

*- p.6, l.150: "in the 1990s"? (this would be consistent with l.3 and others. . . )*
*- p.6, l.160: "in the 1960s and early 1970s"*

Both OK as is, as per above.

*- p.8, ll.188 and 190: I suggest replacing "in the next subsection" by "in Section 2.1.1"*

Done.

*- p.8, l.211: It might be too much of a request, but citing some original work by Maxwell might be a quite unique thing for a paper in this journal.*

Given the early Poincaré references, adding James Clark Maxwell or Pierre Curie would seem quite appropriate. Thank you for sharing our taste for early citations, but we have found the road to the original references rather lengthy and have merely provided further perspectives on the importance of the concept; see latexdiff, p. 9, last full para.

*- p.9, l.212; p.10, l.220; p.11, ll.258-259: citation style of Ghil and Childress (1987) should be adjusted*

Thanks for noticing. Done.

*- p.11, l.257: replace "in the above figure" by "in Fig. 4"*

Thank you, done.

*- p.11, l.262: "periodicity of glacial cycles"*

Not sure what exactly this refers to: "glacial cycles" suggest a unique periodicity, "climatic variability" does not. We beg to differ.

*- p.13, l.319: the condition should read "$t_0 \leq t_1 \leq t_2$" (subscript 1 is missing)*

Correct, thanks for noticing; fixed.

*- p.13, Definition B.1 (why are definitions enumerated as B.1, B.2, etc.?): You introduce the set $\mathbb{R}_+^2$, but refer to $\mathbb{R}_{\geq}^2$ in the definition of the mappings.*

The B's have been removed; thanks for noticing their being redundant herein. The difference of notation for the nonnegative real numbers is a typo and has been fixed.

*- p.14, l.341: "translation in time"*
*- p.14, l.344: I would start the sentence with "As an example, analytical computations. . . "*
*- p.14, l.353: I am not sure if the abbreviation PBA had been defined before; I would suggest to avoid it.*
*- p.15, Eq. (17): use large brackets*
*- p.16, Eqs. (22) and (23) (and also a bit of the text in the remainder of Section 2.2.2 mixes up the symbols $\phi$ and $\varphi$. Please keep consistency.*
*- p.22, l.497: replace "While. . . " with "By contrast. . . " or something similar*
*- p.27, ll.638-639: Why is the set of ODEs (26a,26b) infinite?*
*- p.28, l.649: The solution of the Navier-Stokes equations would be a velocity field, not a streamfunction. Better write: ". . . the streamfunction. . . would not correspond to a solution. . . "*
*- p.32, l.725: "distinct chaotic attractors"*
*- p.38, l.852: type "known"*
*- p.40, l.886: In my understanding, Betti numbers are integers, so any change in this property must be necessarily discontinuous. Or do the authors want to emphasize something different here?*

Yes, they are integers: the phrase "can be quite sudden" has been changed to "is quite sudden, since they are integers."

*- p.42, l.943: "Pacific North America (PNA) pattern"*
*- p.43, l.964: omit abbreviation "TDA"*
*- p.43, l.971: ". . . and provided further. . . "*
*- p.45, l.1038: volume missing in Carlsson & Zomorodian (2007)*
*- p.46, l.1060: remove "20 pp."*
*- p.47, ll.1081-1082: this seems to be a duplicate reference*
*- p.48, l.1123: remove "41 pages"*
*- p.48, ll.1132-1133: if this is a book chapter, add page numbers; otherwise if this is the full book title, adjust citation style accordingly*

Full book, done.

*- p.49, l.1168: there is something odd with the style of this citation, please check/correct*

There is a blank space missing between "Basis." and "Contribution"; fixed.

*- p.51, ll.1232 and 1262: please give full (non-abbreviated) journal names*
*- p.52, l.1295: update reference with volume and page numbers, or provide doi if still only "online first"*
*- p.52, ll.1204-1306: this seems to be another duplicate reference*
*- p.53, l.1307: replace page numbers by proper article ID*
*- p.53, l.1327: volume missing*

All the items that are not specifically addressed in our replies have been taken care of as well. Thank you for your extremely careful reading and very constructive comments.

**Additional reference**

Prasolov, V. V. and Sossinsky, A. B.: Knots, Links, Braids and 3-Manifolds: An Introduction to the New Invariants in Low-dimensional Topology, 154, American Mathematical Society, 1997.